# Shared Semantics, Divergent Mechanisms:
# Unsupervised Feature Discovery by Aligning Semantics and Mechanisms

**Hyunjin Cho** [1] **Youngji Roh** [1] **Jaehyung Kim** [1]

## Abstract

As large language models are increasingly deployed in high-stakes settings, there is a growing need for tools that audit not only model outputs but also the internal computations that produce them. Circuit analysis is a central approach in mechanistic interpretability, but it is typically target-conditioned, explaining a single prompt paired with a chosen completion. This target-conditioned setup can obscure heterogeneity across a model's continuation distribution. We introduce *distribution-level unsupervised feature discovery*, which clusters sampled continuations using both semantic content and sequence-level mechanistic attributions, without manually specifying target outputs. Our method represents each continuation with a semantic embedding and a prefix-to-continuation attribution signature, then optimizes a rate–distortion objective that trades off semantic coherence, mechanistic consistency, and cluster granularity. Across clustering and steering analyses, the discovered clusters expose continuation modes that single-view baselines miss and provide interventional evidence that cluster signatures correspond to actionable mechanistic factors. Overall, our approach complements circuit analysis and behavioral evaluation by providing a scalable audit of the mechanisms underlying a model's continuation distribution. [1]

## 1. Introduction

Despite the remarkable capabilities of large language models (LLMs), their deployment in high-stakes domains, such as medical question answering (Tang et al., 2024b) and au-

tonomous web agents (Chae et al., 2026) demands rigorous mechanisms to audit their reliability and safety (Bommasani et al., 2021). Currently, behavioral evaluations and benchmarks serve as the dominant paradigm for assessing model performance, providing a practical way to quantify risk at scale. However, such evaluations are fundamentally limited as they are restricted to observing input–output correlation and do not provide insight into the internal computations that produce those behaviors (Jacovi & Goldberg, 2020; Doshi-Velez & Kim, 2017). This limitation has motivated growing interest in *mechanistic interpretability* (Zhang et al., 2026; Bereska & Gavves, 2024), which aims to look beyond observable outputs and explain the specific internal computations driving model behavior (Lindsey et al., 2025; Cywiński et al., 2025; Nanda et al., 2023).

Recent progress in mechanistic interpretability has centered on *circuit analysis*, which enables to reveal structured internal computations by isolating components causally responsible for a specific behavior (Marks et al., 2025; Conmy et al., 2023; Wang et al., 2023). Despite these improvements, circuit analysis typically considers a single target response specified in advance. This constraint imposes two critical limitations. First, it is unclear whether a mechanism identified for one chosen completion is generalizable to other semantically similar paraphrases, or whether it reflects a brittle dependence on a particular surface form (Fig. 1 Top). Second, even fixing a target completion, single-target analysis can obscure heterogeneity in internal computation: even for outputs with similar semantics, it could exhibit distinct patterns of internal activation that single-target analysis fails to capture (Fig. 1 Bottom).

**Contribution.** We propose a framework for unsupervised latent discovery from a prompt's continuation distribution. It complements circuit analysis by reducing reliance on hand-chosen targets and surfacing mechanistic hypotheses supported by cluster-level steering. In practice, one can first use our method to discover continuation clusters, then trace the circuits associated with a selected cluster as a follow-up analysis. This reduces the reliance on pre-specifying target outputs, which often confines standard circuit analysis to one (or a few) hand-chosen completions.

Specifically, our framework consists of three distinct stages.

---

[1]Yonsei University. Correspondence to: Jaehyung Kim <jaehyungk@yonsei.ac.kr>.

*Proceedings of the 43$^{rd}$ International Conference on Machine Learning*, Seoul, South Korea. PMLR 306, 2026. Copyright 2026 by the author(s).

[1]We provide the codebase at https://github.com/Merenova/distribution-level-feature-discovery.

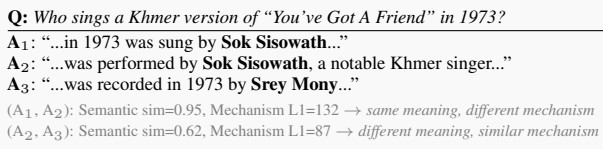

**Q:** *Who sings a Khmer version of "You've Got A Friend" in 1973?*

$A_1$: "...in 1973 was sung by **Sok Sisowath**..."
$A_2$: "...was performed by **Sok Sisowath**, a notable Khmer singer..."
$A_3$: "...was recorded in 1973 by **Srey Mony**..."

($A_1$, $A_2$): Semantic sim=0.95, Mechanism L1=132 → *same meaning, different mechanism*
($A_2$, $A_3$): Semantic sim=0.62, Mechanism L1=87 → *different meaning, similar mechanism*

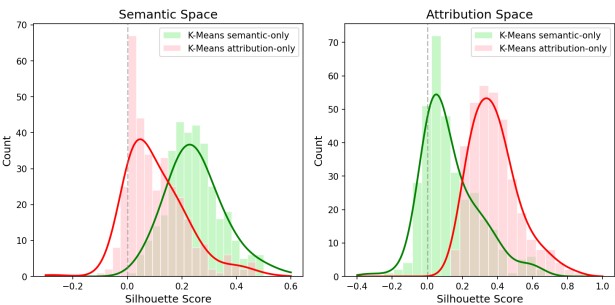

*Figure 1.* **Misalignment between semantic and mechanistic spaces.** *Top:* An example triplet where ($A_1$, $A_2$) share semantic meaning (cosine similarity) but use different internal mechanisms (attributions, L1 distance), while ($A_2$, $A_3$) differ semantically but use similar mechanisms. *Bottom:* Silhouette scores (Rousseeuw, 1987) for K-means partitions learned in one view and evaluated in both views; a partition that is clean in semantic space can be poor in attribution space, and vice versa. This observation motivates clustering continuations with both semantic and mechanistic distortions rather than treating either view as a proxy for the other.

**(1) Sampling:** We first sample diverse continuations for a given input under a fixed decoding scheme, capturing the model's inherent output diversity. **(2) Dual Representation:** We then represent each continuation through two complementary views: a semantic embedding capturing linguistic meaning, and a mechanistic embedding derived from sparse feature activations (*e.g.*, transcoders (Dunefsky et al., 2024)) that approximates a continuation-conditioned mechanistic signature of the model's computation. **(3) Joint Clustering:** Finally, we cluster these continuations using a rate–distortion objective that jointly constrains both views, producing latent modes that are coherent in meaning while exhibiting consistent internal mechanistic representation.

Our experimental results addresses three questions. **(i) Does joint clustering capture complementary structure?** We compare against semantic-only, attribution-only, and multi-view fusion baselines under rate-matched objectives. **(ii) Does the rate–distortion knob expose meaningful granularity?** We show that increasing $\beta$ yields finer continuation modes whose mechanistic signatures inherit structure from coarser clusters. **(iii) Do cluster-derived features provide interventional causal evidence?** We test whether steering along cluster-derived mechanistic directions changes relative logit difference for the target cluster.

In summary, we formulate unsupervised latent discovery as a distribution-level task for mechanistic interpretability. To our knowledge, this is the first framework that clusters a model's continuation distribution using both semantic struc-ture and sequence-level mechanistic attribution, producing human-inspectable lens for downstream circuit analysis.

## 2. Related Work

**Circuit analysis and causal attribution.** Mechanistic interpretability explains model behavior by localizing components and pathways that causally support an output. Causal mediation techniques (Vig et al., 2020; Pearl, 2001), including activation patching (Meng et al., 2022), path patching (Goldowsky-Dill et al., 2023; Wang et al., 2023), and automated circuit discovery (Conmy et al., 2023), identify behavior-relevant components through counterfactual effects. Attribution patching (Syed et al., 2024; Nanda, 2023) scales these analyses with cheaper linearized causal scores. However, most circuit workflows remain *target-conditioned*: the analyst specifies a prompt and target output, so the resulting circuit is defined relative to that pair (Wang et al., 2023; Lindsey et al., 2025). This leaves open whether a mechanism found for one surface form generalizes to semantically equivalent continuations, or whether alternative continuations rely on distinct mechanisms (Zur et al., 2025; Bigelow et al., 2025).

**Feature disentanglement and polysemanticity.** Circuit analysis is further complicated by polysemanticity: individual neurons can mix unrelated concepts under superposition (Elhage et al., 2022; Olah et al., 2020). Sparse autoencoders (Bricken et al., 2023; Huben et al., 2024) and transcoders (Dunefsky et al., 2024) learn overcomplete sparse feature bases for dense activations, clarifying *what* is represented and enabling feature-level attribution. However, these dictionaries do not by themselves determine *how* feature combinations organize downstream behavior. Prior work often studies features in isolation (Tian et al., 2025; Gao et al., 2025) or groups them by dataset-level co-occurrence (Marks et al., 2025). We instead use sparse-feature attributions as a mechanistic view of each sampled continuation, organizing features by the continuation modes.

**Unsupervised discovery of latent modes.** Our work also builds on unsupervised methods for uncovering latent structure in model representations. Some approaches identify global directions for properties such as knowledge (Burns et al., 2023) or sentiment (Tigges et al., 2023), usually assuming a linear and often binary latent axis. More recently, Marks et al. (2025) cluster dataset samples by causal feature similarity to discover recurring global circuits. We make a complementary move from the *dataset* level to the *distribution* level, clustering diverse continuations of a single prompt to expose its computational heterogeneity. Rate–distortion theory (Rose, 1998; Higgins et al., 2017) gives the organizing principle for trading off compression, semantic coherence, and mechanistic consistency, recovering prompt-specific continuation modes that purely mechanistic or dataset-level clustering can obscure.

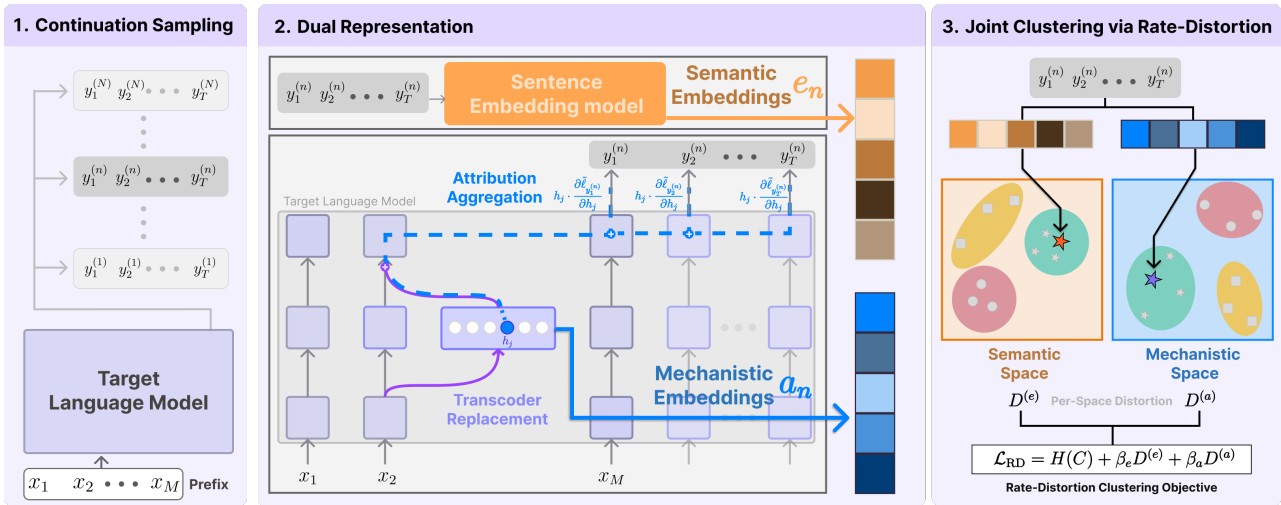

*Figure 1.* **Overview of distribution-level unsupervised feature discovery.** *Continuation Sampling:* For a prefix $\mathbf{x} = x_{1:M}$, we sample continuations $\mathbf{y}^{(n)} = y_{1:T_n}^{(n)}$ with probability weights $P_n$. *Dual Representation:* Each continuation is represented by a semantic embedding $e_n$ and a mechanistic embedding $a_n$, where $a_n$ aggregates prefix-feature effects over the continuation span. *Joint Clustering via Rate-Distortion:* Rate–distortion clustering then groups $\{(e_n, a_n, P_n)\}_{n=1}^N$, balancing semantic coherence and mechanistic consistency.

## 3. Method

In this section, we formally introduce our three-stage pipeline: sample continuations, build semantic and mechanistic views for each continuation, and cluster the weighted two-view representations. The overview is in Figure 1.

### 3.1. Stage 1: Continuation Sampling

Given an input prefix $\mathbf{x} = x_{1:M}$, we sample $N$ distinct continuations $\mathbf{y}^{(n)} = y_{1:T_n}^{(n)}$ using nucleus sampling and assign each a length-normalized probability $P_n$ (Eq. 1). These probabilities are used throughout the clustering objective: they weight each continuation's contribution to the semantic and attribution distortions, and induce the cluster prior used in the rate term. Thus, continuations with higher-probability have greater influence on both the within-cluster distortion and the effective cluster-rate penalty.

$$P_n = \exp\left(\frac{1}{T_n} \sum_{t=1}^{T_n} \log p(y_t^{(n)} \mid \mathbf{x}, y_{<t}^{(n)})\right). \quad (1)$$

### 3.2. Stage 2: Dual Representation

For each continuation $n$, we build two complementary views. The semantic embedding $e_n \in \mathbb{R}^{d_e}$ captures what the continuation says in context. The mechanistic embedding $a_n \in \mathbb{R}^{d_a}$ captures how prefix sparse features affect the continuation's logits on average. Keeping these views separate is important because semantically similar continuations can have different sequence-level attribution signatures, and mechanistically similar signatures can support semantically different continuations.

**Attribution via transcoders.** In our work, we represent the effect of features computed in MLPs on logits in the continuation span for mechanistic embeddings. Following recent works on attribution and sparse dictionary learning (Lindsey et al., 2025; Dunefsky et al., 2024), we use transcoders to decompose MLP computations into interpretable sparse features and compute direct linear attributions (Syed et al., 2024) from features to output logits. In formulas, for an MLP input $x^{(l)} \in \mathbb{R}^{d_{model}}$, a transcoder $TC$ defines sparse feature activations

$$h^{(l)}(x) = \mathrm{ReLU}\big(W_{\mathrm{enc}}^{(l)} x^{(l)} + b_{\mathrm{enc}}^{(l)}\big) \in \mathbb{R}^{d_f}, \quad (2)$$

and reconstructs the MLP output as

$$TC^{(l)}(x) = W_{\mathrm{dec}}^{(l)} h^{(l)}(x) + b_{\mathrm{dec}}^{(l)} = \sum_{f=1}^{d_f} h_f^{(l)}(x)\, d_f^{(l)} + b_{\mathrm{dec}}^{(l)},$$
$$(3)$$

where $h_f^{(l)}(x)$ is the $f$-th activation of the transcoder and $d_f^{(l)}$ is the $f$-th decoder direction (column of $W_{dec}^{(l)}$) and typically $d_f \gg d_{\mathrm{MLP}}$. Transcoders are trained to approximate the MLP module computation $\mathrm{MLP}^{(l)}(x)$ while keeping $h^{(l)}(x)$ sparse (e.g., $\|\mathrm{MLP}^{(l)}(x) - TC^{(l)}(x)\|_2^2 + \lambda \|h^{(l)}(x)\|_1$).

Let $j = (p, l, f)$ index a prefix token position $p \leq M$, layer $l$, and transcoder feature $f$. For a continuation token $y_t^{(n)}$, we measure the direct linear attribution of feature $j$ to the demeaned logit $\tilde{\ell}_{y_t^{(n)}} = \ell_{y_t^{(n)}} - |V|^{-1} \sum_{v \in V} \ell_v$ by multiplying feature activation and gradient (Lindsey et al., 2025; Syed et al., 2024):

$$A_{j \to y_t^{(n)}} = h_j \cdot \frac{\partial \tilde{\ell}_{y_t^{(n)}}}{\partial h_j}. \quad (4)$$

The continuation-level attribution vector averages this token-wise effect over the sampled continuation span:

$$a_n[j] = \frac{1}{T_n} \sum_{t=1}^{T_n} A_{j \to y_t^{(n)}}. \tag{5}$$

Thus $a_n$ can be interpreted as a sequence-level prefix-to-continuation effect. Note that the index $j = (p, l, f)$ is retained, so discovered cluster signatures can still be inspected as position-layer-feature hypotheses for downstream circuit analysis. We provide more details on derivation and the interpretation of the effect in Appendix C.

**Normalization.** Semantic embeddings are L2-normalized: $e_n \leftarrow e_n/\|e_n\|_2$, so that squared L2 distance corresponds to cosine dissimilarity. Attribution vectors are RMS-normalized: $a_n \leftarrow a_n/\mathrm{RMS}(a_n)$, which preserves sparsity structure and sign patterns while standardizing scale.

### 3.3. Stage 3: Joint Clustering via Rate-Distortion

We formulate latent discovery as a rate–distortion problem: find few clusters (low rate) that faithfully capture per-continuation patterns (low distortion) across both views. Unlike $K$-means, which fixes $K$ a priori, our framework lets $K$ emerge from optimization and accommodates multiple distortion terms with interpretable trade-offs. We call this joint-clustering framework *RD-clustering*.

Rate–distortion theory formalizes a trade-off between compression and fidelity: lower rate uses fewer effective codes, while lower distortion preserves more information. Here, cluster assignments are the compressed representation, the entropy $H(C)$ is the rate, and the semantic and mechanistic reconstruction errors are the two distortion terms.

**Problem formulation.** We represent each $n$-th continuation as $(e_n, a_n, P_n)$. Each cluster $k \in \{1, \dots, K\}$ has centers $(\boldsymbol{\mu}_k^{(e)}, \boldsymbol{\mu}_k^{(a)})$ and prior $\pi_k = W_k / \sum_{k'} W_{k'}$, where $W_k = \sum_{n:c(n)=k} P_n$ is the total probability mass in cluster $k$ ($c(n)$ is the assignment function which returns the cluster index $k$ where continuation $n$ is assigned).

**Rate term.** The rate term measures model complexity—how many bits are needed to encode cluster assignments. We use the entropy of the cluster prior:

$$H(C) = -\sum_{k=1}^{K} \bar{P}_k \log \bar{P}_k, \tag{6}$$

where $C$ denotes the assignment configuration over all continuations and $\bar{P}_k = W_k / \sum_{k'} W_{k'}$. Lower entropy means fewer effective clusters and a simpler clustering structure.

**Distortion terms.** The distortion terms measure representation error—how well cluster centers summarize their

members. We define probability-weighted distortions in both views:

$$D^{(e)} = \sum_{k=1}^{K} \bar{P}_k \cdot \mathrm{Var}_k^{(e,w)},$$
$$\mathrm{Var}_k^{(e,w)} = \frac{1}{W_k} \sum_{n:c(n)=k} P_n \|e_n - \boldsymbol{\mu}_k^{(e)}\|_2^2, \tag{7}$$

$$D^{(a)} = \sum_{k=1}^{K} \bar{P}_k \cdot \mathrm{Var}_k^{(a,w)},$$
$$\mathrm{Var}_k^{(a,w)} = \frac{1}{W_k} \sum_{n:c(n)=k} P_n \|a_n - \boldsymbol{\mu}_k^{(a)}\|_1. \tag{8}$$

Semantic distortion uses L2 distance; attribution distortion uses L1, which is robust to outliers in sparse vectors.

**Joint objective.** We set the clustering objective as follows:

$$\mathcal{L}_{\mathrm{RD}} = \underbrace{H(C)}_{\text{Rate}} + \underbrace{\beta_e D^{(e)} + \beta_a D^{(a)}}_{\text{Distortion}} \tag{9}$$

We parameterize $\beta_e = \gamma\beta$ and $\beta_a = (1-\gamma)\beta$, where $\beta > 0$ controls the rate-distortion tradeoff ($\beta \to 0$ yields $K = 1$; $\beta \to \infty$ yields $K = N$) and $\gamma \in [0, 1]$ balances two views.

**Optimization.** We optimize Equation (9) via alternating minimization. The **E-step** updates assignments:

$$c(n) = \underset{k}{\mathrm{argmin}} \Big[ -\log \bar{P}_k$$
$$+ \beta_e \|e_n - \boldsymbol{\mu}_k^{(e)}\|_2^2 + \beta_a \|a_n - \boldsymbol{\mu}_k^{(a)}\|_1 \Big], \tag{10}$$

where the $-\log \bar{P}_k$ term acts as an entropic prior favoring larger clusters. The **M-step** updates centers:

$$\boldsymbol{\mu}_k^{(e)} = \frac{1}{W_k} \sum_{n:c(n)=k} P_n \cdot e_n,$$
$$\boldsymbol{\mu}_k^{(a)} = \mathrm{wmedian}\left(\{a_n\}_{n:c(n)=k}, \{P_n\}\right). \tag{11}$$

where wmedian denotes the *coordinate-wise* weighted median, which is a minimizer of the sum of L1 distance.

**Adaptive operations.** Starting from $K = 1$, we adaptively adjust the number of clusters through two operations after each E- and M-step: **Split** discovers new structure by dividing heterogeneous clusters, while **Junk** removes spurious clusters that do not capture meaningful patterns.

**Proposition 3.1** (Split Criterion). *Cluster $k$ should split into $(k_1, k_2)$ with mass fractions $(\alpha, 1 - \alpha)$ if:*

$$\beta_e \cdot \Delta D_k^{(e)} + \beta_a \cdot \Delta D_k^{(a)} > \bar{P}_k \cdot H_{bin}(\alpha) \tag{12}$$

*where $H_{bin}(\alpha)$ is binary entropy and $\Delta D_k^{(\cdot)}$ is the distortion reduction.*

**Proposition 3.2** (Junk Criterion). *Cluster $k$ should be removed and the members are reassigned if:*

$$\Delta H > \beta_e \cdot \Delta D^{(e)} + \beta_a \cdot \Delta D^{(a)} \qquad (13)$$

*where $\Delta H > 0$ is the entropy reduction and $\Delta D^{(\cdot)} > 0$ is the distortion increase.*

Intuitively, a cluster splits when it contains distinct subgroups whose separation reduces representation error more than the cost of adding a new cluster. For Junk operation, a cluster is removed when it captures noise or outliers rather than a coherent mode. In detail, we use PCA-based initialization for Split operation (Appendix B.5). Overall details for derivations are in Appendix A.

**Proposition 3.3** (Convergence). *The algorithm produces a monotonically non-increasing sequence $\{\mathcal{L}_{RD}^{(t)}\}$ that converges (see Appendix A.5).*

The overall algorithm is presented in Algorithm 1. The per-iteration cost is $O(NKd)$ where $d = \max(d_e, d_a)$.

---

**Algorithm 1** Rate-Distortion Clustering

---

**Require:** continuations $\{(e_n, a_n, P_n)\}_{n=1}^{N}$, parameters $(\beta, \gamma)$
**Ensure:** assignments $\{c(n)\}$, centers $\{(\boldsymbol{\mu}_k^{(e)}, \boldsymbol{\mu}_k^{(a)})\}$
1: Initialize: $K \leftarrow 1$, assign all points to a single cluster
2: **repeat**
3:     **E-step:** Update assignments via Equation (10)
4:     **M-step:** Update centers via Equation (11)
5:     **Split:** For each cluster, split if Equation (12) holds
6:     **Junk:** Remove clusters satisfying Equation (13)
7: **until** convergence
8: **return** assignments, centers

---

# 4. Analysis of Rate-Distortion Dynamics

In this section, we empirically validate three properties of the proposed rate-distortion framework: (1) $\gamma$ controls structural alignment between views, (2) $\beta$ controls cluster resolution, which could show structures with different levels of granularity, and (3) adaptive operations (split/junk) achieve better rate-distortion tradeoffs than fixed-$K$ baselines.

**Experimental setup.** We validate our framework using **Qwen3-8B** (Yang et al., 2025) on **AmbigQA** (Min et al., 2020), where ambiguity in questions requires the model to resolve multiple valid interpretations. We evaluate on 300 questions, treating each question as the input and the model's sampled answers as continuations; for each question, we generate 500 distinct responses. We then use **EmbeddingGemma** (Vera et al., 2025) as a semantic embedding model, `mwhanna/qwen3-8b-transcoders` as a mechanism (attribution) embedding model. After rollout,

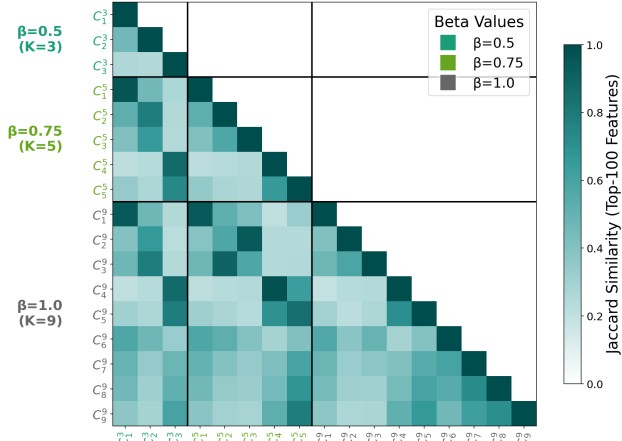

*Figure 3.* **Cluster mechanistic similarity across $\beta$ values.** Jaccard similarity of top-100 mechanistic features between clusters at $\beta \in \{0.5, 0.75, 1.0\}$ for the prompt "Who sang the song 'If god was one of us?'". The notation $C_i^K$ denotes cluster index $i$ among the $K$ clusters produced at a given $\beta$. Higher cross-$\beta$ similarity reflects parent–child structure across resolutions; a score-labeled version is provided in Fig. 13.

we use $\beta \in [1.0, 2.0]$, $\gamma \in [0.1, 0.9]$ as $RD$-clustering parameters. The results on Qwen3-4B are in Appendix E.5. While our main results use **Qwen3-8B** on **AmbigQA**, we additionally report results on **MMLU** (Hendrycks et al., 2021) using **Qwen3** models a **Gemma3** model (Team et al., 2025) in Appendix E.2.

## 4.1. $\beta$–$\gamma$ Sweep Under the Rate-Distortion Objective

We evaluate the rate $H$, the distortions $D^{(e)}$ and $D^{(a)}$, and the mixed distortion $D_\gamma = \gamma D^{(e)} + (1 - \gamma)D^{(a)}$ after RD clustering. We compare against K-means baselines clustered separately in each view, denoted KM-S and KM-A. The comparison is *rate-matched*: for each RD solution, we choose the baseline cluster count whose assignment entropy $H(C)$ is closest to that of the RD clustering. As shown in Table 1, RD clustering achieves the lowest mixed distortion in **94%** of settings. By contrast, KM-S and KM-A exhibit complementary failure modes: KM-S preserves semantic coherence but incurs high mechanistic distortion, whereas KM-A reduces mechanistic distortion at the cost of semantic coherence. The advantage is most pronounced at intermediate $\gamma$, where both views are important, although RD remains competitive near the single-view extremes.

## 4.2. $\beta$ for Different Granularity of Structures

We examine how annealing $\beta$ can be helpful to discover meaningful behaviors with various granularity, using a representative example (Figure 3). For the prompt "Who sang the song 'If god was one of us?'", we first sample 500 continuations, and cluster with our $RD$-clustering method with a fixed $\gamma (= 0.5)$ annealing $\beta$ from $\{0.5, 0.75, 1.0\}$. Then, for each cluster, we normalize each

| Method | $\gamma$ | $\beta$ 0.50 | | | | 0.75 | | | | 1.00 | | | | 1.25 | | | |
|---|---|---|---|---|---|---|---|---|---|---|---|---|---|---|---|---|---|
| | | $H$ | $D^{(e)}$ | $D^{(a)}$ | $D_\gamma$ | $H$ | $D^{(e)}$ | $D^{(a)}$ | $D_\gamma$ | $H$ | $D^{(e)}$ | $D^{(a)}$ | $D_\gamma$ | $H$ | $D^{(e)}$ | $D^{(a)}$ | $D_\gamma$ |
| RD | 0.10 | 1.70 | 0.19 | 9.68 | **8.73** | 2.29 | 0.17 | 7.97 | **7.19** | 2.60 | 0.16 | 7.22 | **6.52** | 2.69 | 0.16 | 7.01 | **6.33** |
| | 0.30 | 1.29 | 0.21 | 11.12 | **7.85** | 1.99 | 0.18 | 8.71 | **6.15** | 2.36 | 0.17 | 7.75 | **5.48** | 2.60 | 0.16 | 7.22 | **5.10** |
| | 0.50 | 0.75 | 0.23 | 13.56 | **6.89** | 1.40 | 0.20 | 10.73 | **5.46** | 1.87 | 0.18 | 9.15 | **4.66** | 2.28 | 0.17 | 7.96 | **4.06** |
| | 0.70 | 0.28 | 0.25 | 16.33 | **5.07** | 0.62 | 0.23 | 14.20 | **4.42** | 1.11 | 0.21 | 11.91 | **3.72** | 1.47 | 0.20 | 10.44 | **3.27** |
| | 0.90 | 0.00 | 0.26 | 18.85 | **2.12** | 0.03 | 0.26 | 18.41 | **2.07** | 0.08 | 0.25 | 17.93 | **2.02** | 0.16 | 0.25 | 17.19 | **1.94** |
| KM-S | 0.10 | 1.65 | 0.14 | 43.61 | 39.27 | 2.16 | 0.11 | 40.11 | 36.11 | 2.35 | 0.10 | 38.78 | 34.91 | 2.40 | 0.10 | 38.72 | 34.86 |
| | 0.30 | 1.34 | 0.15 | 45.43 | 31.85 | 1.93 | 0.12 | 41.50 | 29.09 | 2.20 | 0.11 | 39.78 | 27.88 | 2.35 | 0.10 | 38.86 | 27.24 |
| | 0.50 | 0.88 | 0.18 | 49.15 | 24.66 | 1.42 | 0.15 | 44.95 | 22.55 | 1.83 | 0.13 | 42.07 | 21.10 | 2.15 | 0.11 | 39.85 | 19.98 |
| | 0.70 | 0.65 | 0.19 | 51.00 | 15.43 | 0.81 | 0.18 | 49.85 | 15.08 | 1.18 | 0.16 | 46.90 | 14.18 | 1.49 | 0.14 | 44.40 | 13.42 |
| | 0.90 | 0.59 | 0.19 | 51.40 | 5.31 | 0.59 | 0.19 | 51.40 | 5.31 | 0.59 | 0.19 | 51.40 | 5.31 | 0.62 | 0.19 | 51.08 | 5.28 |
| KM-A | 0.10 | 1.67 | 0.18 | 31.32 | 28.20 | 2.17 | 0.16 | 26.50 | 23.87 | 2.37 | 0.16 | 24.97 | 22.49 | 2.43 | 0.16 | 24.74 | 22.28 |
| | 0.30 | 1.34 | 0.19 | 34.93 | 24.51 | 1.94 | 0.17 | 28.43 | 19.95 | 2.22 | 0.16 | 26.13 | 18.34 | 2.37 | 0.16 | 25.03 | 17.57 |
| | 0.50 | 0.88 | 0.21 | 40.62 | 20.41 | 1.44 | 0.19 | 33.73 | 16.96 | 1.83 | 0.18 | 29.48 | 14.83 | 2.17 | 0.17 | 26.32 | 13.24 |
| | 0.70 | 0.65 | 0.21 | 44.28 | 13.43 | 0.82 | 0.21 | 41.57 | 12.62 | 1.19 | 0.20 | 36.49 | 11.09 | 1.50 | 0.19 | 32.87 | 9.99 |
| | 0.90 | 0.60 | 0.22 | 45.08 | 4.70 | 0.60 | 0.22 | 45.08 | 4.70 | 0.60 | 0.22 | 45.08 | 4.70 | 0.61 | 0.22 | 44.81 | 4.68 |

*Table 1.* **RD vs. K-means at matched rate $H$.** For each $(\beta, \gamma)$ configuration, we find the K-means $K$ that yields entropy closest to the RD result, enabling a fair rate-matched comparison. We report $D_\gamma = \gamma D^{(e)} + (1-\gamma)D^{(a)}$. Bold indicates lowest $D_\gamma$. **KM-S**: K-means on semantic embedding space only; **KM-A**: K-means on attribution embedding space only. RD wins 94% of comparisons.

attribution center with the mean of centers with the same $\beta$ (*i.e.*, $\tilde{\mu}_i^{(a)} = \mu_i^{(a)} - \frac{1}{K}\sum_k \mu_k^{(a)}$), to specify distinct features for each cluster. Finally, we compute the Jaccard similarity of features with top-100 magnitude, which measures the ratio of co-activated features. In this section, we use a notation $C_i^k$, where $i$ indicates the index of the cluster within the $\beta$, and $k$ as the total number of clusters with the same $\beta$, and higher $k$ implies higher $\beta$.

At $\beta = 0.5$ ($K = 3$), the algorithm identifies three coarse **stylistic** modes rather than semantic distinctions:

- $C_1^3$: Formatted solo artist—*"The song **"If God Was One of Us"** was performed by **Gloria Gaynor**"*
- $C_2^3$: Band/group with elaboration—*"...performed by **The Highwaymen**, a country music supergroup consisting of **Johnny Cash, Willie Nelson**..."*
- $C_3^3$: Minimal formatting—*"The song "If God Was One of Us" was performed by **Dusty Springfield**"* (no bold on song title)

Critically, the *semantic content* (artist names) varies widely within each cluster—the model hallucinates many different artists—but the *generation style* determines cluster membership. This results reveals that factual recall is more fine-grained rather than sentence styles.

At $\beta = 0.75$ ($K = 5$), finer distinctions emerge. The Jaccard heatmap reveals the split structure: $C_2^3$ (band elaboration) from $\beta = 0.5$ shows high similarity with both $C_2^5$ (=0.79) and $C_3^5$ (=0.65) at $\beta = 0.75$, indicating that $C_2^3$ splits into:

- $C_2^5$: Supergroup member enumeration—*"...performed by **The Highwaymen**, a supergroup consisting of **Johnny Cash, Willie Nelson, Waylon Jennings**..."*
- $C_3^5$: Genre-based band descriptions—*"...performed by **The Temptations**, a legendary Motown group"*

Meanwhile, $C_1^3$ (solo artist) remain relatively stable, each mapping to a single dominant cluster at $\beta = 0.75$ ($C_x^5$), visible as a distinct high-similarity cell among clusters.

At $\beta = 1.0$ ($K = 9$), the algorithm further resolves earlier clusters which represents the more subtle concepts. The heatmap shows further splitting on $C_5^5$ into $C_4^9$ to $C_9^9$, which varies group names ($C_5^9$) and Solo singer with first/last name ($C_9^9$) as similar to split from $C_2^1$ into $C_2^5$ and $C_3^5$ (but with no formatting on the song title).

To summarize, the heatmap quantifies this hierarchical structure: most clusters within the same $\beta$ share $< 50\%$ of their top mechanistic features, but with elevated values ($> 0.6$) along parent-child lineages. Overall results confirm that successive splits activate genuinely distinct neural pathways, which show the effectiveness in granularity by annealing $\beta$.

### 4.3. $\gamma$ for Balance Between Mechanism and Semantic

The hyperparameter $\gamma$ governs the trade-off between semantic distortion $D_e$ and attribution distortion $D_a$ in the joint objective $D_\gamma = \gamma D_e + (1-\gamma)D_a$. Ideally, we would demonstrate this effect at fixed rate $H$; however, constraining the rate while performing cluster reassignment and adaptive operations is intractable. Instead, we visualize the qualitative effect of $\gamma$ on clustering structure across both representation spaces. Figure 4 shows the example that cluster assignments projected onto semantic (top) and attribution (bottom) spaces for varying $\gamma$ at fixed $\beta = 2$. At low $\gamma$ (e.g., $\gamma = 0.1$), the objective prioritizes attribution distortion, yielding clusters that are compact in attribution space but potentially overlapped in semantic space (dashed ellipse, bottom-left). This produces fine-grained partitions that group generations by shared computational mechanisms, regardless of their surface-level semantic similarity.

Conversely, at high $\gamma$ (*e.g.*, $\gamma = 0.9$), semantic distortion

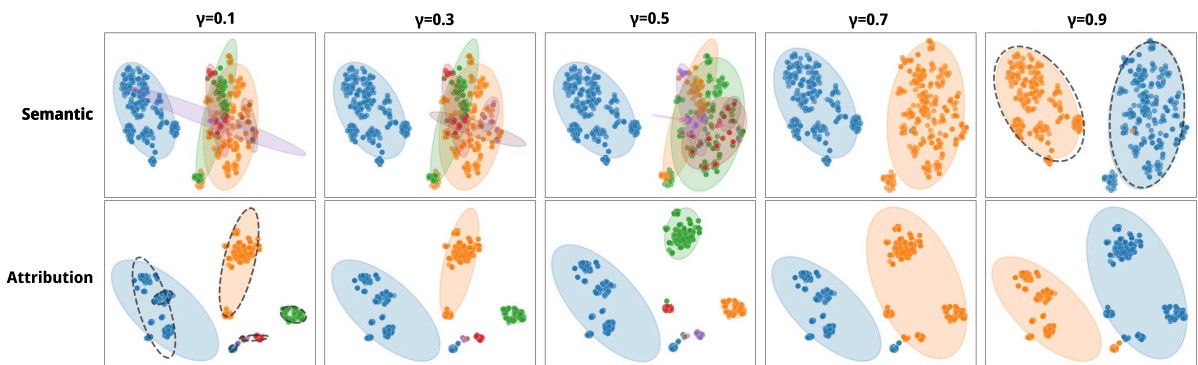

*Figure 4.* **Cluster structure dynamics across $\gamma$ values.** Each plots are drawn by sweeping $\gamma \in [0.1, 0.9]$, keeping consistent rate by varying $\beta \approx 0.9$. Dashed ellipses indicate K-means boundaries in each space with the same number of clusters $k$ as $RD$-clustering.

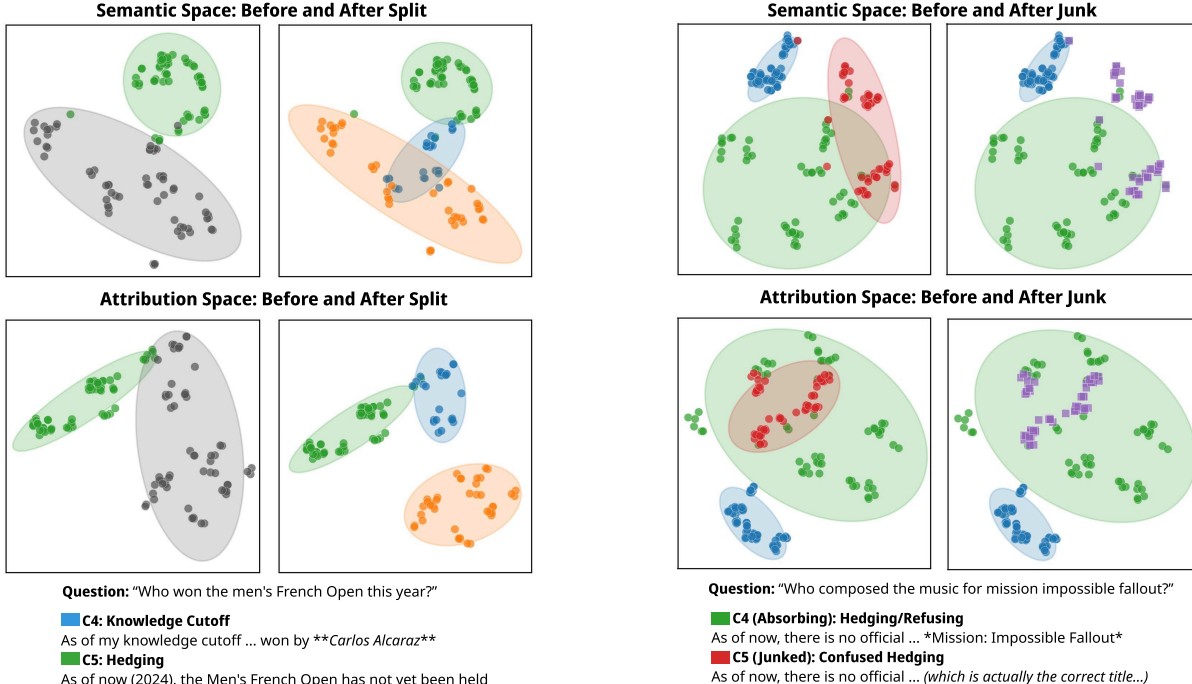

*Figure 5.* **Split operation example.** Top: semantic embedding view; bottom: mechanistic (attribution) view. A coarse cluster is split into two tighter subclusters, separating cluster C4 (*knowledge-cutoff + answer*) from cluster C5 (*as-of-now* framing).

*Figure 6.* **Junk operation example.** Top: semantic embedding view; bottom: mechanistic (attribution) view. Clusters C4 and C5 overlap strongly, so Junk removes cluster C5 and reassigns its points to cluster C4, trading a small distortion increase for a larger rate reduction.

dominates, resulting in clusters that are similar to K-means with semantic embedding space but may span noisy attribution patterns (dashed ellipse, top-right). The number of clusters also decreases, as semantically similar generations are grouped together even when they arise from different underlying mechanisms. At intermediate values ($\gamma = 0.5$), the clustering achieves a balance, producing partitions that are reasonably coherent in both spaces. This demonstrates that $\gamma$ provides a continuous control for navigating the trade-off between mechanism-based and embedding-based views of generation diversity—a flexibility unavailable to single-space clustering methods like K-means.

## 4.4. Adaptive Operations: Split and Junk

In this section, we demonstrate how Split and Junk operations work in optimizing the rate-distortion objective $\mathcal{L}_{\mathrm{RD}}$.

**Split operation: discovering latent structure.** Figure 5 illustrates a representative Split step. Starting from a coarse partition, the algorithm targets a high-dispersion cluster and tests whether dividing it into two subclusters improves the rate–distortion objective. A split is accepted when the drop in joint distortion outweighs the added rate cost ($L_{\mathrm{RD}}$ decreases from 4.61 to 4.57), as captured by Equation (12).

In this example, the pre-split cluster mixes two generation modes that are similar in one view but heterogeneous in the other; after splitting, both subclusters become more coherent in *both* semantic and mechanistic spaces. Thus, Split operation is necessary to resolve mixed clusters and expose the latent structure that a single view would blur.

**Junk operation: rejecting noise.** Figure 6 shows a representative Junk operation. The algorithm detects a small cluster whose points are not meaningfully separated: its center overlaps with another cluster in both views, so keeping it mostly increases the rate term. The junk criterion (Equation (13)) triggers because removing the cluster yields a larger rate reduction than the distortion increase after reassignment, decreasing $\mathcal{L}_{\mathrm{RD}}$ (here from 5.54 to 5.42). Thus, Junk operation is necessary to prevent over-segmentation by pruning redundant or noisy micro-clusters.

## 5. Causal Validation via Intervention

While the previous sections show that our clusters are coherent in both semantic and mechanistic views, a critical question remains: do the corresponding mechanistic directions causally affect the continuations associated with each cluster, or are they merely correlated byproducts (Joshi et al., 2026; Geiger et al., 2025)? We study this at the level of *intervention* (Pearl, 2009): if a cluster-level feature direction captures a causal factor, then increasing or suppressing the associated sparse features should shift the model's relative preference toward or away from continuations in that cluster. We test this prediction using activation steering (Lee et al., 2025; Arditi et al., 2024; Tang et al., 2024a) and measure steering-induced changes in target-cluster logits.

For each cluster $k$, we select a single representative continuation $n_k^\star$ and use its attribution vector as the cluster's steering direction. Concretely, we choose the cluster *medoid*[2] as the member that is closest to the learned two-view prototype under the same distortion used in clustering:

$$n_k^\star := \arg\min_{n:c(n)=k} P_n\left(\beta_e \|e_n - \boldsymbol{\mu}_k^{(e)}\|_2^2 + \beta_a \|a_n - \boldsymbol{\mu}_k^{(a)}\|_1\right),$$
$$(14)$$

and define the cluster steering vector as

$$a^k := a_{n_k^\star}. \quad (15)$$

Using a medoid yields an *interpretable* steering direction as it corresponds to an actual observed continuation in the cluster, rather than an abstract weighted median prototype.

**Steering methodology.** For each cluster $k$, we select the top-$B$ features by magnitude from the medoid attribution vector $a^k$: $\mathcal{F}_k = \text{top-}B\{f : |a^k[f]|\}$. During generation,

---

[2]A medoid is the data point that minimizes the average distance to all other points in the cluster.

we intervene on these features $f \in \mathcal{F}_k$ by scaling their activations:

$$\Delta h_f = h_f \cdot \left(\epsilon \cdot \text{sign}(a^k[f])\right), \quad (16)$$

where $\epsilon \in [-1, 1]$ controls intervention strength. At $\epsilon = -1$ (ablation), features with positive attribution are zeroed; at $\epsilon = +1$ (amplification), they are doubled. Note that each feature corresponds to a specific tuple $(i, l, f)$ of token position, layer, and transcoder feature index. We apply steering locally at each tuple during the forward pass (i.e. intervening only at the exact position-layer) where that feature activates in the prefix, rather than globally across all positions or throughout the continuation. As mentioned before, transcoders imitate the MLP layer's input-output behavior, the target feature difference $\Delta h_f$ is directly added to layer output redisual stream $\mathbf{r}$ after decoding along with original MLP output.

$$\mathbf{r}_{\text{pre}}^{(l+1)} = \mathbf{r}_{\text{mid}}^{(l)} + \text{MLP}^{(l)}(\mathbf{r}_{\text{mid}}^{(l)}) + \sum_{f \in \mathcal{F}_k^{(l)}} \left(\Delta h_f d_f^{(l)} + b_{dec}^{(l)}\right)$$
$$(17)$$

where $\mathbf{r}_{\text{mid}}^{(l)}$ is the residual stream at layer $l$ after attention, $\mathbf{r}_{\text{pre}}^{(l+1)}$ is the residual stream at layer $l+1$ before attention.

We quantify how steering redistributes model preference across clusters using a *cluster-centered logit* score, defined as the sum of *demeaned logits* assigned to continuations in cluster $k$. We define the cluster-centered logit as

$$L_k^{(\epsilon)} := \sum_{n:c(n)=k} \ell_n^{(\epsilon)} - \bar{\ell}_n^{(\epsilon)} = \sum_{n:c(n)=k} \frac{1}{T} \sum_{i=1}^{T} \left(\ell_{n_i}^{(\epsilon)} - \bar{\ell}_{n_i}^{(\epsilon)}\right),$$
$$(18)$$

where $\ell_{n_i}^{(\epsilon)}$ is the steered logit with strength $\epsilon$ at token $y_i$ and report its steering-induced change:

$$\Delta L_k^{(\epsilon)} = L_k^{(\epsilon)} - L_k^{(0)}. \quad (19)$$

Intuitively, $\Delta L_k^{(\epsilon)}$ measures how steering shifts the model's *relative logit preference* toward the set of continuations assigned to cluster $k$, without conflating the effect with global logit shifts across the vocabulary.

**Setup.** For the main AmbigQA experiment, we cluster 500 continuations with $\beta \in 0.75, 1.00, 1.25$ and $\gamma \in 0.3, 0.5, 0.7$, select the top-$B$ features with $B \in 5, 10$ from each cluster, subsample 10 continuations per cluster, and steer with $\epsilon \in -1.0, -0.5, -0.1, 0.1, 0.5, 1.0$. We evaluate clusters with $2 \leq K \leq 10$ because steering all clusters is computationally expensive. We compare $RD$ medoid directions against two baselines. **KM-Sem** uses medoids from semantic-only K-means clusters with matched $K$, testing how much mechanistic faithfulness can be obtained from semantic similarity alone. **Single** uses the attribution vector of a randomly selected continuation from the same $RD$

| Method | Metric | β | | | | | | | | |
| | | 0.75 | | | 1.00 | | | 1.25 | | |
| | γ | 0.3 | 0.5 | 0.7 | 0.3 | 0.5 | 0.7 | 0.3 | 0.5 | 0.7 |
| **RD** | $\rho_s$ | 0.30 | 0.22 | 0.27 | 0.35 | 0.32 | 0.24 | 0.26 | 0.30 | 0.20 |
| | $\rho$ | 0.31 | 0.23 | 0.28 | 0.39 | 0.33 | 0.33 | 0.31 | 0.32 | 0.21 |
| | $\epsilon_{-1.0}$ | -1.75 | -2.16 | -1.70 | -1.66 | -1.80 | -2.03 | -0.91 | -1.17 | -1.96 |
| | $\epsilon_{-0.5}$ | -0.79 | -0.75 | -0.70 | -0.59 | -0.63 | -0.69 | -0.36 | -0.49 | -0.69 |
| | $\epsilon_{-0.1}$ | -0.17 | -0.16 | -0.19 | -0.25 | -0.13 | -0.11 | -0.15 | -0.19 | -0.18 |
| | $\epsilon_{0.1}$ | 0.19 | 0.12 | 0.09 | 0.08 | 0.19 | 0.18 | 0.01 | 0.07 | 0.09 |
| | $\epsilon_{0.5}$ | 0.47 | 0.47 | 0.38 | 0.28 | 0.49 | 0.39 | 0.11 | 0.33 | 0.30 |
| | $\epsilon_{1.0}$ | 0.45 | 0.58 | 0.50 | 0.56 | 0.63 | 0.43 | 0.28 | 0.58 | 0.34 |
| **KM-Sem** | $\rho_s$ | -0.02 | -0.01 | 0.05 | -0.08 | -0.02 | 0.04 | -0.13 | -0.11 | -0.01 |
| | $\rho$ | -0.02 | -0.01 | 0.05 | -0.09 | -0.02 | 0.04 | -0.13 | -0.11 | -0.01 |
| | $\epsilon_{-1.0}$ | -0.72 | -0.59 | -0.55 | -0.08 | -0.38 | -0.84 | -0.08 | 0.01 | -0.52 |
| | $\epsilon_{-0.5}$ | -0.13 | -0.10 | -0.14 | 0.01 | 0.06 | -0.14 | 0.02 | 0.18 | -0.17 |
| | $\epsilon_{-0.1}$ | 0.00 | -0.00 | -0.04 | -0.06 | 0.16 | 0.03 | 0.01 | 0.06 | -0.06 |
| | $\epsilon_{0.1}$ | 0.04 | 0.00 | 0.01 | -0.05 | 0.12 | 0.09 | -0.08 | -0.08 | -0.06 |
| | $\epsilon_{0.5}$ | -0.22 | -0.14 | -0.08 | -0.27 | -0.05 | -0.02 | -0.24 | -0.45 | -0.17 |
| | $\epsilon_{1.0}$ | -0.77 | -0.73 | -0.44 | -0.59 | -0.51 | -0.50 | -0.44 | -0.87 | -0.76 |
| **Single** | $\rho_s$ | 0.13 | 0.13 | 0.14 | 0.22 | 0.14 | 0.12 | 0.17 | 0.17 | 0.10 |
| | $\rho$ | 0.14 | 0.14 | 0.15 | 0.23 | 0.15 | 0.13 | 0.19 | 0.17 | 0.12 |
| | $\epsilon_{-1.0}$ | -0.92 | -1.15 | -0.95 | -1.38 | -1.13 | -1.11 | -1.31 | -1.25 | -1.19 |
| | $\epsilon_{-0.5}$ | -0.35 | -0.41 | -0.38 | -0.51 | -0.45 | -0.37 | -0.46 | -0.48 | -0.40 |
| | $\epsilon_{-0.1}$ | -0.08 | -0.07 | -0.06 | -0.17 | -0.09 | -0.02 | -0.20 | -0.14 | -0.07 |
| | $\epsilon_{0.1}$ | 0.04 | 0.09 | 0.06 | 0.07 | 0.08 | 0.08 | 0.04 | 0.12 | 0.03 |
| | $\epsilon_{0.5}$ | 0.04 | 0.18 | 0.16 | 0.16 | 0.18 | 0.12 | 0.16 | 0.25 | 0.09 |
| | $\epsilon_{1.0}$ | 0.07 | 0.16 | 0.18 | 0.23 | 0.20 | 0.07 | 0.13 | 0.27 | 0.07 |

*Table 2.* **Steering results by $\beta$ and $\gamma$ configuration top-$B$ and sign types are pooled.** Rows show methods with metric sub-rows. Gray rows: correlation metrics; white rows: steering effect at strength $\epsilon$. **RD**: $RD$-clustering medoids; **KM-Sem**: medoids from semantic-only K-means clusters; **Single**: Single attribution vector randomly chosen from $RD$ cluster continuations. We provide the full results in Table 11.

cluster, approximating the generalizability of conventional single-target analysis. For each method, we report Pearson and Spearman correlations between steering strength $\epsilon$ and logit change $\Delta L_k^{(\epsilon)}$, averaged across clusters and prefixes.

**Results.** Table 2 shows that $RD$-clustering with medoid-based directions achieves the strongest and most consistent monotonic response: increasing $\epsilon$ yields larger $\Delta L_k^{(\epsilon)}$ and higher correlation across configurations. In contrast, KM-Sem exhibits near-zero correlations, suggesting that semantic similarity alone does not reliably identify causally relevant features for cluster-specific continuations. The Single baseline yields positive but weaker correlations than $RD$-medoid, indicating that a randomly chosen direction often fails to represent the cluster-level mechanism, consistent with the brittleness of single-continuation causal claims.

**Case study.** We examine a qualitative study of two representative continuation clusters. Table 3 illustrates that the $RD$ medoid transfers more reliably to held-out continuations. In the production-funding cluster, amplification raises held-out centered logits by $+0.598$ ($RD$) versus $+0.068$ (Single); in the not-recognized cluster, $RD$ stays directional ($+0.239$ amplification, $-0.228$ ablation) while Single reverses sign on average. This result suggests the discovered direction is not tied to one surface string but captures a cluster-level causal factor shared across distinct continuations.

---

**Q:** *Who pays for the renovations on* Holmes Next Generation*?*

| Source | idx | Continuation | $\Delta L_{+1}$ | Pos. | $\Delta L_{-1}$ |
|---|---|---|---|---|---|
| **$C_1$:** *production funding* | | | | | |
| *Held-out:* | | | | | |
| $A_1$ "…funded by the production company or network…" | | | | | |
| $A_2$ "…HNG…funded by the production company…" | | | | | |
| $A_3$ "…reality TV show…funded by the production company…" | | | | | |
| **RD-medoid** | 215 | "…funded through the production budget…" | **+0.598** | 100% | −0.189 |
| Single | 151 | "…spinoff of the original Holmes on Homes series…" | +0.068 | 70% | −0.334 |
| **$C_2$:** *not recognized* | | | | | |
| *Held-out:* | | | | | |
| $A_1$ "…not a widely recognized or officially named project…" | | | | | |
| $A_2$ "…not a well-known or officially recognized title…" | | | | | |
| $A_3$ "…no well-known entity…associated with a property or renovation project…" | | | | | |
| **RD-medoid** | 349 | "…not a widely recognized or officially documented show…" | **+0.239** | 100% | −0.228 |
| Single | 410 | "…not a well-known or officially recognized show…" | −0.011 | 40% | +0.258 |

*Table 3.* **Cluster-level transfer case study (AmbigQA, Qwen3-8B).** Steering from the $RD$ medoid vs. a random single continuation from the same cluster; held-out continuations shown for context. *idx* is the source continuation defining the steering direction (text in *Continuation*). $\Delta L_{+1}/\Delta L_{-1}$ are mean centered target-logit changes under amplification/ablation; Pos. is the fraction of held-out continuations with positive $\Delta L_{+1}$.

We further inspect steering validation on reasoning tasks by tracing the effects across reasoning steps, following Bogdan et al. (2025). Specifically, we examine the causal effect of previous reasoning steps on a fixed target reasoning step. We also analyze the dynamics of cluster merging and splitting from intermediate reasoning steps to the target step. Results are reported in Appendix E.3.

# 6. Conclusion

We propose an unsupervised feature discovery framework that analyzes a prompt's continuation distribution jointly in semantic and mechanistic views. We show that joint objective recovers interpretable continuation modes that semantic-only or mechanism-only baselines miss, and that cluster-derived feature sets yield consistent, directional steering effects, providing evidence that discovered modes correspond to actionable mechanistic factors. Together, these results suggest that distribution-level analysis can complement conventional circuit analysis by summarizing heterogeneity across continuations and producing mechanistic hypotheses for targeted follow-up investigation. Nevertheless, limitations on computational overhead and feature labeling remain, leaving it as future work. We provide a detailed discussion in Appendix D.

## Impact Statement

This paper contributes a method for auditing language-model continuations by grouping sampled outputs according to both semantic content and sequence-level mechanistic attribution. The intended positive impact is to help researchers inspect heterogeneous model behaviors without hand-selecting a single target completion, which may support more systematic safety and reliability analysis. However, the method can also be misused if cluster-level steering is applied to amplify undesirable behaviors, so we frame steering experiments as diagnostic interventions rather than deployment recommendations. In addition, validation under intervention does not guarantee the uniqueness of the identified features, which requires further counterfactual inspection. The approach relies on pretrained embedding models, including a transcoder and a sentence embedding model, as well as attribution estimates derived from first-order linear approximations over transcoder features. The approach depends on pretrained embedding models including a transcoder and a sentence embedding model, and attribution quality drawn by linear approximation on transcoder features; these dependencies should be considered when interpreting clusters in sensitive domains.

## Acknowledgement

Hyunjin is affiliated with the Department of Computer Science and Engineering, while Jaehyung and Youngji are affiliated with the Department of Artificial Intelligence at Yonsei University. This research was supported in part by Institute for Information & communications Technology Planning & Evaluation (IITP) grant funded by the Korea government (MSIT) (No. RS-2020-II201361, Artificial Intelligence Graduate School Program (Yonsei University); No. RS-2026-25522672, Development of Unified Reasoning Technology Mimicking Human Cognition for Hierarchical Understanding and Unbounded Problem Solving).

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

# A. Detailed Derivations

## A.1. E-step Derivation

The assignment rule Equation (10) follows from minimizing the R-D objective with respect to assignments while holding centers fixed.

*Derivation of E-step.* The contribution of sample $n$ to $\mathcal{L}_{\text{RD}}$ depends on its assignment $c(n) = k$:

$$\mathcal{L}_n = -\bar{P}_k \log \bar{P}_k + \beta_e \bar{P}_k \frac{\|e_n - \boldsymbol{\mu}_k^{(e)}\|_2^2}{W_k} P_n + \beta_a \bar{P}_k \frac{\|a_n - \boldsymbol{\mu}_k^{(a)}\|_1}{W_k} P_n \tag{20}$$

Since $\bar{P}_k = W_k/W_{\text{total}}$ and the entropy term is shared across all samples in cluster $k$, minimizing the per-sample cost yields:

$$c(n) = \underset{k}{\operatorname{argmin}} \left[ -\log \bar{P}_k + \beta_e \|e_n - \boldsymbol{\mu}_k^{(e)}\|_2^2 + \beta_a \|a_n - \boldsymbol{\mu}_k^{(a)}\|_1 \right] \tag{21}$$

The rate term $-\log \bar{P}_k$ favors assignment to larger clusters, acting as a regularizer against excessive fragmentation. $\square$

## A.2. M-step Derivation

*Derivation of M-step.* For fixed assignments, the optimal centers minimize the weighted sum of distances.

**Semantic Centers (L2).** Taking the derivative with respect to $\boldsymbol{\mu}_k^{(e)}$:

$$\frac{\partial}{\partial \boldsymbol{\mu}_k^{(e)}} \sum_{n:c(n)=k} P_n \|e_n - \boldsymbol{\mu}_k^{(e)}\|_2^2 = -2 \sum_{n:c(n)=k} P_n(e_n - \boldsymbol{\mu}_k^{(e)}) = 0 \tag{22}$$

Solving yields the probability-weighted mean:

$$\boldsymbol{\mu}_k^{(e)} = \frac{\sum_{n:c(n)=k} P_n \cdot e_n}{\sum_{n:c(n)=k} P_n} = \frac{1}{W_k} \sum_{n:c(n)=k} P_n \cdot e_n \tag{23}$$

**Attribution Centers (L1).** For L1 distance, the optimal center is the probability-weighted coordinate-wise median (Sabo & Scitovski, 2008):

$$\boldsymbol{\mu}_k^{(a)}[f] = \text{weighted-median}\left(\{a_n[f]\}_{n:c(n)=k}, \{P_n\}_{n:c(n)=k}\right) \tag{24}$$

where the weighted median is defined as the value $m$ satisfying $\sum_{n:a_n[f]<m} P_n \leq W_k/2$ and $\sum_{n:a_n[f]>m} P_n \leq W_k/2$. $\square$

## A.3. Split Criterion Derivation

*Derivation of Split Criterion.* Consider splitting cluster $k$ with mass $W_k$ into $(k_1, k_2)$ with masses $(W_1, W_2)$ where $W_1 = \alpha W_k$ and $W_2 = (1-\alpha)W_k$.

**Rate Change.** The entropy before split: $H_{\text{before}} = -\bar{P}_k \log \bar{P}_k + \text{other terms}$

After split with $\bar{P}_1 = \alpha \bar{P}_k$ and $\bar{P}_2 = (1-\alpha)\bar{P}_k$:

$$H_{\text{after}} = -\alpha \bar{P}_k \log(\alpha \bar{P}_k) - (1-\alpha)\bar{P}_k \log((1-\alpha)\bar{P}_k) + \text{other terms} \tag{25}$$

$$= -\bar{P}_k[\alpha \log \alpha + (1-\alpha)\log(1-\alpha)] - \bar{P}_k \log \bar{P}_k + \text{other terms} \tag{26}$$

Thus $\Delta H = H_{\text{after}} - H_{\text{before}} = \bar{P}_k H_{\text{bin}}(\alpha)$ (rate increases).

**Distortion Change.** Let $D_k^{(e)} = \bar{P}_k \text{Var}_k^{(e,w)}$ be the contribution to total distortion. After split:

$$D_{\text{after}}^{(e)} = \alpha \bar{P}_k \text{Var}_1^{(e,w)} + (1-\alpha)\bar{P}_k \text{Var}_2^{(e,w)} \tag{27}$$

The distortion reduction is:

$$\Delta D_k^{(e)} = D_k^{(e)} - D_{\text{after}}^{(e)} = \bar{P}_k \left[ \text{Var}_k^{(e,w)} - \alpha \text{Var}_1^{(e,w)} - (1-\alpha)\text{Var}_2^{(e,w)} \right] \tag{28}$$

**Split Criterion.** Split is beneficial iff $\Delta\mathcal{L}_{\mathrm{RD}} < 0$:

$$\bar{P}_k H_{\mathrm{bin}}(\alpha) < \beta_e \Delta D_k^{(e)} + \beta_a \Delta D_k^{(a)} \tag{29}$$

$\square$

### A.4. Junk Criterion Derivation

*Derivation of Junk Criterion.* Consider removing cluster $k$ and reassigning its points to remaining clusters using the R-D assignment rule.

**Rate Change.** Removing $k$ reduces entropy (fewer clusters), so $\Delta H = H_{\mathrm{before}} - H_{\mathrm{after}} > 0$ is the rate savings.

**Distortion Change.** Reassigning points from $k$ to other clusters increases distortion: $\Delta D = D_{\mathrm{after}} - D_{\mathrm{before}} > 0$. The distortion increase is computed as the probability-weighted sum of the difference between distances to new centers versus old center.

**Junk Criterion.** Junk is beneficial iff $\Delta\mathcal{L}_{\mathrm{RD}} < 0$:

$$-\Delta H + \beta_e \Delta D^{(e)} + \beta_a \Delta D^{(a)} < 0 \tag{30}$$

Rearranging: $\Delta H > \beta_e \Delta D^{(e)} + \beta_a \Delta D^{(a)}$ $\square$

### A.5. Convergence Proof

*Proof of Proposition 3.3.* Let $L^{(t,0)}$ denote the value of $\mathcal{L}_{\mathrm{RD}}$ at the beginning of outer iteration $t$, and let $L^{(t,1)}$, $L^{(t,2)}$, and $L^{(t,3)}$ denote the values after the E-step, M-step, and adaptive operations (Split and Junk), respectively. By optimality of the assignment update Equation (10),

$$L^{(t,1)} \leq L^{(t,0)}. \tag{31}$$

By optimality of the center update Equation (11),

$$L^{(t,2)} \leq L^{(t,1)}. \tag{32}$$

Finally, Split and Junk are applied only when they strictly decrease the objective by Propositions 3.1 and 3.2, so

$$L^{(t,3)} \leq L^{(t,2)}. \tag{33}$$

Since $L^{(t+1,0)} = L^{(t,3)}$, we obtain the chain

$$L^{(t,0)} \geq L^{(t,1)} \geq L^{(t,2)} \geq L^{(t,3)} = L^{(t+1,0)}. \tag{34}$$

Therefore, writing $L^{(t)} := L^{(t,0)}$, the outer-loop objective values satisfy

$$L^{(t+1)} \leq L^{(t)} \qquad \forall t \geq 0, \tag{35}$$

so $\{L^{(t)}\}_{t \geq 0}$ is monotonically non-increasing.

Moreover,

$$L^{(t)} = H(C^{(t)}) + \beta_e D_t^{(e)} + \beta_a D_t^{(a)} \geq 0, \tag{36}$$

where $D_t^{(e)}$ and $D_t^{(a)}$ denote the semantic and attribution distortions at iteration $t$. Hence $\{L^{(t)}\}$ is bounded below by 0. By the monotone convergence theorem for real sequences, there exists a finite value $L^\star \geq 0$ such that

$$L^{(t)} \downarrow L^\star \qquad \text{as } t \to \infty. \tag{37}$$

If we define the per-iteration decrease

$$\Delta_t := L^{(t)} - L^{(t+1)} \geq 0, \tag{38}$$

then

$$\Delta_t \to L^\star - L^\star = 0. \tag{39}$$

Thus the objective decreases monotonically, converges to a finite limit. $\square$

This bounded monotone-descent argument is standard in iterative clustering and related EM-style learning algorithms (Banerjee et al., 2005; Dhillon et al., 2003). Regarding convergence to the global minimum, the presence of hard cluster assignments and discrete Split/Junk decisions makes the overall optimization problem non-convex. Consequently, the algorithm is not guaranteed to reach the global minimum; in general, it converges only to a local optimum or a fixed point of the update procedure.

## B. Implementation Details

We provide the implementation details for main experiments (§ 4 and § 5).

### B.1. Continuation Sampling

**Sampling Strategy.** We sample continuations using nucleus (top-$p$) sampling with temperature scaling. For each prefix, we draw multiple batches of continuations until reaching a target count of distinct continuations or a maximum number of batches:

- Temperature: $T = 1.0$
- Nucleus probability: $p = 0.9$
- Maximum tokens per continuation: 20
- Batch size: 32 continuations per batch
- Maximum batches: 200
- Target distinct continuations: 500

**Instruction Template.** **Qwen3** models support switching reasoning mode, we use non-thinking mode for AmbigQA. For all experiments, we use raw questions as prefixes from each dataset without additional instruction.

**Deduplication.** Continuations are deduplicated by their text representation. For each unique text, we keep the sample with the highest per-token log probability to avoid bias toward shorter sequences:

$$\text{score}(y) = \frac{1}{|y|} \sum_{t=1}^{|y|} \log p(y_t \mid y_{<t}, x) \tag{40}$$

**Temperature Rescoring.** When sampling with temperature $T \neq 1$, we rescore all continuations at $T = 1$ using the model's logprobs interface. This ensures path probabilities reflect the true model distribution rather than the sampling distribution.

### B.2. Attribution Computation

We compute attribution from prefix features to continuation tokens using gradient-based methods with transcoders. We adapt the code from the `circuit-tracer` repository [3].

**Implementation Parameters.**

- Maximum feature nodes: 4096
- Batch size: 128 (continuation tokens per backward pass)
- Model: Qwen/Qwen3-8B or Qwen/Qwen3-4B
- Transcoder: mwhanna/qwen3-8b-transcoders or mwhanna/qwen3-4b-transcoders
- Layers: All MLP layers
- Data type: bfloat16

### B.3. Semantic Embedding Computation

We compute *contextual continuation embeddings* using a pretrained embedding model.

**Method.** For each continuation, we:

1. Concatenate prefix and continuation: full = prefix ⊕ continuation

---

[3]https://github.com/decoderesearch/circuit-tracer

2. Run a forward pass through the embedding model
3. Extract hidden states for *only* continuation tokens (which attend to the prefix through self-attention)
4. Mean pool the continuation hidden states
5. L2-normalize the resulting embedding

This captures the semantic meaning of the continuation in the context of the prefix, rather than treating them independently.

### B.4. Normalization

We normalize both semantic and attribution embeddings before clustering to ensure comparable scales across continuations.

**Semantic Embedding: Spherical Normalization.** Semantic embeddings are L2-normalized to lie on the unit sphere:

$$e_n \leftarrow \frac{e_n}{\|e_n\|_2} \tag{41}$$

This is standard practice for sentence embeddings and ensures that cosine similarity equals the dot product. Since our semantic distortion uses squared L2 distance, this normalization means $\|e_n - e_m\|_2^2 = 2(1 - e_n^\top e_m)$, directly relating distance to cosine similarity.

**Attribution Embedding: RMS Normalization.** Attribution vectors are normalized by their root-mean-square (RMS):

$$a_n \leftarrow \frac{a_n}{\text{RMS}(a_n)}, \quad \text{where} \quad \text{RMS}(a) = \sqrt{\frac{1}{d_a} \sum_{f=1}^{d_a} a[f]^2} \tag{42}$$

We choose RMS normalization over L2 normalization for attribution vectors because:

1. **Sparsity preservation:** RMS normalization preserves the sparsity pattern and relative magnitudes within each vector, only standardizing the overall scale.
2. **Robustness to dimensionality:** RMS is independent of dimensionality $d_a$, whereas L2 norm grows with $\sqrt{d_a}$ for vectors with similar per-coordinate magnitudes.
3. **Interpretability:** After RMS normalization, attribution values can still be interpreted as relative contributions—a feature with value 2.0 contributes twice as much as one with value 1.0.

**When Normalization is Applied.** Normalization is applied once after computing embeddings and attributions, before the clustering algorithm runs. Cluster centers inherit the normalization: $\boldsymbol{\mu}_k^{(e)}$ lies approximately on the unit sphere, and $\boldsymbol{\mu}_k^{(a)}$ has approximately unit RMS.

### B.5. PCA-based Split Initialization

The Split operation first constructs a candidate two-way partition of cluster $k$ using a PCA-based split, refines the candidate with an alternating assignment-and-center update, and accepts it only if the split criterion in Equation (12) is satisfied; otherwise, the original cluster is kept unchanged. To initialize the candidate split, we compute the leading principal direction in a joint semantic/attribution representation of the current cluster and split the cluster by a hyperplane orthogonal to that direction (Ding & He, 2004; Boley, 1998).

This procedure provides a structured alternative to random initialization. In the two-cluster squared-Euclidean case, the leading principal component is the continuous relaxation of the binary K-means cluster-indicator vector (Ding & He, 2004). Therefore, thresholding the projected scores gives an initialization aligned with the two-means objective. While, in our mixed $\ell_2^2/\ell_1$ objective, this exact relaxation no longer holds because the attribution term uses an $\ell_1$ dissimilarity. We therefore use the PCA split as a heuristic initialization and rely on the subsequent refinement step and split-acceptance criterion to ensure that accepted splits improve the target objective, as well as considering rate increase.

Let

$$\mathcal{I}_k = \{n : c(n) = k\}, \qquad W_k = \sum_{n \in \mathcal{I}_k} P_n, \qquad \tilde{P}_n = \frac{P_n}{W_k}.$$

For each sample in cluster $k$, define the centered semantic and attribution representations

$$\bar{e}_n = e_n - \boldsymbol{\mu}_k^{(e)}, \qquad \bar{a}_n = a_n - \boldsymbol{\mu}_k^{(a)}.$$

We perform PCA in the $\beta$-scaled joint representation

$$z_n = \begin{bmatrix} \sqrt{\beta_e}\, \bar{e}_n \\ \sqrt{\beta_a}\, \bar{a}_n \end{bmatrix}.$$

Equivalently, we seek the leading eigenvector of the weighted covariance matrix

$$\Sigma_k = \sum_{n \in \mathcal{I}_k} \tilde{P}_n z_n z_n^\top.$$

The direction is estimated by power iteration as follows. Initialize a random direction

$$v = \begin{bmatrix} v_e \\ v_a \end{bmatrix}, \qquad \|v\|_2 = \sqrt{\|v_e\|_2^2 + \|v_a\|_2^2} = 1.$$

Then, for up to 20 iterations, compute

$$\begin{aligned}
\mathrm{proj}_n &= z_n^\top v \\
&= \sqrt{\beta_e}\, \bar{e}_n^\top v_e + \sqrt{\beta_a}\, \bar{a}_n^\top v_a, & (43) \\
u_e &= \sqrt{\beta_e} \sum_{n \in \mathcal{I}_k} \tilde{P}_n \mathrm{proj}_n \bar{e}_n, & (44) \\
u_a &= \sqrt{\beta_a} \sum_{n \in \mathcal{I}_k} \tilde{P}_n \mathrm{proj}_n \bar{a}_n, & (45) \\
(v_e, v_a) &\leftarrow \frac{(u_e, u_a)}{\sqrt{\|u_e\|_2^2 + \|u_a\|_2^2}}. & (46)
\end{aligned}$$

We stop early if the direction stabilizes.

After the final direction is obtained, we bisect the cluster at the weighted median of the projected scores. Sort the samples in $\mathcal{I}_k$ by their final projection values:

$$\mathrm{proj}_{(1)} \leq \mathrm{proj}_{(2)} \leq \cdots \leq \mathrm{proj}_{(m)},$$

where $m = |\mathcal{I}_k|$. Define the cumulative probability mass

$$S_j = \sum_{i=1}^{j} P_{(i)}.$$

We choose

$$j^\star = \min\left\{ j : S_j \geq \frac{W_k}{2} \right\}.$$

The initial child assignment is then defined by rank:

$$\ell_{(i)} = \mathbf{1}\{i > j^\star\}.$$

Equivalently, when there are no ties at the weighted median $\tau = \mathrm{proj}_{(j^\star)}$, this corresponds to

$$\ell_n = \mathbf{1}\{\mathrm{proj}_n > \tau\}.$$

This produces an approximately balanced split in probability mass.

We then refine this initialization with 5 iterations of a Lloyd-style two-cluster update under the combined dissimilarity

$$d(n, r) = \beta_e \left\| e_n - \boldsymbol{\mu}_r^{(e)} \right\|_2^2 + \beta_a \left\| a_n - \boldsymbol{\mu}_r^{(a)} \right\|_1, \qquad r \in \{1, 2\}.$$

At each iteration, assignments are updated by

$$\ell_n \leftarrow \operatorname*{argmin}_{r \in \{1,2\}} d(n, r),$$

semantic child centers are updated by probability-weighted means,

$$\boldsymbol{\mu}_r^{(e)} = \frac{\sum_{n \in \mathcal{I}_k : \ell_n = r} P_n e_n}{\sum_{n \in \mathcal{I}_k : \ell_n = r} P_n},$$

and attribution child centers are updated by coordinate-wise probability-weighted medians,

$$\boldsymbol{\mu}_r^{(a)} = \mathrm{wmedian}\left(\{a_n : n \in \mathcal{I}_k, \ \ell_n = r\}, \{P_n : n \in \mathcal{I}_k, \ \ell_n = r\}\right).$$

The $\sqrt{\beta}$-scaled representation makes variance along the split direction reflect the relative weighting of the two views in the squared-Euclidean part of the objective, while the final mixed-distance refinement adapts the candidate split to the actual objective used by the Split operation.

### B.6. Convergence Criterion

The algorithm terminates when the relative change in $\mathcal{L}_{\mathrm{RD}}$ falls below threshold:

$$\frac{|\mathcal{L}_{\mathrm{RD}}^{(t)} - \mathcal{L}_{\mathrm{RD}}^{(t-1)}|}{|\mathcal{L}_{\mathrm{RD}}^{(t-1)}| + \epsilon_0} < \epsilon \tag{47}$$

with default $\epsilon = 10^{-3}$ and $\epsilon_0 = 10^{-10}$ for numerical stability when $\mathcal{L}_{\mathrm{RD}}^{(t-1)} \approx 0$.

## C. Continuation-Level Attribution Analysis.

Many attribution studies (Wang et al., 2023; Conmy et al., 2023) focus on explaining the *next-token* logit, largely because this setting aligns naturally with feature-to-feature connectivity graphs. In this work, we instead represent each sampled continuation using **direct prefix-feature-to-continuation** attributions. While this representation discards multi-step mediation and higher-order interactions, we hypothesize that it preserves the dominant continuation-conditional signal needed to differentiate computational patterns across continuations.

### C.1. What causal effect does pooled attribution represent?

The aggregated attribution can be interpreted as capturing the average sequence-level effect of position-wise attribution patterns, rather than implying a uniform attribution trend across all continuation positions. This distinction matches our target quantity: unlike conventional next-token analyses, we measure intervention effects on the mean demeaned continuation logit.

Let $j \equiv (i, l, f)$ index a prefix transcoder feature, and define the token-wise demeaned logit as

$$\tilde{\ell}_{y_t} := \ell_{y_t} - \bar{\ell}, \qquad \bar{\ell} := \frac{1}{|V|} \sum_{v \in V} \ell_v. \tag{48}$$

Using the notation $do(h_j = h')$ for an intervention (Pearl, 2009) that forces feature $h_j$ to the value $h'$, the token-level intervention effect (Syed et al., 2024) between two feature values $h_j^+$ and $h_j^-$ is

$$\tau_{j,t}(h_j^+, h_j^-) := \tilde{\ell}_{y_t}\left(x_{\mathrm{clean}} \mid do(h_j = h_j^+)\right) - \tilde{\ell}_{y_t}\left(x_{\mathrm{clean}} \mid do(h_j = h_j^-)\right). \tag{49}$$

For a continuation of length $T$, define the mean demeaned continuation logit as

$$\tilde{\ell}_{1:T}^{\mathrm{mean}} := \frac{1}{T} \sum_{t=1}^{T} \tilde{\ell}_{y_t}. \tag{50}$$

The corresponding sequence-level intervention effect is

$$\tau_j^{\mathrm{mean}}(h_j^+, h_j^-) := \tilde{\ell}_{1:T}^{\mathrm{mean}}\left(x_{\mathrm{clean}} \mid do(h_j = h_j^+)\right) - \tilde{\ell}_{1:T}^{\mathrm{mean}}\left(x_{\mathrm{clean}} \mid do(h_j = h_j^-)\right). \tag{51}$$

By linearity of the mean,

$$\tau_j^{\text{mean}}(h_j^+, h_j^-) = \left( \frac{1}{T} \sum_{t=1}^{T} \tilde{\ell}_{y_t} \left( x_{\text{clean}} \mid do(h_j = h_j^+) \right) \right) - \left( \frac{1}{T} \sum_{t=1}^{T} \tilde{\ell}_{y_t} \left( x_{\text{clean}} \mid do(h_j = h_j^-) \right) \right)$$

$$= \frac{1}{T} \sum_{t=1}^{T} \tau_{j,t}(h_j^+, h_j^-). \tag{52}$$

Thus, the sequence-level effect is exactly the average of the token-level effects. The pooled attribution therefore summarizes average generated-token causal effects, even when token-wise effects are heterogeneous across positions.

In our causal validation, this contrast is approximated by the signed steering intervention in Eq. 16. Positive steering $\epsilon > 0$ implements an amplification intervention $do(h_j = h_j^+)$, while negative steering $\epsilon < 0$ implements a suppression intervention $do(h_j = h_j^-)$. The empirical test therefore asks whether continuous feature scaling produces the continuation-level logit shifts predicted by the underlying causal direction. In practice, we steer the top $|S| \in \{5, 10\}$ features jointly (§ 5), so the measured effect is a feature-set intervention effect rather than an isolated single-feature effect.

### C.2. Token-level Attribution Analysis

We complement the pooled analysis with token-level diagnostics that examine how attribution structure changes across generated positions. These diagnostics use AmbigQA continuations with direct prefix-to-token attributions computed following the procedure in Section 4. For the Qwen3-4B scaled diagnostic reported below, the token-similarity analysis uses 100 sampled continuations from 150 Stage-3 context files. The model-dependent source-mass and span-steering analyses use five sampled continuations from 40 context files; for dynamic source-mass attribution, we cap the recomputed attribution graph at 512 feature nodes and evaluate up to 32 target positions.

**Attribution magnitude across generated positions.** For each continuation, we compute the $\ell_1$ norm of the attribution vector at every generated token position before pooling. As shown in Figure 7, attribution magnitude is largest at the first one or two generated tokens and then decays smoothly with position. The decay is gradual rather than abrupt: by position 31, the mean norm remains more than half of its peak value. Thus, the original prefix continues to supply measurable attribution mass throughout generation, which supports pooling over the full generated span when constructing continuation-level representations.

**Token-wise attribution similarity.** We next ask whether nearby generated tokens rely on similar attribution patterns. For each pair of generated positions, we compare their local attribution vectors using top-100 Jaccard overlap and $\ell_1$ distance, and aggregate the result by positional gap. As expected, attribution vectors are most similar for nearby targets: Jaccard overlap is high and $\ell_1$ distance is low at small gaps. This local advantage over a random continuation from the same prefix weakens as the comparison moves from nearby tokens to earlier–later token pairs, disappearing after roughly gap 10 under top-100 Jaccard and after roughly gap 13 under $\ell_1$ distance. The lower-triangular pairwise Jaccard heatmap in Figure 9 shows the same local structure before gap averaging. This indicates that local attribution patterns are coherent over short ranges but become substantially reconfigured over longer generated spans.

**Original-prefix versus generated-history source mass.** To separate static prefix effects from autoregressive effects, we recompute attribution for each target token using the original prefix together with the preceding generated tokens as the source context. We then partition source-node $\ell_1$ mass according to whether each source node belongs to the original prefix or to prior generated tokens. The original prefix dominates early and middle target positions, while generated-history mass grows steadily over the continuation. In the scaled Qwen3-4B run, generated-history mass becomes comparable to original-prefix mass around position 25 and exceeds it by position 30, where the split is approximately 45:55 between original-prefix and generated-history mass. This transition is consistent with a generation process in which early tokens remain strongly conditioned on the prompt, while later tokens increasingly depend on the model's generated history.

**Causal validation of pooled features.** Finally, we test whether the pooled attribution vectors identify causally relevant features. For each pooled vector, we select the top-10 features by absolute attribution magnitude and compare them with 10 randomly sampled features. We then steer these features using the intervention procedure in Eq. 16. The attribution-selected features produce clearer token-level logit changes than the random baseline: negative steering reliably decreases demeaned logits, while positive steering generally increases them. Comparing attribution spans on the same Qwen3-4B subset shows the same signed trend for full-span, first-token, last-token, and first-five-token vectors. The full-span vector gives the largest average shift, from $-0.093$ at $\epsilon = -1$ to $0.232$ at $\epsilon = 1$, while single-token and first-five-token vectors show smaller but

directionally consistent effects. These results provide causal evidence that pooled continuation-span attributions preserve meaningful information about the generated sequence.

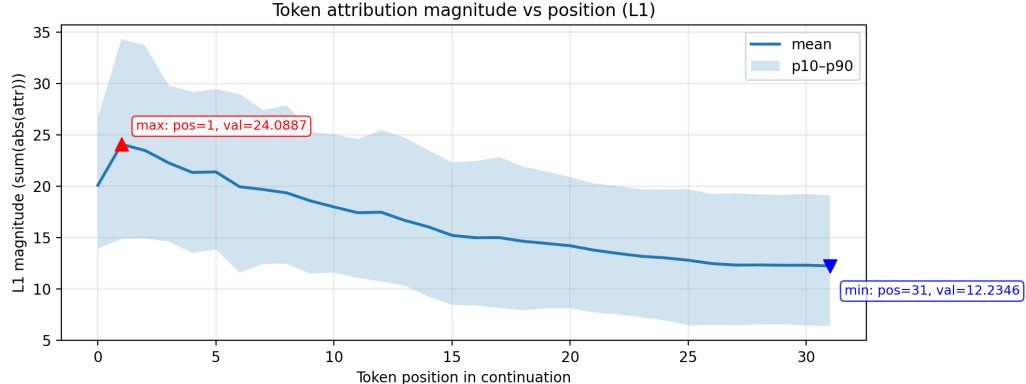

*Figure 7.* Mean $\ell_1$ norm of the attribution vector at each generated-token position.

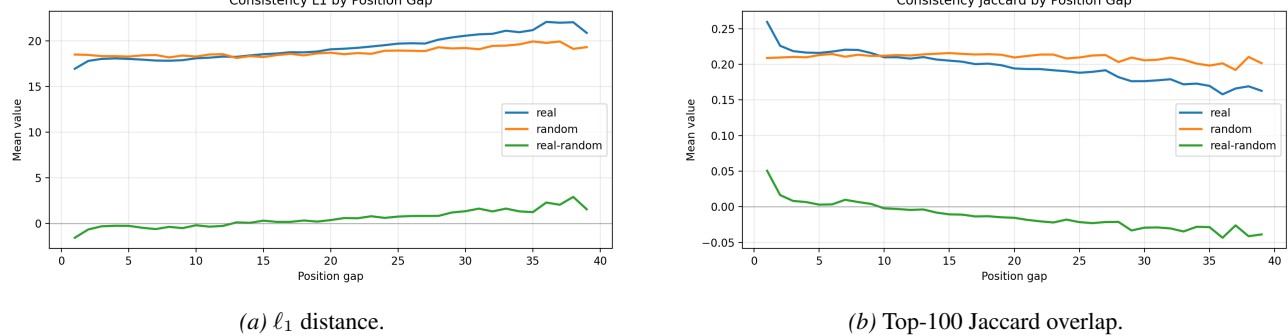

*(a)* $\ell_1$ distance.

*(b)* Top-100 Jaccard overlap.

*Figure 8.* Token-wise attribution similarity as a function of positional gap. Real pairs are compared against random continuations from the same prefix; lower $\ell_1$ and higher Jaccard indicate greater similarity.

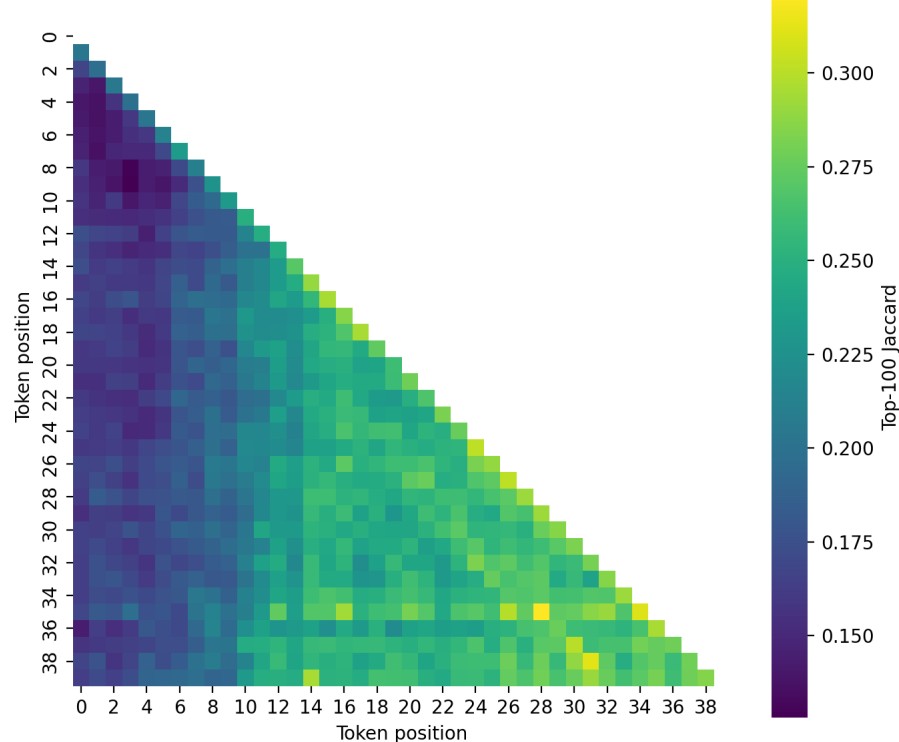

*Figure 9.* Lower-triangular heatmap of top-100 Jaccard overlap between token-level attribution vectors, averaged across the Qwen3-4B consistency subset.

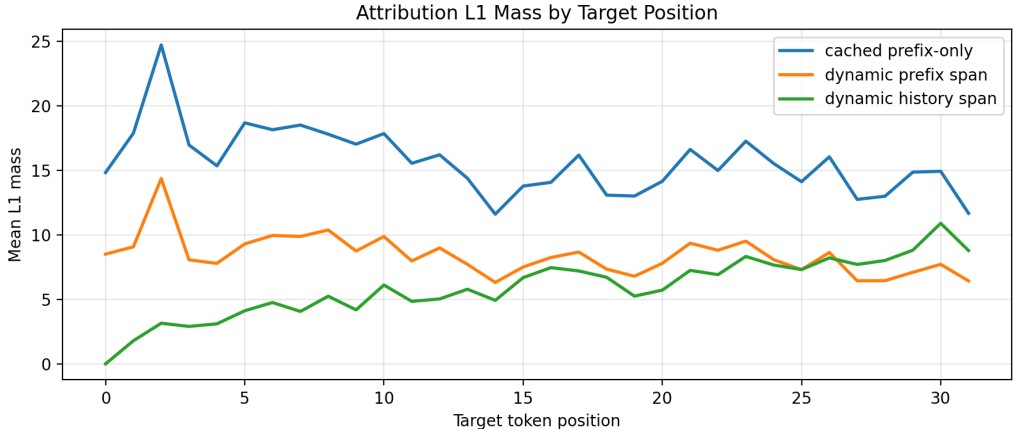

*Figure 10.* Dynamic source-mass decomposition by target position. The cached prefix-only curve uses the saved full-continuation attribution, while the dynamic prefix and history curves recompute attribution for each target token using the original prefix plus preceding generated tokens as source context.

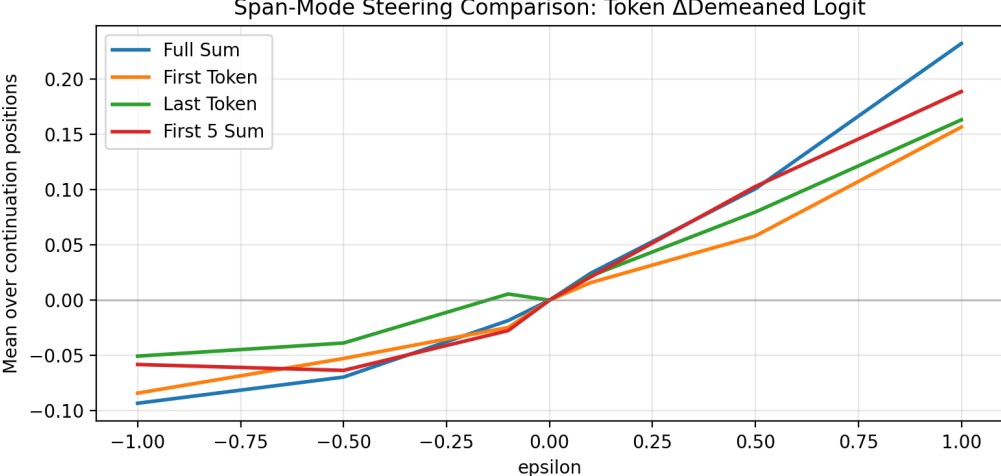

*Figure 11.* Span-mode steering comparison on the Qwen3-4B scaled subset. The full continuation span gives the strongest average signed response, while single-token spans are noisier but remain directionally aligned.

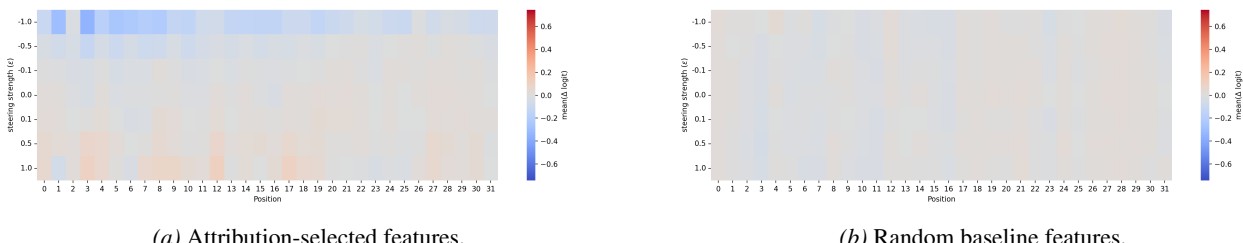

*(a)* Attribution-selected features.  *(b)* Random baseline features.

*Figure 12.* Average token-level steering effect, measured as demeaned target-logit change across continuation positions. Attribution-selected features produce a more coherent signed response than matched random feature sets.

## D. Limitations and Future Work.

We introduce an unsupervised feature-discovery framework that clusters *continuations* while jointly controlling semantic variance and mechanistic variance. Despite the flexibility and empirical causal effectiveness of the approach, several limitations remain.

- **Computational overhead.** Our pipeline requires computing semantic embeddings and mechanistic representations for each sampled continuation. In particular, attribution-based mechanisms typically require a backward pass (and, depending on the attribution choice, additional forward passes), which becomes expensive for long continuations and long-context prefixes. This cost can be a bottleneck in regimes such as long-context QA, agentic interaction traces, or multi-step reasoning where both the prefix and continuation lengths grow. That said, designing attribution methods that are simultaneously *efficient* and *faithful* remains an open challenge in mechanistic interpretability, and our framework is intentionally *representation-agnostic*: it can incorporate alternative mechanism summaries (e.g., cheaper proxies, sparse feature readouts, cached gradients, or layer-/token-subsampled attributions) without changing the clustering objective. We view scaling to long-context settings as a concrete direction for future work, where more efficient mechanism representations could make continuation-level unsupervised discovery practical in agent and reasoning-model deployments.

- **Feature labeling and interpretability.** While our method discovers coherent groups of continuations and associated mechanistic signatures, we currently lack a principled evaluation framework for assigning *human-interpretable* labels to clusters. In practice, a single cluster can entangle multiple axes of variation (e.g., answer identity, formatting/framing style, hedging, citation patterns), and the resulting cluster centers in mechanism space are high-dimensional and not immediately legible. Future work could integrate automatic labeling (Paulo et al., 2025) via (i) exemplar-based

summaries (e.g., cluster medoids used in our steering pipeline), (ii) feature-level interpretation tools (SAE or TC feature descriptions and activation-based retrieval) (Lindsey et al., 2025), and (iii) lightweight natural-language explanations that are explicitly validated by causal tests (e.g., targeted feature ablations or steering). Developing robust labeling protocols is especially important if clusters are treated as scientific hypotheses about circuits rather than merely descriptive groupings.

- **Token-level dynamics.** As discussed in Appendix C, our analysis does not attempt to trace the complete interventional effect to a target logit. Rather, we estimate effects from the *prefix* to the continuation as a whole, instead of attributing influence from all preceding tokens to a single next-token prediction. A full tracing analysis of token-level dynamics remains an important direction for future work.

## E. Additional Results.

### E.1. Detailed Jaccard Similarity Figure

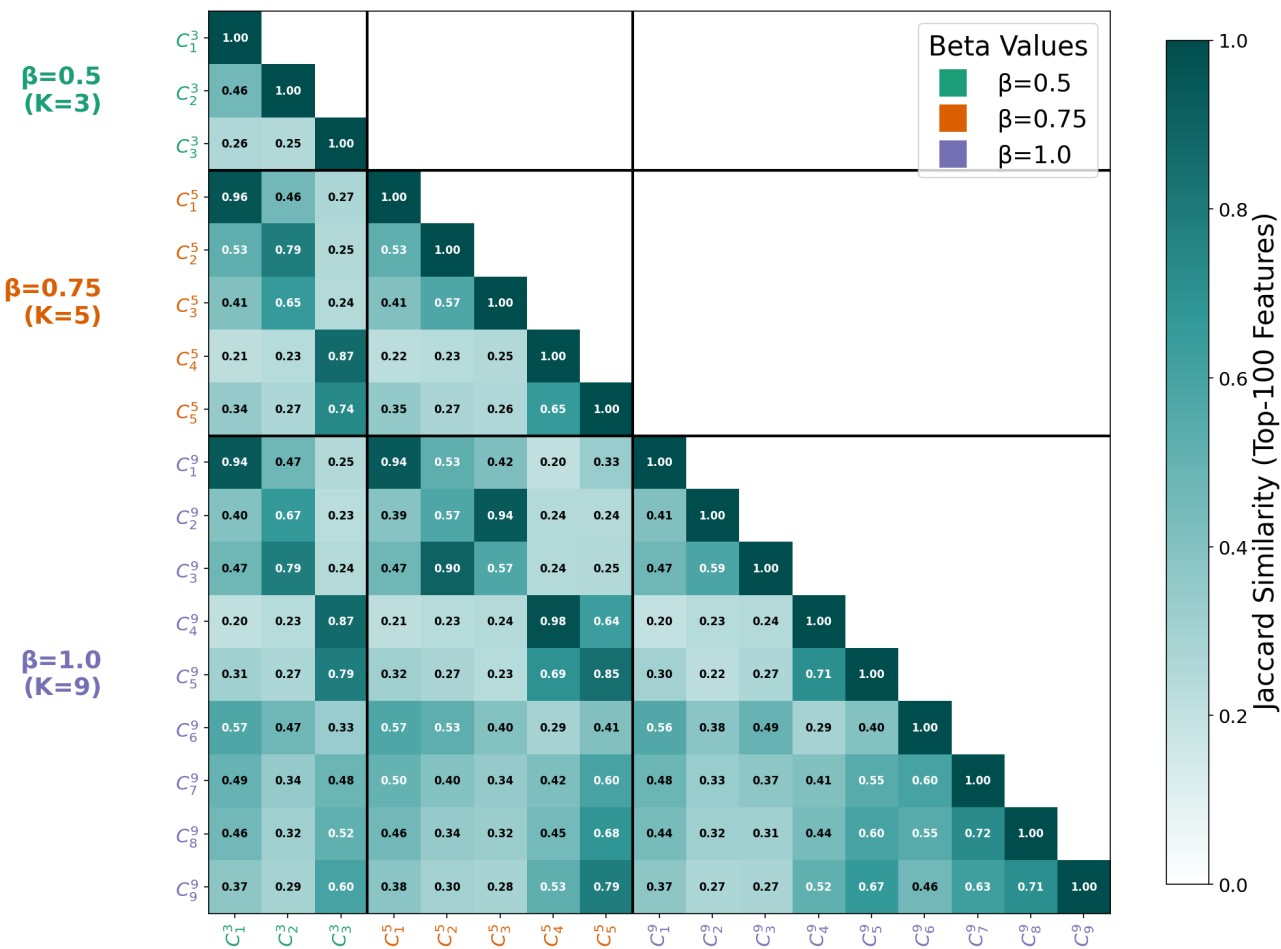

*Figure 13.* Figure 3 with Jaccard similiarity scores labeled.

### E.2. Extended Experiments

**Setup.** We include additional MMLU and Gemma3 extensions. The MMLU experiment evaluates Qwen3-4B and Qwen3-8B on a 209-question validation subset spanning international law, logical fallacies, moral disputes, philosophy, professional psychology, sociology, and world religions. The Gemma3 experiment evaluates AmbigQA with Gemma3-4B-it. For each raw question or prompt, we sample continuations and apply the same binary prefix-continuation analysis as the main AmbigQA setting: each continuation receives a semantic embedding and a prefix-to-continuation attribution vector, and

| Model | Method | $\gamma$ | $\beta = 0.50$ | $\beta = 0.75$ | $\beta = 1.25$ | $\beta = 1.50$ |
|---|---|---|---|---|---|---|
| | RD | 0.30 | **7.78** | **6.10** | **5.12** | **5.07** |
| | RD | 0.50 | **6.76** | **5.37** | **4.02** | **3.76** |
| | RD | 0.70 | **5.00** | **4.25** | **3.17** | **2.87** |
| Qwen3-4B | KM-S | 0.30 | 29.00 | 25.58 | 23.75 | 23.66 |
| | KM-S | 0.50 | 22.77 | 20.21 | 17.46 | 17.10 |
| | KM-S | 0.70 | 14.62 | 13.91 | 11.98 | 11.36 |
| | KM-A | 0.30 | 23.17 | 19.03 | 17.14 | 17.10 |
| | KM-A | 0.50 | 19.43 | 15.98 | 12.77 | 12.40 |
| | KM-A | 0.70 | 13.06 | 12.00 | 9.45 | 8.64 |
| | RD | 0.30 | **7.36** | **5.85** | **4.75** | **4.61** |
| | RD | 0.50 | **6.21** | **5.11** | **3.84** | **3.57** |
| | RD | 0.70 | **4.43** | **3.94** | **3.11** | **2.74** |
| Qwen3-8B | KM-S | 0.30 | 28.06 | 25.19 | 23.16 | 22.98 |
| | KM-S | 0.50 | 21.55 | 19.59 | 17.35 | 16.77 |
| | KM-S | 0.70 | 13.63 | 13.15 | 11.80 | 11.26 |
| | KM-A | 0.30 | 23.83 | 20.10 | 17.59 | 17.40 |
| | KM-A | 0.50 | 19.12 | 16.58 | 13.58 | 12.92 |
| | KM-A | 0.70 | 12.52 | 11.75 | 9.96 | 9.18 |

*Table 4.* **MMLU clustering results.** Rows report mixed distortion $D_\gamma$ on the 209-question MMLU validation subset; lower is better. KM-S denotes K-means on semantic embeddings, and KM-A denotes K-means on attribution vectors.

clustering is performed on the resulting two-view representation. This differs from Appendix E.3, where each cell is a pair of reasoning steps $(i, j)$; here each cell is a standard question-level continuation distribution.

**Results.** Table 4 reports the MMLU mixed distortion $D_\gamma$. Across both model sizes, RD gives the lowest $D_\gamma$ in every listed $(\beta, \gamma)$ configuration, with especially large gaps over semantic-only K-means. The result supports the main clustering claim outside AmbigQA: jointly optimizing semantic and mechanistic distortion recovers lower-distortion partitions than matching the rate with a single-view baseline. Tables 5 and 6 add MMLU steering results for the same Qwen model family. Qwen3-4B shows clear monotonic steering for RD and Single directions, while KM-Sem correlations remain near zero. Qwen3-8B is substantially weaker: correlations are small across all three methods, so the 8B steering result is less decisive than the clustering result. Table 7 reports the Gemma3-4B-it AmbigQA clustering result. RD is consistently better than semantic-only K-means, while attribution-only K-means is competitive at larger $\gamma$; this makes the Gemma result a useful stress test rather than a uniformly favorable case.

| Method | Metric | $\beta$ 0.75 | | | 1.25 | | | 1.50 | | |
|---|---|---|---|---|---|---|---|---|---|---|
| | $\gamma$ | 0.3 | 0.5 | 0.7 | 0.3 | 0.5 | 0.7 | 0.3 | 0.5 | 0.7 |
| RD | $\rho_s$ | 0.476 | 0.358 | 0.323 | 0.477 | 0.524 | 0.313 | 0.489 | 0.474 | 0.359 |
| | $\rho$ | 0.376 | 0.274 | 0.243 | 0.388 | 0.417 | 0.240 | 0.386 | 0.373 | 0.280 |
| | $\epsilon_{-1.0}$ | -2.494 | -2.642 | -2.789 | -1.389 | -1.404 | -2.377 | -1.073 | -1.453 | -2.308 |
| | $\epsilon_{-0.5}$ | -2.161 | -2.250 | -2.448 | -1.458 | -1.543 | -2.208 | -1.288 | -1.639 | -2.134 |
| | $\epsilon_{-0.1}$ | -0.562 | -0.602 | -0.665 | -0.387 | -0.443 | -0.550 | -0.339 | -0.412 | -0.552 |
| | $\epsilon_{0.1}$ | 0.724 | 0.667 | 0.674 | 0.535 | 0.604 | 0.640 | 0.544 | 0.597 | 0.743 |
| | $\epsilon_{0.5}$ | 3.969 | 4.139 | 4.349 | 3.013 | 3.277 | 4.144 | 2.977 | 3.236 | 4.243 |
| | $\epsilon_{1.0}$ | 8.676 | 9.238 | 9.537 | 6.699 | 7.252 | 9.337 | 6.611 | 7.212 | 9.444 |
| KM-Sem | $\rho_s$ | 0.004 | -0.003 | -0.018 | 0.032 | -0.040 | -0.015 | 0.001 | -0.046 | 0.021 |
| | $\rho$ | -0.011 | -0.001 | -0.012 | 0.016 | -0.042 | -0.011 | -0.005 | -0.047 | 0.005 |
| | $\epsilon_{-1.0}$ | 1.257 | 1.570 | 1.761 | 0.745 | 0.972 | 1.836 | 0.731 | 0.886 | 1.501 |
| | $\epsilon_{-0.5}$ | 0.366 | 0.294 | 0.451 | 0.155 | 0.205 | 0.422 | 0.142 | 0.191 | 0.357 |
| | $\epsilon_{-0.1}$ | 0.111 | -0.043 | -0.011 | 0.104 | 0.059 | 0.008 | 0.107 | 0.127 | 0.040 |
| | $\epsilon_{0.1}$ | 0.006 | -0.022 | -0.002 | 0.157 | 0.060 | -0.015 | 0.182 | 0.108 | 0.064 |
| | $\epsilon_{0.5}$ | 0.209 | 0.393 | 0.329 | 0.618 | 0.437 | 0.320 | 0.605 | 0.466 | 0.336 |
| | $\epsilon_{1.0}$ | 0.901 | 1.411 | 1.253 | 1.154 | 1.144 | 1.309 | 1.188 | 1.259 | 1.406 |
| Single | $\rho_s$ | 0.476 | 0.354 | 0.317 | 0.477 | 0.525 | 0.304 | 0.489 | 0.474 | 0.355 |
| | $\rho$ | 0.377 | 0.275 | 0.239 | 0.388 | 0.416 | 0.240 | 0.386 | 0.372 | 0.289 |
| | $\epsilon_{-1.0}$ | -2.499 | -2.652 | -2.845 | -1.308 | -1.444 | -2.391 | -1.102 | -1.506 | -2.267 |
| | $\epsilon_{-0.5}$ | -2.125 | -2.277 | -2.486 | -1.415 | -1.534 | -2.210 | -1.293 | -1.630 | -2.087 |
| | $\epsilon_{-0.1}$ | -0.560 | -0.603 | -0.697 | -0.377 | -0.436 | -0.548 | -0.342 | -0.401 | -0.561 |
| | $\epsilon_{0.1}$ | 0.732 | 0.648 | 0.681 | 0.527 | 0.621 | 0.609 | 0.552 | 0.612 | 0.711 |
| | $\epsilon_{0.5}$ | 3.933 | 4.138 | 4.314 | 2.983 | 3.236 | 4.095 | 2.973 | 3.192 | 4.177 |
| | $\epsilon_{1.0}$ | 8.689 | 9.235 | 9.476 | 6.709 | 7.236 | 9.293 | 6.613 | 7.181 | 9.379 |

*Table 5.* **MMLU Qwen3-4B steering results.** Rows report correlations between steering strength and cluster-demeaned target-logit change, followed by the mean logit change at each steering strength. **RD** uses the two-view cluster medoid direction; **KM-Sem** uses medoids from matched semantic-only K-means clusters; **Single** uses a single sampled attribution direction. Rows are ordered as RD, KM-Sem, and Single; columns are grouped by $\beta$ and $\gamma$.

| Method | Metric | $\beta$ 0.75 | | | 1.25 | | | 1.50 | | |
|---|---|---|---|---|---|---|---|---|---|---|
| | $\gamma$ | 0.3 | 0.5 | 0.7 | 0.3 | 0.5 | 0.7 | 0.3 | 0.5 | 0.7 |
| RD | $\rho_s$ | -0.041 | 0.058 | 0.051 | 0.060 | -0.015 | 0.014 | 0.082 | 0.070 | 0.017 |
| | $\rho$ | -0.030 | 0.042 | 0.020 | 0.054 | -0.007 | -0.006 | 0.065 | 0.055 | 0.002 |
| | $\epsilon_{-1.0}$ | 4.442 | 4.943 | 4.739 | 4.240 | 4.945 | 4.640 | 4.013 | 5.127 | 3.960 |
| | $\epsilon_{-0.5}$ | 1.766 | 1.382 | 0.992 | 1.472 | 1.463 | 1.224 | 1.627 | 1.592 | 1.516 |
| | $\epsilon_{-0.1}$ | 0.562 | 0.389 | 0.211 | 0.205 | 0.805 | 0.154 | 0.380 | 0.704 | 0.317 |
| | $\epsilon_{0.1}$ | -0.004 | 0.358 | 0.063 | -0.194 | 0.164 | 0.129 | 0.027 | -0.099 | 0.075 |
| | $\epsilon_{0.5}$ | 0.113 | 0.616 | -0.064 | -0.516 | -0.079 | 0.363 | -0.012 | 0.172 | 0.255 |
| | $\epsilon_{1.0}$ | 0.247 | 1.147 | 0.608 | -0.236 | -0.319 | 0.717 | 0.761 | 1.119 | 0.355 |
| KM-Sem | $\rho_s$ | 0.018 | -0.027 | -0.005 | 0.020 | 0.018 | -0.018 | 0.033 | -0.006 | -0.038 |
| | $\rho$ | 0.010 | -0.028 | -0.016 | 0.012 | 0.010 | -0.013 | 0.013 | 0.008 | -0.038 |
| | $\epsilon_{-1.0}$ | 2.056 | 3.282 | 1.947 | 1.738 | 2.438 | 3.050 | 1.844 | 3.552 | 3.555 |
| | $\epsilon_{-0.5}$ | 0.429 | 0.903 | 0.533 | 0.137 | 0.680 | 0.878 | 0.127 | 1.031 | 1.065 |
| | $\epsilon_{-0.1}$ | -0.016 | 0.259 | 0.181 | -0.233 | 0.111 | 0.135 | -0.096 | 0.116 | 0.401 |
| | $\epsilon_{0.1}$ | 0.521 | 0.539 | 0.203 | 0.102 | 0.863 | 0.324 | 0.479 | 0.367 | 0.518 |
| | $\epsilon_{0.5}$ | 1.588 | 1.154 | 0.739 | 1.437 | 2.021 | 0.775 | 1.717 | 1.703 | 1.008 |
| | $\epsilon_{1.0}$ | 3.798 | 3.777 | 2.828 | 2.792 | 3.834 | 3.270 | 3.464 | 4.329 | 2.570 |
| Single | $\rho_s$ | -0.006 | 0.067 | 0.051 | 0.079 | 0.045 | 0.047 | 0.103 | 0.085 | 0.037 |
| | $\rho$ | -0.018 | 0.060 | 0.036 | 0.072 | 0.046 | 0.043 | 0.091 | 0.078 | 0.022 |
| | $\epsilon_{-1.0}$ | 4.600 | 4.533 | 3.452 | 4.041 | 5.077 | 4.398 | 3.826 | 5.040 | 3.442 |
| | $\epsilon_{-0.5}$ | 1.658 | 1.228 | 0.917 | 1.437 | 1.659 | 1.330 | 1.626 | 1.847 | 1.287 |
| | $\epsilon_{-0.1}$ | 0.575 | 0.586 | 0.220 | 0.120 | 0.878 | 0.343 | 0.339 | 0.684 | 0.271 |
| | $\epsilon_{0.1}$ | -0.020 | 0.458 | 0.175 | -0.237 | 0.067 | 0.319 | -0.049 | -0.084 | 0.139 |
| | $\epsilon_{0.5}$ | 0.055 | 0.490 | 0.022 | -0.542 | -0.170 | 0.143 | -0.063 | 0.334 | 0.263 |
| | $\epsilon_{1.0}$ | 0.456 | 1.227 | 0.489 | -0.114 | 0.083 | 0.581 | 0.887 | 1.463 | 0.444 |

*Table 6.* **MMLU Qwen3-8B steering results.** Rows report correlations between steering strength and cluster-demeaned target-logit change, followed by the mean logit change at each steering strength. **RD** uses the two-view cluster medoid direction; **KM-Sem** uses medoids from matched semantic-only K-means clusters; **Single** uses a single sampled attribution direction. Rows are ordered as RD, KM-Sem, and Single; columns are grouped by $\beta$ and $\gamma$.

| $\gamma$ | $\beta$ | RD | KM-S | KM-A |
|------|------|------|------|------|
| 0.30 | 0.50 | **4.81** | 6.62 | 4.88 |
| 0.30 | 0.75 | **3.31** | 5.58 | 3.73 |
| 0.30 | 1.00 | **2.78** | 5.18 | 3.41 |
| 0.50 | 0.50 | 4.70 | 5.50 | **4.42** |
| 0.50 | 0.75 | **3.20** | 4.58 | 3.32 |
| 0.50 | 1.00 | **2.52** | 4.10 | 2.79 |
| 0.70 | 0.50 | 3.76 | 3.79 | **3.27** |
| 0.70 | 0.75 | 3.03 | 3.43 | **2.82** |
| 0.70 | 1.00 | 2.43 | 3.09 | **2.40** |

*Table 7.* **AmbigQA+Gemma3-4B-it clustering results.** Rows report mixed distortion $D_\gamma$; lower is better.

### E.3. Reasoning Task Results

**Setup.** Reasoning tasks differ from the binary prefix-continuation setting used in the main experiments because the unit of analysis is a pair of reasoning steps rather than a single fixed prefix. We evaluate Qwen3-0.6B on GSM8K (Cobbe et al., 2021) and MATH500 (Lightman et al., 2023) using the Qwen3-0.6B transcoders. For each example, we roll out up to 12 reasoning steps (delimited by "\n\n") and sample 64 candidate traces per step. The committed reasoning path used for later prefixes is deterministic: at each step, we select the candidate with the highest normalized sequence probability. For every target step $j$, we then construct explicit pairwise inputs for each previous source step $i < j$. The attribution and semantic embedding for a pair are computed from the fixed prefix containing the problem and committed earlier steps, while the continuation span is the target-step trace. Thus, unlike the other datasets where one prefix is compared to one continuation span, the reasoning analysis directly asks how a previous reasoning step $i$ changes the distribution over a later step $j$.

We use the same clustering objective as in the main experiments. The reported run uses combined-medoid clustering with $\beta = 0.75$ and $\gamma = 0.7$, selected from the sweep because it produced non-degenerate cluster counts in the target range. For steering validation, we run the steering run separately for each explicit pair $(i, j)$ using the full cluster direction $h_c$, top-$B = 5$ transcoder features, and steering strengths $\epsilon \in \{-1.0, -0.5, -0.1, 0, 0.1, 0.5, 1.0\}$. We report correlations between steering strength and the target cluster-centered logit change.

**Quantitative results.** Table 8 summarizes the steering validation across the two reasoning benchmarks. Across 2,898 step-pair prefix files, 2,445 valid pairwise steering evaluations, and 9,536 target-step clusters, the overall Spearman correlation is positive ($\rho_s = 0.13$). MATH500 is slightly stronger than GSM8K (0.15 versus 0.11), but both are positive, indicating that the discovered source-step feature directions tend to shift probability mass toward their associated target-step clusters.

The aggregate number is modest because the effect is highly step-dependent. Table 9 and Figure 14 report the same validation grouped by explicit source-target pairs $(i, j)$. The earliest source step has the largest and most persistent effect: for example, the pooled step-pair correlations are $\rho_s = 0.64$ for $(1, 2)$, $0.50$ for $(1, 3)$, and remain above $0.30$ through several later targets. In contrast, many adjacent or late-step pairs are close to zero or negative. This supports the use of explicit pairwise reasoning validation: treating "all previous steps" as one prefix would obscure which prior step is causally aligned with a target-step cluster.

*Table 9.* Reasoning centered-logit steering correlations grouped by explicit reasoning step pairs $(i, j)$ with $i < j$. Values average the per-pair cluster means across GSM8K and MATH500.

| $i$ | $j$ | $n_{\text{pair}}$ | $n_{\text{cluster}}$ | $\rho_s$ | $\rho$ |
|-----|-----|------|------|-------|-------|
| 1 | 2 | 109 | 394 | 0.64 | 0.65 |
| 1 | 3 | 79 | 287 | 0.50 | 0.50 |
| 2 | 3 | 88 | 364 | 0.14 | 0.14 |
| 1 | 4 | 74 | 218 | 0.46 | 0.46 |
| 2 | 4 | 89 | 381 | 0.10 | 0.11 |
| 3 | 4 | 81 | 304 | 0.09 | 0.08 |
| 1 | 5 | 48 | 166 | 0.48 | 0.48 |
| 2 | 5 | 56 | 241 | 0.10 | 0.11 |
| 3 | 5 | 58 | 229 | 0.05 | 0.04 |
| 4 | 5 | 44 | 145 | -0.17 | -0.19 |
| 1 | 6 | 55 | 197 | 0.46 | 0.47 |
| 2 | 6 | 63 | 299 | 0.02 | 0.01 |
| 3 | 6 | 59 | 230 | 0.02 | 0.03 |
| 4 | 6 | 54 | 172 | 0.07 | 0.05 |
| 5 | 6 | 41 | 136 | -0.03 | -0.05 |
| 1 | 7 | 58 | 220 | 0.49 | 0.51 |
| 2 | 7 | 57 | 257 | 0.03 | 0.03 |
| 3 | 7 | 60 | 257 | 0.01 | 0.01 |
| 4 | 7 | 55 | 206 | 0.01 | 0.02 |
| 5 | 7 | 44 | 141 | 0.03 | 0.05 |
| 6 | 7 | 36 | 123 | -0.03 | -0.02 |
| 1 | 8 | 45 | 182 | 0.39 | 0.41 |
| 2 | 8 | 47 | 203 | 0.01 | 0.02 |
| 3 | 8 | 46 | 185 | 0.03 | 0.03 |
| 4 | 8 | 41 | 186 | 0.01 | 0.01 |
| 5 | 8 | 43 | 167 | 0.06 | 0.02 |

| $i$ | $j$ | $n_{\text{pair}}$ | $n_{\text{cluster}}$ | $\rho_s$ | $\rho$ |
|---|---|---|---|---|---|
| 6 | 8 | 32 | 110 | -0.09 | -0.08 |
| 7 | 8 | 27 | 73 | -0.06 | -0.09 |
| 1 | 9 | 45 | 166 | 0.38 | 0.39 |
| 2 | 9 | 42 | 170 | 0.04 | 0.02 |
| 3 | 9 | 39 | 176 | -0.00 | -0.03 |
| 4 | 9 | 36 | 131 | 0.01 | -0.01 |
| 5 | 9 | 39 | 154 | -0.01 | -0.03 |
| 6 | 9 | 35 | 137 | 0.05 | 0.04 |
| 7 | 9 | 30 | 102 | -0.03 | -0.01 |
| 8 | 9 | 19 | 61 | 0.13 | 0.13 |
| 1 | 10 | 38 | 160 | 0.32 | 0.34 |
| 2 | 10 | 36 | 175 | 0.05 | 0.04 |
| 3 | 10 | 32 | 124 | 0.00 | 0.02 |
| 4 | 10 | 29 | 115 | 0.01 | 0.01 |
| 5 | 10 | 30 | 121 | 0.04 | 0.03 |
| 6 | 10 | 28 | 125 | -0.02 | -0.03 |
| 7 | 10 | 18 | 65 | -0.07 | -0.07 |
| 8 | 10 | 20 | 77 | -0.03 | -0.03 |
| 9 | 10 | 13 | 46 | 0.06 | 0.07 |
| 1 | 11 | 25 | 93 | 0.33 | 0.35 |
| 2 | 11 | 23 | 113 | 0.11 | 0.09 |
| 3 | 11 | 21 | 83 | 0.01 | 0.04 |
| 4 | 11 | 17 | 72 | 0.05 | 0.02 |
| 5 | 11 | 15 | 46 | -0.11 | -0.11 |
| 6 | 11 | 21 | 82 | 0.02 | 0.04 |
| 7 | 11 | 12 | 47 | -0.02 | 0.02 |
| 8 | 11 | 11 | 47 | 0.08 | 0.05 |
| 9 | 11 | 10 | 49 | 0.04 | 0.07 |
| 10 | 11 | 9 | 33 | 0.16 | 0.15 |
| 1 | 12 | 25 | 99 | 0.27 | 0.29 |
| 2 | 12 | 18 | 70 | -0.07 | -0.03 |
| 3 | 12 | 15 | 65 | -0.02 | -0.01 |
| 4 | 12 | 16 | 70 | 0.07 | 0.07 |
| 5 | 12 | 15 | 74 | 0.06 | 0.09 |
| 6 | 12 | 19 | 93 | 0.00 | -0.02 |
| 7 | 12 | 12 | 47 | 0.06 | 0.02 |
| 8 | 12 | 13 | 52 | -0.00 | -0.03 |
| 9 | 12 | 11 | 45 | -0.00 | -0.03 |
| 10 | 12 | 10 | 39 | 0.08 | 0.11 |
| 11 | 12 | 9 | 39 | -0.08 | -0.06 |

**Qualitative results.** Figures 15 and 16 show two representative examples. The fixed target-cluster view holds the target step $j$ fixed and projects each prior source step $i < j$ onto the same target-step clusters. For the GSM8K example with $j = 6$, both target clusters are strongly positive at $i = 1$ ($\rho_s \approx 0.96$ and $1.00$), but become negative for later source steps such as $i = 4$ and $i = 5$. For the MATH500 geometry example with $j = 7$, the first source step is again strongly positive, while the third source step is negative for both target clusters and the sixth source step separates the two clusters with opposite signs. These examples show that the influence of previous reasoning steps is not monotone in distance from the target; different prior steps can strengthen, suppress, or separate different target-step clusters.

This view complements the fixed-cluster projection by visualizing how local clusters split and merge as the source step changes. Node heights encode local cluster probability mass, ribbons encode continuation-overlap mass between adjacent source-step clusterings, and node color encodes local $\rho_s$. This results reveals when a later source step changes only the steering strength of an otherwise stable cluster, versus when the local continuation distribution itself re-partitions into different clusters.

**Qualitative target-step cluster labels for MATH500, target $j = 7$**

*Table 10.* Raw qualitative reasoning case. Each row is a sampled continuation for target step $j = 7$. The label vector $(C_1, \ldots, C_6)$ gives the local cluster label assigned when the source step is fixed to previous reasoning step $i$.

| Dataset | $n_{\text{pair}}$ | $n_{\text{valid}}$ | $n_{\text{cluster}}$ | $\rho_s$ | $\rho$ |
|---------|---------|---------|-----------|--------|------|
| GSM8K | 1353 | 1136 | 4377 | 0.11 | 0.11 |
| MATH500 | 1545 | 1309 | 5159 | 0.15 | 0.15 |
| All | 2898 | 2445 | 9536 | 0.13 | 0.13 |

*Table 8.* Reasoning centered-logit steering correlations for Qwen3-0.6B, combined medoid, $\beta = 0.75$, $\gamma = 0.7$, full $h_c$, and top-$B = 5$. Correlations are averaged over valid target-step clusters.

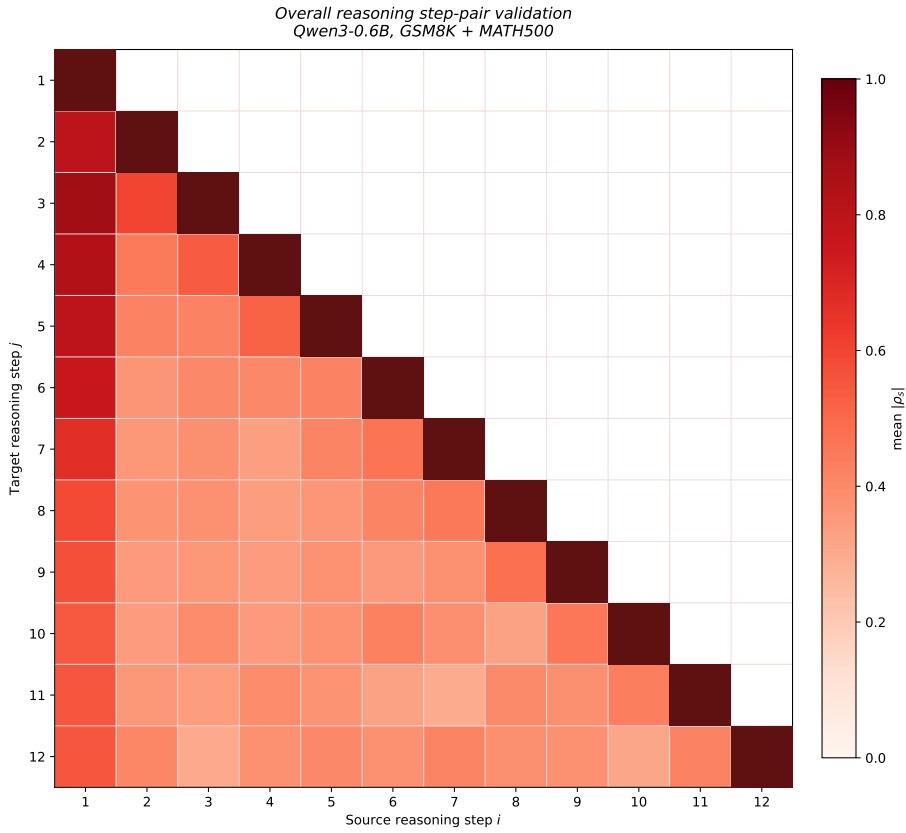

*Figure 14.* Reasoning step-pair validation heatmap for Qwen3-0.6B. Rows are target steps $j$, columns are previous source steps $i < j$, and cells show the mean absolute Spearman correlation $|\rho_s|$ across GSM8K and MATH500.

**Question.** What is the length, in units, of the radius of a sphere whose volume and surface area, in cubic units and square units, respectively, are numerically equal?

**Committed previous reasoning steps.**
1. Recall the formulas for the volume and surface area of a sphere.
2. The volume is $V = \frac{4}{3}\pi r^3$ and the surface area is $A = 4\pi r^2$.
3. Set the numerical equality $\frac{4}{3}\pi r^3 = 4\pi r^2$ and solve for $r$.
4. Divide both sides by $4\pi r^2$, assuming $r \neq 0$.
5. Obtain $\frac{1}{3}r = 1$.
6. Re-check the division of $\frac{4}{3}\pi r^3 = 4\pi r^2$ by $4\pi r^2$.

**Candidate continuations for target step $j = 7$.**

| $p$ | Continuation text | $(C_1, \ldots, C_6)$ |
|-----|------------------|---------------------|
| 0.923 | Left side: the volume term divided by the surface-area term gives $\frac{1}{3}r$. | $(3, 2, 3, 2, 3, 3)$ |
| 0.876 | Left side becomes $\frac{1}{3}r$, and the right side is 1. So this simplifies to $\frac{1}{3}r = 1$; multiplying both sides by 3 gives $r = 3$. | $(2, 11, 5, 7, 3, 3)$ |
| 0.858 | Left side becomes $\frac{r}{3}$ and right side is 1. Thus $r/3 = 1$, so $r = 3$. | $(2, 10, 5, 7, 3, 3)$ |
| 0.843 | Left side becomes $\frac{1}{3}r$, right side becomes 1. Therefore $\frac{1}{3}r = 1$, so $r = 3$ units. | $(2, 5, 4, 5, 2, 2)$ |
| 0.836 | Left side becomes $\frac{1}{3}r^3$ divided by $r^2$, which is $\frac{1}{3}r$. | $(3, 6, 3, 6, 3, 3)$ |
| 0.748 | $\frac{1}{3}r = 1$. | $(3, 2, 3, 2, 3, 2)$ |

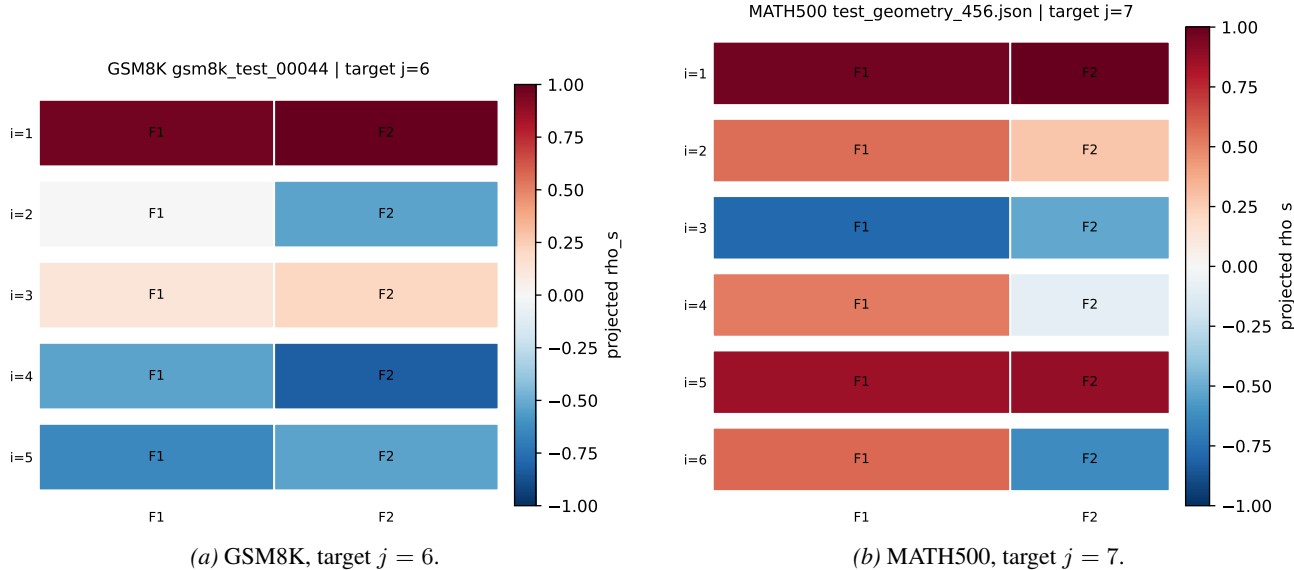

*(a)* GSM8K, target $j = 6$.

*(b)* MATH500, target $j = 7$.

*Figure 15.* Fixed target-cluster effects. Rows are prior source steps $i < j$, columns are fixed clusters of target-step continuations, column widths are target-cluster probability mass, and color is the projected Spearman correlation.

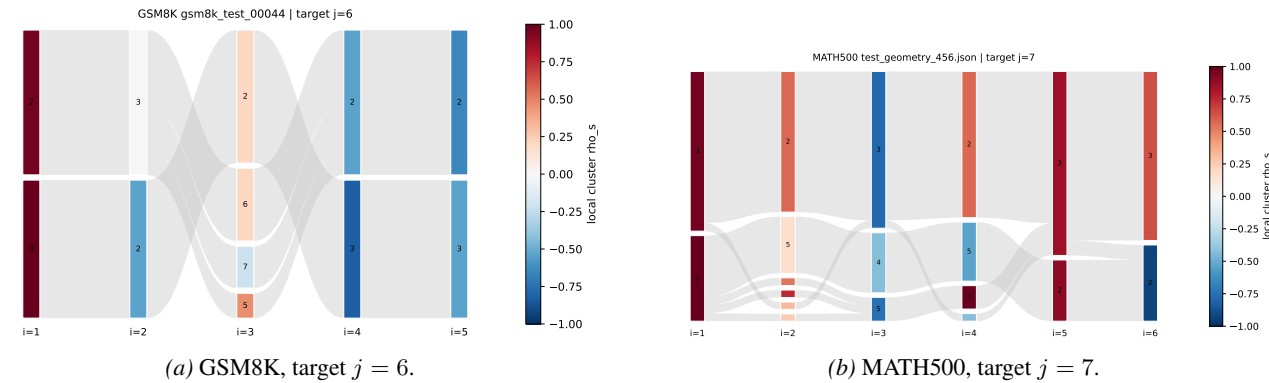

*(a)* GSM8K, target $j = 6$.  *(b)* MATH500, target $j = 7$.

*Figure 16.* Local cluster dynamics across previous reasoning steps. Columns are source steps $i < j$, node heights are local cluster probability mass, ribbons track adjacent continuation-overlap mass, and node colors show local $\rho_s$.

## E.4. AmbigQA on Qwen3-8B: Full steering results

*Table 11.* Complete Steering Results: Method $\times$ Sign $\times$ Beam $\times$ $\beta$ $\times$ $\gamma$

| Method | Sign | B | $\beta$ | $\gamma$ | $\rho_s$ | $\rho$ | $\epsilon_{-1.0}$ | $\epsilon_{-0.5}$ | $\epsilon_{-0.1}$ | $\epsilon_{0.1}$ | $\epsilon_{0.5}$ | $\epsilon_{1.0}$ |
|---|---|---|---|---|---|---|---|---|---|---|---|---|
| **RD** | Full | 5 | 0.75 | 0.3 | 0.35 | 0.38 | -1.94 | -0.76 | -0.16 | 0.16 | 0.39 | 0.36 |
| | | | | 0.5 | 0.27 | 0.30 | -2.55 | -0.72 | -0.17 | 0.10 | 0.46 | 0.58 |
| | | | | 0.7 | 0.32 | 0.33 | -1.67 | -0.78 | -0.21 | 0.04 | 0.26 | 0.44 |
| | | | 1.00 | 0.3 | 0.40 | 0.44 | -1.79 | -0.49 | -0.16 | 0.03 | 0.03 | -0.03 |
| | | | | 0.5 | 0.40 | 0.43 | -2.12 | -0.73 | -0.13 | 0.18 | 0.55 | 0.65 |
| | | | | 0.7 | 0.28 | 0.31 | -2.29 | -0.72 | -0.10 | 0.12 | 0.40 | 0.55 |
| | | | 1.25 | 0.3 | 0.39 | 0.45 | -1.06 | -0.43 | -0.11 | -0.01 | 0.03 | -0.02 |
| | | | | 0.5 | 0.40 | 0.42 | -1.13 | -0.48 | -0.18 | -0.03 | 0.18 | 0.28 |
| | | | | 0.7 | 0.22 | 0.24 | -2.32 | -0.63 | -0.17 | 0.05 | 0.23 | 0.22 |
| | | 10 | 0.75 | 0.3 | 0.36 | 0.38 | -3.05 | -1.25 | -0.19 | 0.27 | 0.61 | 0.59 |
| | | | | 0.5 | 0.26 | 0.28 | -3.42 | -1.14 | -0.18 | 0.24 | 0.66 | 0.85 |
| | | | | 0.7 | 0.31 | 0.33 | -2.79 | -1.04 | -0.27 | 0.19 | 0.59 | 0.70 |
| | | | 1.00 | 0.3 | 0.49 | 0.53 | -2.88 | -0.82 | -0.30 | 0.14 | 0.35 | 0.69 |
| | | | | 0.5 | 0.41 | 0.41 | -3.04 | -0.94 | -0.18 | 0.30 | 0.59 | 0.67 |
| | | | | 0.7 | 0.32 | 0.34 | -3.26 | -1.14 | -0.16 | 0.29 | 0.57 | 0.60 |
| | | | 1.25 | 0.3 | 0.31 | 0.37 | -1.72 | -0.53 | -0.15 | 0.01 | 0.09 | 0.24 |
| | | | | 0.5 | 0.41 | 0.43 | -2.14 | -0.71 | -0.21 | 0.14 | 0.39 | 0.59 |
| | | | | 0.7 | 0.24 | 0.25 | -3.18 | -1.08 | -0.18 | 0.20 | 0.46 | 0.47 |
| | Positive | 5 | 0.75 | 0.3 | 0.32 | 0.35 | -2.06 | -0.77 | -0.14 | 0.12 | 0.29 | 0.23 |
| | | | | 0.5 | 0.29 | 0.31 | -2.62 | -0.76 | -0.13 | 0.08 | 0.45 | 0.58 |
| | | | | 0.7 | 0.31 | 0.33 | -1.91 | -0.79 | -0.15 | 0.07 | 0.36 | 0.51 |
| | | | 1.00 | 0.3 | 0.39 | 0.43 | -1.98 | -0.53 | -0.20 | -0.05 | -0.13 | -0.20 |
| | | | | 0.5 | 0.39 | 0.42 | -2.51 | -0.76 | -0.09 | 0.16 | 0.51 | 0.60 |
| | | | | 0.7 | 0.29 | 0.32 | -2.55 | -0.77 | -0.08 | 0.12 | 0.45 | 0.57 |
| | | | 1.25 | 0.3 | 0.42 | 0.48 | -1.24 | -0.44 | -0.18 | -0.06 | -0.07 | -0.13 |
| | | | | 0.5 | 0.36 | 0.38 | -1.19 | -0.43 | -0.18 | -0.05 | 0.01 | -0.01 |
| | | | | 0.7 | 0.24 | 0.26 | -2.39 | -0.70 | -0.15 | 0.03 | 0.24 | 0.30 |
| | | 10 | 0.75 | 0.3 | 0.36 | 0.38 | -2.38 | -1.10 | -0.17 | 0.21 | 0.61 | 0.42 |
| | | | | 0.5 | 0.30 | 0.33 | -3.11 | -0.99 | -0.18 | 0.14 | 0.55 | 0.62 |
| | | | | 0.7 | 0.34 | 0.36 | -2.51 | -0.89 | -0.25 | 0.14 | 0.59 | 0.68 |

Continued on next page

| Method | Sign | B | $\beta$ | $\gamma$ | $\rho_s$ | $\rho$ | $\epsilon_{-1.0}$ | $\epsilon_{-0.5}$ | $\epsilon_{-0.1}$ | $\epsilon_{0.1}$ | $\epsilon_{0.5}$ | $\epsilon_{1.0}$ |
|---|---|---|---|---|---|---|---|---|---|---|---|---|
| | | | 1.00 | 0.3 | 0.37 | 0.43 | -1.94 | -0.61 | -0.15 | -0.01 | 0.10 | 0.04 |
| | | | | 0.5 | 0.42 | 0.44 | -2.53 | -0.79 | -0.14 | 0.21 | 0.55 | 0.45 |
| | | | | 0.7 | 0.31 | 0.33 | -3.00 | -0.96 | -0.16 | 0.27 | 0.61 | 0.64 |
| | | | 1.25 | 0.3 | 0.30 | 0.38 | -1.45 | -0.38 | -0.05 | 0.04 | -0.01 | -0.12 |
| | | | | 0.5 | 0.33 | 0.37 | -1.58 | -0.50 | -0.08 | -0.01 | 0.22 | 0.05 |
| | | | | 0.7 | 0.27 | 0.29 | -2.94 | -0.98 | -0.22 | 0.14 | 0.44 | 0.45 |
| | Negative | 5 | 0.75 | 0.3 | 0.19 | 0.17 | -0.12 | -0.25 | -0.19 | 0.10 | 0.33 | 0.41 |
| | | | | 0.5 | 0.06 | 0.05 | -0.38 | -0.34 | -0.15 | 0.06 | 0.34 | 0.47 |
| | | | | 0.7 | 0.14 | 0.13 | -0.22 | -0.18 | -0.11 | 0.06 | 0.21 | 0.37 |
| | | | 1.00 | 0.3 | 0.19 | 0.20 | -0.47 | -0.51 | -0.37 | 0.12 | 0.67 | 1.44 |
| | | | | 0.5 | 0.12 | 0.12 | -0.13 | -0.22 | -0.14 | 0.08 | 0.33 | 0.61 |
| | | | | 0.7 | 0.08 | 0.07 | -0.16 | -0.11 | -0.06 | 0.14 | 0.18 | 0.17 |
| | | | 1.25 | 0.3 | 0.07 | 0.09 | 0.03 | -0.22 | -0.22 | 0.04 | 0.34 | 0.89 |
| | | | | 0.5 | 0.13 | 0.13 | -0.30 | -0.37 | -0.29 | 0.10 | 0.57 | 1.25 |
| | | | | 0.7 | 0.09 | 0.08 | -0.16 | -0.26 | -0.21 | 0.03 | 0.15 | 0.20 |
| | | 10 | 0.75 | 0.3 | 0.19 | 0.17 | -0.15 | -0.27 | -0.19 | 0.09 | 0.34 | 0.43 |
| | | | | 0.5 | 0.07 | 0.06 | -0.40 | -0.35 | -0.14 | 0.06 | 0.35 | 0.48 |
| | | | | 0.7 | 0.14 | 0.14 | -0.25 | -0.19 | -0.11 | 0.06 | 0.22 | 0.39 |
| | | | 1.00 | 0.3 | 0.20 | 0.21 | -0.47 | -0.51 | -0.36 | 0.11 | 0.67 | 1.45 |
| | | | | 0.5 | 0.12 | 0.12 | -0.13 | -0.21 | -0.14 | 0.07 | 0.32 | 0.62 |
| | | | | 0.7 | 0.08 | 0.07 | -0.20 | -0.13 | -0.06 | 0.14 | 0.19 | 0.19 |
| | | | 1.25 | 0.3 | 0.08 | 0.10 | 0.02 | -0.22 | -0.22 | 0.04 | 0.34 | 0.90 |
| | | | | 0.5 | 0.13 | 0.13 | -0.30 | -0.36 | -0.28 | 0.09 | 0.57 | 1.26 |
| | | | | 0.7 | 0.09 | 0.08 | -0.18 | -0.26 | -0.21 | 0.02 | 0.17 | 0.22 |
| **KM-Sem** | Full | 5 | 0.75 | 0.3 | -0.05 | -0.05 | -0.59 | -0.08 | -0.05 | 0.04 | -0.11 | -0.84 |
| | | | | 0.5 | -0.04 | -0.03 | -0.60 | -0.17 | -0.02 | -0.01 | -0.04 | -0.65 |
| | | | | 0.7 | 0.04 | 0.04 | -0.70 | -0.22 | -0.12 | -0.02 | -0.02 | -0.35 |
| | | | 1.00 | 0.3 | -0.05 | -0.06 | -0.12 | -0.09 | -0.13 | -0.09 | -0.17 | -0.44 |
| | | | | 0.5 | -0.07 | -0.07 | -0.30 | 0.03 | 0.14 | 0.10 | 0.07 | -0.41 |
| | | | | 0.7 | 0.02 | 0.02 | -0.90 | -0.16 | 0.00 | 0.11 | 0.06 | -0.53 |
| | | | 1.25 | 0.3 | -0.12 | -0.13 | -0.16 | -0.04 | -0.03 | -0.11 | -0.13 | -0.27 |
| | | | | 0.5 | -0.12 | -0.13 | 0.14 | 0.18 | 0.04 | -0.15 | -0.40 | -0.77 |
| | | | | 0.7 | -0.05 | -0.04 | -0.40 | -0.17 | -0.02 | -0.05 | -0.11 | -0.73 |
| | | 10 | 0.75 | 0.3 | -0.02 | -0.01 | -1.07 | -0.20 | 0.05 | 0.05 | -0.32 | -1.09 |
| | | | | 0.5 | -0.02 | -0.02 | -0.80 | -0.10 | 0.09 | 0.03 | -0.20 | -0.98 |
| | | | | 0.7 | 0.04 | 0.04 | -0.71 | -0.12 | 0.02 | -0.02 | -0.07 | -0.54 |
| | | | 1.00 | 0.3 | -0.15 | -0.17 | -0.13 | 0.01 | -0.02 | -0.02 | -0.27 | -0.90 |
| | | | | 0.5 | -0.04 | -0.05 | -0.61 | 0.10 | 0.20 | 0.12 | -0.17 | -0.90 |
| | | | | 0.7 | 0.04 | 0.05 | -1.07 | -0.19 | 0.10 | 0.05 | -0.00 | -0.66 |
| | | | 1.25 | 0.3 | -0.20 | -0.21 | 0.02 | 0.16 | 0.10 | -0.03 | -0.38 | -0.60 |
| | | | | 0.5 | -0.17 | -0.17 | -0.09 | 0.23 | 0.13 | -0.04 | -0.64 | -1.18 |
| | | | | 0.7 | -0.02 | -0.03 | -0.66 | -0.21 | -0.03 | -0.06 | -0.17 | -0.88 |
| | Positive | 5 | 0.75 | 0.3 | 0.10 | 0.12 | -1.09 | -0.31 | 0.01 | 0.09 | 0.04 | -0.02 |
| | | | | 0.5 | 0.11 | 0.12 | -1.01 | -0.27 | -0.06 | 0.02 | 0.10 | 0.06 |
| | | | | 0.7 | 0.12 | 0.13 | -0.94 | -0.27 | -0.12 | 0.08 | 0.15 | 0.18 |
| | | | 1.00 | 0.3 | 0.01 | 0.02 | -0.36 | -0.14 | -0.09 | -0.01 | -0.14 | 0.03 |
| | | | | 0.5 | 0.11 | 0.13 | -0.65 | -0.13 | 0.13 | 0.14 | 0.08 | 0.22 |
| | | | | 0.7 | 0.15 | 0.17 | -1.43 | -0.37 | -0.03 | 0.14 | 0.23 | 0.28 |
| | | | 1.25 | 0.3 | -0.01 | -0.01 | -0.33 | -0.04 | -0.02 | -0.09 | -0.18 | -0.14 |

| Method | Sign | B | $\beta$ | $\gamma$ | $\rho_s$ | $\rho$ | $\epsilon_{-1.0}$ | $\epsilon_{-0.5}$ | $\epsilon_{-0.1}$ | $\epsilon_{0.1}$ | $\epsilon_{0.5}$ | $\epsilon_{1.0}$ |
|---|---|---|---|---|---|---|---|---|---|---|---|---|
| | | | | 0.5 | 0.02 | 0.04 | -0.42 | -0.03 | 0.02 | -0.04 | -0.14 | -0.10 |
| | | | | 0.7 | 0.08 | 0.10 | -0.83 | -0.26 | -0.07 | -0.07 | -0.03 | -0.08 |
| | | 10 | 0.75 | 0.3 | 0.08 | 0.08 | -1.29 | -0.26 | -0.12 | 0.10 | 0.01 | -0.30 |
| | | | | 0.5 | 0.09 | 0.10 | -1.46 | -0.42 | -0.11 | 0.12 | 0.25 | 0.13 |
| | | | | 0.7 | 0.11 | 0.12 | -1.31 | -0.29 | -0.10 | 0.16 | 0.30 | 0.44 |
| | | | 1.00 | 0.3 | 0.06 | 0.05 | -0.38 | -0.08 | -0.15 | 0.06 | -0.21 | -0.20 |
| | | | | 0.5 | 0.12 | 0.13 | -0.81 | -0.16 | 0.05 | 0.21 | 0.22 | 0.14 |
| | | | | 0.7 | 0.13 | 0.14 | -1.93 | -0.38 | -0.05 | 0.24 | 0.34 | 0.46 |
| | | | 1.25 | 0.3 | 0.01 | 0.00 | -0.20 | 0.03 | 0.01 | -0.01 | -0.12 | -0.19 |
| | | | | 0.5 | 0.03 | 0.03 | -0.48 | 0.01 | -0.02 | 0.00 | -0.13 | -0.26 |
| | | | | 0.7 | 0.06 | 0.08 | -1.17 | -0.41 | -0.14 | 0.01 | 0.04 | -0.06 |
| | Negative | 5 | 0.75 | 0.3 | -0.11 | -0.12 | -0.11 | 0.02 | 0.05 | -0.04 | -0.46 | -1.10 |
| | | | | 0.5 | -0.11 | -0.12 | 0.07 | 0.15 | 0.00 | -0.09 | -0.38 | -1.22 |
| | | | | 0.7 | 0.00 | 0.01 | 0.10 | 0.00 | -0.01 | -0.09 | -0.30 | -0.94 |
| | | | 1.00 | 0.3 | -0.15 | -0.16 | 0.10 | 0.11 | -0.10 | -0.13 | -0.43 | -0.95 |
| | | | | 0.5 | -0.12 | -0.13 | 0.06 | 0.27 | 0.16 | 0.03 | -0.21 | -0.88 |
| | | | | 0.7 | -0.05 | -0.06 | 0.03 | 0.11 | 0.08 | 0.02 | -0.27 | -1.04 |
| | | | 1.25 | 0.3 | -0.22 | -0.23 | 0.12 | 0.02 | -0.03 | -0.12 | -0.30 | -0.62 |
| | | | | 0.5 | -0.19 | -0.20 | 0.39 | 0.30 | 0.03 | -0.16 | -0.59 | -1.25 |
| | | | | 0.7 | -0.07 | -0.09 | -0.07 | 0.01 | -0.06 | -0.12 | -0.35 | -1.22 |
| | | 10 | 0.75 | 0.3 | -0.10 | -0.12 | -0.19 | 0.01 | 0.07 | -0.02 | -0.50 | -1.30 |
| | | | | 0.5 | -0.12 | -0.13 | 0.22 | 0.19 | 0.08 | -0.05 | -0.55 | -1.68 |
| | | | | 0.7 | -0.03 | -0.03 | 0.21 | 0.02 | 0.07 | -0.07 | -0.52 | -1.43 |
| | | | 1.00 | 0.3 | -0.19 | -0.20 | 0.40 | 0.22 | 0.12 | -0.10 | -0.40 | -1.06 |
| | | | | 0.5 | -0.13 | -0.15 | -0.01 | 0.21 | 0.26 | 0.12 | -0.29 | -1.25 |
| | | | | 0.7 | -0.08 | -0.09 | 0.26 | 0.11 | 0.09 | 0.00 | -0.46 | -1.49 |
| | | | 1.25 | 0.3 | -0.21 | -0.21 | 0.05 | 0.02 | 0.02 | -0.14 | -0.35 | -0.79 |
| | | | | 0.5 | -0.24 | -0.25 | 0.54 | 0.40 | 0.19 | -0.12 | -0.82 | -1.68 |
| | | | | 0.7 | -0.08 | -0.10 | 0.01 | -0.01 | -0.04 | -0.07 | -0.42 | -1.59 |
| **Single** | Full | 5 | 0.75 | 0.3 | 0.18 | 0.20 | -0.89 | -0.34 | -0.13 | 0.01 | 0.04 | 0.13 |
| | | | | 0.5 | 0.21 | 0.23 | -1.38 | -0.46 | -0.11 | 0.11 | 0.29 | 0.30 |
| | | | | 0.7 | 0.19 | 0.20 | -1.08 | -0.42 | -0.10 | 0.05 | 0.22 | 0.30 |
| | | | 1.00 | 0.3 | 0.26 | 0.28 | -1.40 | -0.50 | -0.19 | 0.10 | 0.18 | 0.23 |
| | | | | 0.5 | 0.19 | 0.20 | -1.17 | -0.46 | -0.09 | 0.08 | 0.26 | 0.33 |
| | | | | 0.7 | 0.16 | 0.17 | -1.28 | -0.41 | -0.04 | 0.11 | 0.17 | 0.12 |
| | | | 1.25 | 0.3 | 0.28 | 0.32 | -1.30 | -0.47 | -0.19 | 0.05 | 0.22 | 0.42 |
| | | | | 0.5 | 0.20 | 0.20 | -1.26 | -0.48 | -0.18 | 0.15 | 0.17 | 0.23 |
| | | | | 0.7 | 0.15 | 0.18 | -1.29 | -0.40 | -0.09 | 0.03 | 0.16 | 0.13 |
| | | 10 | 0.75 | 0.3 | 0.19 | 0.21 | -1.59 | -0.62 | -0.09 | 0.08 | 0.12 | 0.17 |
| | | | | 0.5 | 0.17 | 0.18 | -1.81 | -0.65 | -0.08 | 0.14 | 0.26 | 0.26 |
| | | | | 0.7 | 0.19 | 0.20 | -1.58 | -0.66 | -0.11 | 0.10 | 0.27 | 0.28 |
| | | | 1.00 | 0.3 | 0.34 | 0.36 | -2.28 | -0.87 | -0.20 | 0.05 | 0.15 | 0.28 |
| | | | | 0.5 | 0.19 | 0.20 | -1.78 | -0.72 | -0.11 | 0.12 | 0.23 | 0.15 |
| | | | | 0.7 | 0.15 | 0.16 | -1.85 | -0.63 | -0.09 | 0.10 | 0.12 | 0.08 |
| | | | 1.25 | 0.3 | 0.20 | 0.23 | -2.02 | -0.84 | -0.25 | 0.06 | 0.19 | -0.01 |
| | | | | 0.5 | 0.24 | 0.24 | -2.05 | -0.79 | -0.13 | 0.15 | 0.31 | 0.35 |
| | | | | 0.7 | 0.12 | 0.14 | -1.92 | -0.66 | -0.09 | 0.07 | 0.15 | 0.12 |
| | Positive | 5 | 0.75 | 0.3 | 0.17 | 0.19 | -0.93 | -0.31 | -0.07 | -0.00 | 0.01 | 0.03 |

| Method | Sign | B | $\beta$ | $\gamma$ | $\rho_s$ | $\rho$ | $\epsilon_{-1.0}$ | $\epsilon_{-0.5}$ | $\epsilon_{-0.1}$ | $\epsilon_{0.1}$ | $\epsilon_{0.5}$ | $\epsilon_{1.0}$ |
|---|---|---|---|---|---|---|---|---|---|---|---|---|
| | | | | 0.5 | 0.20 | 0.22 | -1.38 | -0.43 | -0.08 | 0.11 | 0.29 | 0.28 |
| | | | | 0.7 | 0.17 | 0.18 | -1.11 | -0.40 | -0.08 | 0.05 | 0.19 | 0.23 |
| | | | 1.00 | 0.3 | 0.28 | 0.29 | -1.49 | -0.48 | -0.13 | 0.06 | 0.16 | 0.13 |
| | | | | 0.5 | 0.19 | 0.20 | -1.17 | -0.47 | -0.12 | 0.06 | 0.26 | 0.28 |
| | | | | 0.7 | 0.15 | 0.16 | -1.31 | -0.42 | -0.04 | 0.08 | 0.17 | 0.11 |
| | | | 1.25 | 0.3 | 0.27 | 0.30 | -1.45 | -0.49 | -0.15 | 0.04 | 0.12 | 0.17 |
| | | | | 0.5 | 0.18 | 0.18 | -1.28 | -0.47 | -0.16 | 0.09 | 0.15 | 0.22 |
| | | | | 0.7 | 0.13 | 0.15 | -1.32 | -0.39 | -0.07 | 0.02 | 0.11 | 0.04 |
| | | 10 | 0.75 | 0.3 | 0.17 | 0.19 | -1.58 | -0.59 | -0.10 | 0.09 | 0.13 | 0.17 |
| | | | | 0.5 | 0.16 | 0.18 | -1.76 | -0.63 | -0.10 | 0.16 | 0.27 | 0.21 |
| | | | | 0.7 | 0.15 | 0.16 | -1.40 | -0.57 | -0.08 | 0.08 | 0.22 | 0.18 |
| | | | 1.00 | 0.3 | 0.30 | 0.32 | -2.03 | -0.72 | -0.19 | 0.09 | 0.16 | 0.15 |
| | | | | 0.5 | 0.19 | 0.20 | -1.76 | -0.72 | -0.12 | 0.14 | 0.26 | 0.20 |
| | | | | 0.7 | 0.13 | 0.14 | -1.71 | -0.58 | -0.07 | 0.12 | 0.16 | 0.06 |
| | | | 1.25 | 0.3 | 0.19 | 0.21 | -2.10 | -0.72 | -0.18 | 0.13 | 0.07 | 0.01 |
| | | | | 0.5 | 0.16 | 0.17 | -1.90 | -0.75 | -0.09 | 0.18 | 0.28 | 0.16 |
| | | | | 0.7 | 0.11 | 0.12 | -1.87 | -0.63 | -0.08 | 0.08 | 0.15 | -0.01 |
| | Negative | 5 | 0.75 | 0.3 | 0.01 | 0.01 | -0.14 | -0.07 | -0.05 | 0.02 | -0.06 | -0.05 |
| | | | | 0.5 | 0.00 | -0.00 | -0.09 | -0.07 | -0.03 | -0.01 | -0.03 | -0.08 |
| | | | | 0.7 | 0.06 | 0.06 | -0.14 | -0.05 | 0.00 | 0.02 | 0.03 | 0.03 |
| | | | 1.00 | 0.3 | 0.02 | 0.02 | -0.30 | -0.17 | -0.17 | -0.01 | 0.13 | 0.25 |
| | | | | 0.5 | 0.04 | 0.04 | -0.24 | -0.05 | -0.03 | 0.03 | -0.01 | 0.10 |
| | | | | 0.7 | 0.05 | 0.05 | -0.12 | -0.02 | 0.05 | 0.05 | 0.04 | 0.05 |
| | | | 1.25 | 0.3 | 0.02 | 0.02 | -0.36 | -0.10 | -0.22 | -0.04 | 0.25 | 0.31 |
| | | | | 0.5 | 0.10 | 0.10 | -0.30 | -0.17 | -0.15 | 0.03 | 0.35 | 0.37 |
| | | | | 0.7 | 0.05 | 0.05 | -0.21 | -0.13 | -0.06 | -0.01 | 0.01 | 0.13 |
| | | 10 | 0.75 | 0.3 | 0.02 | 0.01 | -0.15 | -0.07 | -0.04 | 0.02 | -0.06 | -0.07 |
| | | | | 0.5 | 0.00 | -0.00 | -0.09 | -0.06 | -0.03 | -0.00 | -0.02 | -0.08 |
| | | | | 0.7 | 0.06 | 0.06 | -0.14 | -0.06 | 0.00 | 0.02 | 0.02 | 0.02 |
| | | | 1.00 | 0.3 | 0.03 | 0.04 | -0.32 | -0.18 | -0.17 | 0.01 | 0.13 | 0.28 |
| | | | | 0.5 | 0.03 | 0.04 | -0.23 | -0.05 | -0.03 | 0.03 | -0.01 | 0.08 |
| | | | | 0.7 | 0.05 | 0.06 | -0.11 | -0.01 | 0.06 | 0.04 | 0.05 | 0.06 |
| | | | 1.25 | 0.3 | 0.03 | 0.03 | -0.34 | -0.10 | -0.24 | -0.04 | 0.27 | 0.32 |
| | | | | 0.5 | 0.10 | 0.10 | -0.30 | -0.16 | -0.15 | 0.04 | 0.35 | 0.38 |
| | | | | 0.7 | 0.04 | 0.05 | -0.20 | -0.13 | -0.05 | -0.01 | 0.01 | 0.12 |

### E.5. AmbigQA on Qwen3-4B: Replication Results

**Clustering results.**

| Method | $\gamma$ | 0.50 | | | | 0.75 | | | | 1.00 | | | | 1.25 | | | |
|---|---|---|---|---|---|---|---|---|---|---|---|---|---|---|---|---|---|
| | | $H$ | $D^{(e)}$ | $D^{(a)}$ | $D_\gamma$ | $H$ | $D^{(e)}$ | $D^{(a)}$ | $D_\gamma$ | $H$ | $D^{(e)}$ | $D^{(a)}$ | $D_\gamma$ | $H$ | $D^{(e)}$ | $D^{(a)}$ | $D_\gamma$ |
| **RD** | 0.1 | 1.95 | 0.19 | 10.4 | **9.34** | 2.39 | 0.18 | 9.0 | **8.15** | 2.60 | 0.17 | 8.4 | **7.61** | 2.67 | 0.17 | 8.3 | **7.52** |
| | 0.3 | 1.55 | 0.20 | 11.8 | **8.31** | 2.17 | 0.18 | 9.6 | **6.80** | 2.45 | 0.18 | 8.9 | **6.25** | 2.56 | 0.17 | 8.5 | **6.01** |
| | 0.5 | 1.09 | 0.22 | 13.9 | **7.04** | 1.64 | 0.20 | 11.5 | **5.84** | 2.13 | 0.19 | 9.7 | **4.96** | 2.37 | 0.18 | 9.1 | **4.62** |
| | 0.7 | 0.40 | 0.25 | 18.1 | **5.60** | 0.86 | 0.23 | 15.1 | **4.69** | 1.35 | 0.21 | 12.6 | **3.93** | 1.64 | 0.20 | 11.4 | **3.57** |
| | 0.9 | 0.00 | 0.28 | 22.3 | **2.49** | 0.10 | 0.27 | 20.7 | **2.31** | 0.18 | 0.27 | 19.8 | **2.22** | 0.29 | 0.26 | 18.9 | **2.12** |
| **KM-S** | 0.1 | 1.92 | 0.15 | 38.7 | 34.81 | 2.21 | 0.14 | 37.0 | 33.29 | 2.35 | 0.13 | 36.6 | 32.93 | 2.40 | 0.13 | 36.4 | 32.78 |
| | 0.3 | 1.57 | 0.16 | 41.0 | 28.77 | 2.06 | 0.14 | 38.1 | 26.68 | 2.26 | 0.13 | 36.6 | 25.67 | 2.33 | 0.13 | 36.6 | 25.67 |
| | 0.5 | 1.14 | 0.18 | 44.2 | 22.21 | 1.65 | 0.16 | 40.4 | 20.27 | 2.04 | 0.14 | 38.1 | 19.11 | 2.20 | 0.13 | 37.2 | 18.66 |
| | 0.7 | 0.68 | 0.20 | 47.7 | 14.46 | 0.97 | 0.19 | 45.8 | 13.87 | 1.38 | 0.17 | 42.6 | 12.89 | 1.65 | 0.16 | 39.9 | 12.07 |
| | 0.9 | 0.60 | 0.21 | 48.7 | 5.06 | 0.61 | 0.21 | 48.7 | 5.05 | 0.61 | 0.21 | 48.6 | 5.05 | 0.64 | 0.21 | 48.3 | 5.02 |
| **KM-A** | 0.1 | 1.92 | 0.19 | 26.1 | 23.52 | 2.20 | 0.18 | 24.1 | 21.69 | 2.34 | 0.18 | 23.3 | 20.98 | 2.37 | 0.17 | 23.2 | 20.91 |
| | 0.3 | 1.56 | 0.20 | 29.8 | 20.93 | 2.06 | 0.18 | 25.1 | 17.65 | 2.24 | 0.18 | 23.8 | 16.70 | 2.32 | 0.18 | 23.4 | 16.43 |
| | 0.5 | 1.15 | 0.21 | 35.1 | 17.67 | 1.66 | 0.19 | 28.8 | 14.49 | 2.03 | 0.18 | 25.3 | 12.72 | 2.20 | 0.18 | 24.1 | 12.14 |
| | 0.7 | 0.68 | 0.23 | 41.9 | 12.74 | 0.98 | 0.22 | 37.3 | 11.34 | 1.40 | 0.20 | 31.8 | 9.68 | 1.66 | 0.19 | 28.7 | 8.75 |
| | 0.9 | 0.59 | 0.23 | 44.0 | 4.61 | 0.60 | 0.23 | 43.7 | 4.58 | 0.60 | 0.23 | 43.7 | 4.58 | 0.63 | 0.23 | 43.0 | 4.51 |

*Table 12.* **RD vs. K-means at matched rate $H$ (Qwen3-4B).** Bold indicates lowest $D_\gamma$.

**Steering results.**

*Table 13.* Complete Steering Results: Method $\times$ Sign $\times$ Beam $\times$ $\beta$ $\times$ $\gamma$ (Qwen3-4B)

| Method | Sign | B | $\beta$ | $\gamma$ | $\rho_s$ | $\rho$ | $\epsilon_{-1.0}$ | $\epsilon_{-0.5}$ | $\epsilon_{-0.1}$ | $\epsilon_{0.1}$ | $\epsilon_{0.5}$ | $\epsilon_{1.0}$ |
|---|---|---|---|---|---|---|---|---|---|---|---|---|
| **RD** | Full | 5 | 0.75 | 0.3 | 0.83 | 0.84 | -0.82 | -0.43 | -0.06 | 0.14 | 0.65 | 1.30 |
| | | | | 0.5 | 0.82 | 0.84 | -0.78 | -0.45 | -0.11 | 0.11 | 0.63 | 1.44 |
| | | | | 0.7 | 0.74 | 0.75 | -0.88 | -0.51 | -0.13 | 0.13 | 0.72 | 1.56 |
| | | | 1.00 | 0.3 | 0.75 | 0.79 | -0.81 | -0.41 | -0.06 | 0.10 | 0.63 | 1.34 |
| | | | | 0.5 | 0.84 | 0.85 | -0.88 | -0.49 | -0.09 | 0.12 | 0.67 | 1.40 |
| | | | | 0.7 | 0.76 | 0.77 | -0.83 | -0.48 | -0.13 | 0.13 | 0.66 | 1.47 |
| | | | 1.25 | 0.3 | 0.74 | 0.77 | -0.79 | -0.47 | -0.06 | 0.08 | 0.53 | 1.02 |
| | | | | 0.5 | 0.80 | 0.81 | -0.84 | -0.45 | -0.07 | 0.13 | 0.66 | 1.33 |
| | | | | 0.7 | 0.83 | 0.84 | -0.82 | -0.47 | -0.11 | 0.11 | 0.67 | 1.52 |
| | | 10 | 0.75 | 0.3 | 0.81 | 0.82 | -1.30 | -0.74 | -0.12 | 0.19 | 0.89 | 1.68 |
| | | | | 0.5 | 0.76 | 0.77 | -1.27 | -0.77 | -0.15 | 0.16 | 0.91 | 1.90 |
| | | | | 0.7 | 0.68 | 0.67 | -1.36 | -0.82 | -0.18 | 0.20 | 1.01 | 2.15 |
| | | | 1.00 | 0.3 | 0.75 | 0.78 | -1.35 | -0.76 | -0.10 | 0.13 | 0.84 | 1.67 |
| | | | | 0.5 | 0.82 | 0.83 | -1.43 | -0.83 | -0.16 | 0.17 | 0.98 | 1.97 |
| | | | | 0.7 | 0.71 | 0.71 | -1.41 | -0.81 | -0.17 | 0.18 | 0.95 | 2.02 |
| | | | 1.25 | 0.3 | 0.73 | 0.74 | -1.01 | -0.57 | -0.14 | 0.15 | 0.60 | 1.09 |
| | | | | 0.5 | 0.77 | 0.77 | -1.30 | -0.79 | -0.16 | 0.19 | 0.91 | 1.75 |
| | | | | 0.7 | 0.78 | 0.79 | -1.38 | -0.85 | -0.17 | 0.18 | 1.00 | 2.09 |
| | Positive | 5 | 0.75 | 0.3 | 0.49 | 0.51 | -0.49 | -0.21 | -0.03 | 0.04 | 0.25 | 0.41 |
| | | | | 0.5 | 0.37 | 0.38 | -0.43 | -0.21 | -0.06 | 0.02 | 0.22 | 0.47 |
| | | | | 0.7 | 0.22 | 0.23 | -0.39 | -0.23 | -0.06 | 0.05 | 0.25 | 0.55 |
| | | | 1.00 | 0.3 | 0.36 | 0.40 | -0.43 | -0.17 | -0.03 | -0.04 | 0.21 | 0.34 |
| | | | | 0.5 | 0.46 | 0.48 | -0.52 | -0.26 | -0.04 | 0.02 | 0.29 | 0.52 |
| | | | | 0.7 | 0.34 | 0.36 | -0.51 | -0.26 | -0.07 | 0.03 | 0.27 | 0.57 |
| | | | 1.25 | 0.3 | 0.26 | 0.28 | -0.23 | -0.11 | 0.02 | 0.01 | 0.04 | 0.02 |
| | | | | 0.5 | 0.41 | 0.43 | -0.45 | -0.21 | -0.03 | 0.01 | 0.22 | 0.37 |

Continued on next page

| Method | Sign | B | $\beta$ | $\gamma$ | $\rho_s$ | $\rho$ | $\epsilon_{-1.0}$ | $\epsilon_{-0.5}$ | $\epsilon_{-0.1}$ | $\epsilon_{0.1}$ | $\epsilon_{0.5}$ | $\epsilon_{1.0}$ |
|---|---|---|---|---|---|---|---|---|---|---|---|---|
| | | | | 0.7 | 0.35 | 0.37 | -0.48 | -0.26 | -0.07 | 0.02 | 0.29 | 0.58 |
| | | 10 | 0.75 | 0.3 | 0.51 | 0.52 | -0.66 | -0.26 | -0.03 | 0.12 | 0.36 | 0.73 |
| | | | | 0.5 | 0.48 | 0.49 | -0.71 | -0.36 | -0.09 | 0.09 | 0.39 | 0.90 |
| | | | | 0.7 | 0.30 | 0.31 | -0.73 | -0.40 | -0.08 | 0.09 | 0.46 | 1.01 |
| | | | 1.00 | 0.3 | 0.43 | 0.45 | -0.58 | -0.16 | -0.04 | 0.08 | 0.28 | 0.64 |
| | | | | 0.5 | 0.54 | 0.54 | -0.82 | -0.38 | -0.08 | 0.13 | 0.48 | 1.10 |
| | | | | 0.7 | 0.41 | 0.43 | -0.83 | -0.45 | -0.09 | 0.11 | 0.49 | 1.08 |
| | | | 1.25 | 0.3 | 0.29 | 0.31 | -0.05 | 0.10 | 0.03 | 0.02 | -0.13 | -0.16 |
| | | | | 0.5 | 0.47 | 0.47 | -0.62 | -0.25 | -0.03 | 0.12 | 0.34 | 0.73 |
| | | | | 0.7 | 0.48 | 0.49 | -0.78 | -0.41 | -0.11 | 0.10 | 0.48 | 1.04 |
| | Negative | 5 | 0.75 | 0.3 | 0.84 | 0.83 | -0.71 | -0.48 | -0.10 | 0.17 | 0.78 | 1.57 |
| | | | | 0.5 | 0.86 | 0.86 | -0.73 | -0.49 | -0.12 | 0.15 | 0.74 | 1.60 |
| | | | | 0.7 | 0.81 | 0.81 | -0.77 | -0.51 | -0.11 | 0.16 | 0.81 | 1.70 |
| | | | 1.00 | 0.3 | 0.89 | 0.88 | -0.70 | -0.50 | -0.10 | 0.17 | 0.80 | 1.59 |
| | | | | 0.5 | 0.85 | 0.86 | -0.72 | -0.49 | -0.12 | 0.15 | 0.75 | 1.58 |
| | | | | 0.7 | 0.79 | 0.80 | -0.75 | -0.48 | -0.10 | 0.15 | 0.78 | 1.63 |
| | | | 1.25 | 0.3 | 0.87 | 0.87 | -0.66 | -0.44 | -0.11 | 0.11 | 0.63 | 1.31 |
| | | | | 0.5 | 0.87 | 0.87 | -0.73 | -0.51 | -0.11 | 0.18 | 0.84 | 1.70 |
| | | | | 0.7 | 0.88 | 0.88 | -0.78 | -0.53 | -0.12 | 0.16 | 0.82 | 1.74 |
| | | 10 | 0.75 | 0.3 | 0.80 | 0.80 | -0.58 | -0.46 | -0.09 | 0.20 | 0.83 | 1.70 |
| | | | | 0.5 | 0.85 | 0.86 | -0.66 | -0.49 | -0.13 | 0.15 | 0.82 | 1.79 |
| | | | | 0.7 | 0.82 | 0.82 | -0.78 | -0.57 | -0.14 | 0.18 | 0.90 | 1.91 |
| | | | 1.00 | 0.3 | 0.80 | 0.80 | -0.51 | -0.45 | -0.08 | 0.19 | 0.85 | 1.72 |
| | | | | 0.5 | 0.83 | 0.85 | -0.64 | -0.47 | -0.10 | 0.18 | 0.84 | 1.74 |
| | | | | 0.7 | 0.80 | 0.81 | -0.72 | -0.51 | -0.11 | 0.17 | 0.85 | 1.80 |
| | | | 1.25 | 0.3 | 0.80 | 0.80 | -0.51 | -0.40 | -0.06 | 0.14 | 0.65 | 1.33 |
| | | | | 0.5 | 0.81 | 0.82 | -0.56 | -0.48 | -0.11 | 0.19 | 0.88 | 1.83 |
| | | | | 0.7 | 0.87 | 0.87 | -0.72 | -0.54 | -0.14 | 0.16 | 0.89 | 1.93 |
| **KM-Sem** | Full | 5 | 0.75 | 0.3 | 0.13 | 0.12 | 0.08 | -0.00 | 0.02 | 0.06 | 0.05 | 0.11 |
| | | | | 0.5 | 0.05 | 0.05 | 0.14 | 0.04 | 0.00 | 0.01 | 0.07 | 0.23 |
| | | | | 0.7 | 0.05 | 0.04 | 0.11 | 0.03 | 0.02 | 0.03 | 0.08 | 0.29 |
| | | | 1.00 | 0.3 | 0.14 | 0.11 | 0.13 | -0.04 | -0.01 | 0.06 | 0.02 | 0.10 |
| | | | | 0.5 | 0.04 | 0.04 | 0.14 | 0.02 | 0.02 | 0.02 | 0.01 | 0.05 |
| | | | | 0.7 | 0.07 | 0.07 | 0.19 | 0.07 | 0.02 | 0.03 | 0.06 | 0.24 |
| | | | 1.25 | 0.3 | 0.05 | 0.07 | -0.04 | -0.04 | 0.02 | 0.03 | 0.01 | -0.03 |
| | | | | 0.5 | 0.08 | 0.07 | 0.15 | 0.01 | 0.02 | 0.05 | -0.00 | 0.06 |
| | | | | 0.7 | 0.00 | 0.01 | 0.20 | 0.06 | 0.02 | 0.01 | 0.07 | 0.24 |
| | | 10 | 0.75 | 0.3 | 0.04 | 0.04 | -0.03 | -0.05 | -0.02 | 0.06 | 0.05 | 0.23 |
| | | | | 0.5 | -0.01 | -0.02 | 0.06 | -0.01 | -0.03 | 0.03 | 0.08 | 0.38 |
| | | | | 0.7 | 0.04 | 0.04 | 0.08 | -0.04 | -0.01 | 0.04 | 0.13 | 0.52 |
| | | | 1.00 | 0.3 | 0.05 | 0.02 | -0.01 | -0.12 | -0.02 | 0.05 | 0.03 | 0.30 |
| | | | | 0.5 | -0.01 | -0.02 | 0.06 | -0.04 | -0.02 | 0.02 | 0.01 | 0.15 |
| | | | | 0.7 | 0.03 | 0.03 | 0.08 | 0.01 | -0.00 | 0.04 | 0.09 | 0.41 |
| | | | 1.25 | 0.3 | -0.02 | -0.03 | -0.23 | -0.06 | -0.01 | -0.00 | -0.03 | 0.05 |
| | | | | 0.5 | -0.02 | -0.02 | 0.09 | -0.05 | -0.02 | -0.00 | -0.03 | 0.10 |
| | | | | 0.7 | -0.00 | -0.01 | 0.15 | 0.02 | -0.03 | 0.03 | 0.07 | 0.44 |
| | Positive | 5 | 0.75 | 0.3 | -0.04 | -0.04 | 0.08 | 0.04 | 0.04 | -0.01 | -0.01 | -0.03 |
| | | | | 0.5 | -0.09 | -0.09 | 0.14 | 0.10 | 0.01 | -0.01 | -0.01 | 0.04 |

Continued on next page

| Method | Sign | B | $\beta$ | $\gamma$ | $\rho_s$ | $\rho$ | $\epsilon_{-1.0}$ | $\epsilon_{-0.5}$ | $\epsilon_{-0.1}$ | $\epsilon_{0.1}$ | $\epsilon_{0.5}$ | $\epsilon_{1.0}$ |
|---|---|---|---|---|---|---|---|---|---|---|---|---|
| | | | | 0.7 | -0.15 | -0.15 | 0.16 | 0.10 | 0.03 | 0.00 | -0.01 | 0.02 |
| | | | 1.00 | 0.3 | 0.01 | 0.01 | -0.06 | -0.01 | -0.01 | -0.06 | -0.03 | 0.03 |
| | | | | 0.5 | -0.10 | -0.11 | 0.08 | 0.03 | 0.01 | -0.03 | -0.03 | -0.04 |
| | | | | 0.7 | -0.11 | -0.12 | 0.24 | 0.15 | 0.03 | -0.00 | -0.04 | -0.03 |
| | | | 1.25 | 0.3 | -0.10 | -0.09 | -0.16 | 0.04 | 0.02 | -0.04 | -0.05 | -0.09 |
| | | | | 0.5 | -0.03 | -0.04 | 0.08 | -0.00 | -0.00 | -0.05 | -0.04 | -0.02 |
| | | | | 0.7 | -0.14 | -0.14 | 0.25 | 0.14 | 0.02 | -0.03 | -0.03 | 0.01 |
| | | 10 | 0.75 | 0.3 | 0.01 | 0.01 | 0.04 | 0.04 | 0.03 | 0.03 | 0.04 | 0.16 |
| | | | | 0.5 | -0.09 | -0.09 | 0.21 | 0.12 | 0.01 | -0.00 | 0.02 | 0.20 |
| | | | | 0.7 | -0.05 | -0.05 | 0.21 | 0.07 | 0.01 | 0.01 | 0.09 | 0.31 |
| | | | 1.00 | 0.3 | 0.14 | 0.12 | -0.13 | -0.04 | -0.03 | -0.01 | 0.11 | 0.33 |
| | | | | 0.5 | -0.01 | -0.03 | 0.07 | 0.03 | -0.01 | 0.00 | 0.04 | 0.23 |
| | | | | 0.7 | -0.07 | -0.07 | 0.24 | 0.13 | 0.03 | 0.01 | 0.06 | 0.25 |
| | | | 1.25 | 0.3 | -0.00 | -0.01 | -0.22 | 0.03 | 0.02 | 0.02 | -0.02 | 0.05 |
| | | | | 0.5 | 0.03 | 0.00 | 0.12 | 0.00 | -0.03 | -0.03 | 0.01 | 0.19 |
| | | | | 0.7 | -0.10 | -0.11 | 0.33 | 0.17 | 0.02 | 0.00 | 0.03 | 0.25 |
| | Negative | 5 | 0.75 | 0.3 | 0.23 | 0.25 | -0.21 | -0.11 | -0.02 | 0.08 | 0.21 | 0.43 |
| | | | | 0.5 | 0.19 | 0.19 | -0.19 | -0.13 | -0.04 | 0.05 | 0.22 | 0.50 |
| | | | | 0.7 | 0.26 | 0.28 | -0.17 | -0.13 | -0.03 | 0.04 | 0.24 | 0.60 |
| | | | 1.00 | 0.3 | 0.09 | 0.10 | -0.09 | -0.09 | -0.02 | 0.05 | 0.16 | 0.19 |
| | | | | 0.5 | 0.17 | 0.20 | -0.11 | -0.08 | -0.03 | 0.03 | 0.15 | 0.32 |
| | | | | 0.7 | 0.22 | 0.23 | -0.14 | -0.08 | -0.01 | 0.05 | 0.22 | 0.53 |
| | | | 1.25 | 0.3 | 0.09 | 0.10 | -0.12 | -0.08 | -0.00 | 0.05 | 0.13 | 0.07 |
| | | | | 0.5 | 0.12 | 0.14 | -0.13 | -0.11 | -0.04 | 0.04 | 0.12 | 0.25 |
| | | | | 0.7 | 0.21 | 0.22 | -0.14 | -0.12 | -0.03 | 0.05 | 0.22 | 0.54 |
| | | 10 | 0.75 | 0.3 | 0.18 | 0.19 | -0.12 | -0.10 | 0.00 | 0.04 | 0.21 | 0.60 |
| | | | | 0.5 | 0.09 | 0.10 | -0.08 | -0.06 | -0.02 | 0.02 | 0.19 | 0.59 |
| | | | | 0.7 | 0.23 | 0.24 | -0.09 | -0.10 | -0.02 | 0.02 | 0.23 | 0.64 |
| | | | 1.00 | 0.3 | 0.17 | 0.19 | 0.04 | -0.02 | -0.04 | 0.03 | 0.13 | 0.56 |
| | | | | 0.5 | 0.13 | 0.15 | -0.02 | -0.05 | -0.01 | 0.01 | 0.14 | 0.48 |
| | | | | 0.7 | 0.17 | 0.19 | -0.06 | -0.05 | -0.00 | 0.01 | 0.19 | 0.60 |
| | | | 1.25 | 0.3 | 0.16 | 0.19 | -0.13 | -0.07 | -0.01 | 0.08 | 0.10 | 0.27 |
| | | | | 0.5 | 0.14 | 0.15 | -0.02 | -0.07 | -0.04 | 0.01 | 0.07 | 0.41 |
| | | | | 0.7 | 0.16 | 0.17 | -0.05 | -0.05 | -0.01 | 0.03 | 0.21 | 0.65 |
| **Single** | Full | 5 | 0.75 | 0.3 | 0.83 | 0.84 | -0.81 | -0.44 | -0.07 | 0.14 | 0.66 | 1.29 |
| | | | | 0.5 | 0.82 | 0.83 | -0.80 | -0.47 | -0.12 | 0.11 | 0.64 | 1.45 |
| | | | | 0.7 | 0.71 | 0.73 | -0.82 | -0.48 | -0.12 | 0.12 | 0.68 | 1.51 |
| | | | 1.00 | 0.3 | 0.75 | 0.79 | -0.82 | -0.42 | -0.07 | 0.10 | 0.64 | 1.35 |
| | | | | 0.5 | 0.83 | 0.84 | -0.87 | -0.49 | -0.09 | 0.12 | 0.67 | 1.40 |
| | | | | 0.7 | 0.75 | 0.76 | -0.82 | -0.47 | -0.13 | 0.12 | 0.66 | 1.45 |
| | | | 1.25 | 0.3 | 0.74 | 0.76 | -0.78 | -0.47 | -0.06 | 0.08 | 0.52 | 1.01 |
| | | | | 0.5 | 0.80 | 0.81 | -0.86 | -0.47 | -0.07 | 0.13 | 0.67 | 1.34 |
| | | | | 0.7 | 0.82 | 0.84 | -0.82 | -0.48 | -0.11 | 0.11 | 0.67 | 1.52 |
| | | 10 | 0.75 | 0.3 | 0.82 | 0.83 | -1.26 | -0.73 | -0.11 | 0.20 | 0.91 | 1.70 |
| | | | | 0.5 | 0.74 | 0.74 | -1.24 | -0.77 | -0.14 | 0.17 | 0.89 | 1.87 |
| | | | | 0.7 | 0.65 | 0.65 | -1.27 | -0.76 | -0.17 | 0.19 | 0.94 | 2.04 |
| | | | 1.00 | 0.3 | 0.75 | 0.78 | -1.32 | -0.74 | -0.12 | 0.13 | 0.82 | 1.70 |
| | | | | 0.5 | 0.83 | 0.84 | -1.43 | -0.84 | -0.15 | 0.20 | 1.00 | 1.99 |
| | | | | 0.7 | 0.71 | 0.72 | -1.37 | -0.79 | -0.16 | 0.20 | 0.96 | 1.97 |

| Method | Sign | B | $\beta$ | $\gamma$ | $\rho_s$ | $\rho$ | $\epsilon_{-1.0}$ | $\epsilon_{-0.5}$ | $\epsilon_{-0.1}$ | $\epsilon_{0.1}$ | $\epsilon_{0.5}$ | $\epsilon_{1.0}$ |
|---|---|---|---|---|---|---|---|---|---|---|---|---|
| | | | 1.25 | 0.3 | 0.71 | 0.72 | -0.94 | -0.55 | -0.14 | 0.14 | 0.58 | 1.10 |
| | | | | 0.5 | 0.78 | 0.78 | -1.27 | -0.78 | -0.17 | 0.19 | 0.92 | 1.77 |
| | | | | 0.7 | 0.78 | 0.79 | -1.37 | -0.84 | -0.16 | 0.18 | 0.99 | 2.08 |
| | Positive | 5 | 0.75 | 0.3 | 0.46 | 0.48 | -0.50 | -0.22 | -0.03 | 0.04 | 0.29 | 0.43 |
| | | | | 0.5 | 0.39 | 0.40 | -0.41 | -0.20 | -0.06 | 0.02 | 0.23 | 0.43 |
| | | | | 0.7 | 0.20 | 0.21 | -0.40 | -0.22 | -0.05 | 0.04 | 0.24 | 0.48 |
| | | | 1.00 | 0.3 | 0.36 | 0.39 | -0.47 | -0.18 | -0.04 | -0.04 | 0.17 | 0.29 |
| | | | | 0.5 | 0.47 | 0.49 | -0.54 | -0.29 | -0.04 | 0.03 | 0.36 | 0.58 |
| | | | | 0.7 | 0.34 | 0.36 | -0.46 | -0.23 | -0.06 | 0.04 | 0.28 | 0.57 |
| | | | 1.25 | 0.3 | 0.25 | 0.27 | -0.23 | -0.13 | 0.02 | 0.00 | 0.05 | -0.02 |
| | | | | 0.5 | 0.40 | 0.43 | -0.46 | -0.23 | -0.03 | 0.02 | 0.23 | 0.35 |
| | | | | 0.7 | 0.33 | 0.35 | -0.45 | -0.24 | -0.06 | 0.01 | 0.28 | 0.54 |
| | | 10 | 0.75 | 0.3 | 0.45 | 0.47 | -0.61 | -0.21 | -0.01 | 0.08 | 0.33 | 0.65 |
| | | | | 0.5 | 0.44 | 0.45 | -0.69 | -0.34 | -0.07 | 0.07 | 0.33 | 0.79 |
| | | | | 0.7 | 0.29 | 0.30 | -0.70 | -0.37 | -0.07 | 0.09 | 0.42 | 0.93 |
| | | | 1.00 | 0.3 | 0.41 | 0.43 | -0.55 | -0.17 | -0.05 | 0.07 | 0.28 | 0.61 |
| | | | | 0.5 | 0.53 | 0.53 | -0.86 | -0.36 | -0.09 | 0.12 | 0.46 | 1.05 |
| | | | | 0.7 | 0.43 | 0.44 | -0.84 | -0.42 | -0.09 | 0.11 | 0.50 | 1.07 |
| | | | 1.25 | 0.3 | 0.28 | 0.29 | 0.01 | 0.10 | 0.03 | 0.00 | -0.08 | -0.10 |
| | | | | 0.5 | 0.45 | 0.45 | -0.65 | -0.26 | -0.04 | 0.11 | 0.35 | 0.68 |
| | | | | 0.7 | 0.46 | 0.47 | -0.70 | -0.39 | -0.09 | 0.07 | 0.44 | 0.97 |
| | Negative | 5 | 0.75 | 0.3 | 0.83 | 0.83 | -0.71 | -0.46 | -0.11 | 0.15 | 0.79 | 1.58 |
| | | | | 0.5 | 0.85 | 0.85 | -0.72 | -0.48 | -0.13 | 0.13 | 0.75 | 1.61 |
| | | | | 0.7 | 0.80 | 0.81 | -0.77 | -0.49 | -0.11 | 0.14 | 0.82 | 1.72 |
| | | | 1.00 | 0.3 | 0.88 | 0.88 | -0.71 | -0.49 | -0.12 | 0.15 | 0.81 | 1.63 |
| | | | | 0.5 | 0.86 | 0.86 | -0.74 | -0.48 | -0.13 | 0.12 | 0.77 | 1.58 |
| | | | | 0.7 | 0.78 | 0.78 | -0.73 | -0.47 | -0.11 | 0.13 | 0.77 | 1.61 |
| | | | 1.25 | 0.3 | 0.86 | 0.87 | -0.64 | -0.44 | -0.10 | 0.10 | 0.62 | 1.31 |
| | | | | 0.5 | 0.86 | 0.86 | -0.73 | -0.49 | -0.12 | 0.17 | 0.84 | 1.68 |
| | | | | 0.7 | 0.88 | 0.88 | -0.78 | -0.52 | -0.13 | 0.14 | 0.83 | 1.76 |
| | | 10 | 0.75 | 0.3 | 0.80 | 0.80 | -0.58 | -0.46 | -0.10 | 0.17 | 0.83 | 1.69 |
| | | | | 0.5 | 0.85 | 0.86 | -0.67 | -0.50 | -0.13 | 0.13 | 0.82 | 1.75 |
| | | | | 0.7 | 0.81 | 0.82 | -0.77 | -0.56 | -0.14 | 0.14 | 0.86 | 1.86 |
| | | | 1.00 | 0.3 | 0.80 | 0.80 | -0.51 | -0.44 | -0.09 | 0.17 | 0.86 | 1.71 |
| | | | | 0.5 | 0.83 | 0.84 | -0.64 | -0.48 | -0.11 | 0.15 | 0.85 | 1.72 |
| | | | | 0.7 | 0.80 | 0.81 | -0.73 | -0.53 | -0.10 | 0.13 | 0.83 | 1.76 |
| | | | 1.25 | 0.3 | 0.79 | 0.80 | -0.51 | -0.40 | -0.06 | 0.12 | 0.65 | 1.31 |
| | | | | 0.5 | 0.80 | 0.80 | -0.53 | -0.44 | -0.11 | 0.20 | 0.89 | 1.79 |
| | | | | 0.7 | 0.86 | 0.87 | -0.75 | -0.55 | -0.14 | 0.14 | 0.90 | 1.90 |

The single attribution vector steering and $RD$ method is near same, since we observe that, top-{5, 10} are nearly identical among continuations in the same cluster (93% for top-5 and 95% overlap with top-10 features in the same cluster.).

