# OpenReview forum: "Shared Semantics, Divergent Mechanisms: Unsupervised Feature Discovery by Aligning Semantics and Mechanisms"
_ICML.cc/2026/Conference — ICML 2026 spotlight_

### Official Review · Reviewer_f1iC · 2026-03-12

**Soundness:** 3
**Presentation:** 2
**Significance:** 3
**Originality:** 3
**Overall Recommendation:** 4
**Confidence:** 3

**Summary:**

This paper introduces a framework for analyzing the distribution of possible continuations generated by an LLM for a given prompt. The method samples multiple responses and represents each continuation using both a semantic embedding and a mechanistic attribution vector that reflects internal model activity. It then applies a rate–distortion–based clustering objective to group responses that are similar in meaning and produced by similar internal mechanisms. The resulting clusters reveal different behavioral modes of the model for the same prompt. The authors further validate these clusters by steering model activations to increase the likelihood of specific response groups.

**Compliance With Llm Reviewing Policy:**

Affirmed.

**Final Justification:**

The authors have substantially improved empirical coverage and clarified the method. Given the novelty and potential impact, I lean toward weak acceptance.

**Key Questions For Authors:**

1. How interpretable were the clusters overall? Do you think it would be feasible to use a state-of-the-art LLM for labeling the clusters?
2. How was the dimension of the attribution vector (d_{a}​) determined? Why were all the features in the attribution vector not used for causal validation?
3. What are your thoughts on applying the proposed method to better understand the chain-of-thought in reasoning models?

**Limitations:**

Refer to the weaknesses.

**Strengths And Weaknesses:**

**Strengths**
1. This work addresses an important gap in the current mechanistic interpretability literature, which largely focuses on analyzing single model outputs, primarily due to the lack of robust techniques for multi-token output analysis.
2. The key insight that similar semantic outputs can arise from varied internal mechanisms is particularly noteworthy.
3. The idea of leveraging joint embeddings, semantic information, and transcoder features is compelling.
4. Although the proposed clustering method based on the Rate-Distortion formulation is somewhat complex, the limited empirical results suggest that it is highly effective.

**Weaknesses**
1. My primary concern is the limited empirical results demonstrating the efficacy of the proposed framework. It is evaluated on only a single model and benchmark, i.e., AmbigQA, which requires the model to generate different kinds of outputs. Consequently, it remains unclear how much the results depend on this specific dataset. For instance, would we observe similar findings, i.e. semantically similar outputs do not correspond to similar features, on a different dataset like MMLU?
2. The evaluation primarily compares the method against single-view clustering baselines (e.g., K-means on semantic or attribution features separately). Since the proposed approach explicitly integrates both representations, these comparisons may favor it by design. It also lacks simple fusion baselines (e.g., clustering on concatenated semantic and attribution features), making it unclear whether the observed improvements arise from the rate–distortion formulation or merely from combining the two feature types.
3. It is also unclear how the proposed method would complement the existing circuit analysis pipeline.
4. While causal validation results are included, their impact seems limited, as they are somewhat expected given that the attribution scores are computed using gradients.
5. The presentation could be improved in several ways:
   - Including a single overview diagram early in the paper would help readers quickly understand the workflow before delving into technical details.
   - Discussing how the proposed method integrates with existing circuit analysis pipelines would be valuable.
   - Providing insights on the computational cost, dependence on attribution quality, and generalization across prompts would strengthen the evaluation.
   - The paper would benefit from a dedicated related works section mentioning existing works, such as [1, 2] and others, which steer models toward particular goals by analyzing multiple generated texts via sampling.

[1] - Zur et. al. “Are language models aware of the road not taken? Token-level uncertainty and hidden state dynamics”, 2025.

[2] - Prakash et. al, “Constructive Circuit Amplification: Improving Math Reasoning in LLMs via Targeted Sub-Network Updates”, 2025.

---

> ### Author Rebuttal · Authors · 2026-03-31
>
> **[W1] Limited experiment scale**
>
> We extended our experiment on Gemma3- IT 1B/4B (on AmbigQA) and a new dataset, HarmBench (50 sets).
> We affirm that Harmbench clusters also provide interesting findings, splitting continuations into 1) poisonous plants with everyday names, 2) more taxonomic (e.g.“*crowbane (Strychnos nux-vomica)*”) and 3) refusals. And steering correlation records (RD, K-SEM, Single) = (0.270, 0.204, 0.243) for $\beta=1, \gamma=0.5$ .
> Below are clustering results ($D_{\gamma}$) on AmbigQA+Gemma4B. Due to the text limit, we will provide more details in the revision.
>
> |$\gamma$|$\beta$|RD|KM-S|KM-A|
> |-|-|-|-|-|
> |0.3|0.50|4.81|6.62|4.88|
> |0.3|0.75|3.31|5.58|3.73|
> |0.3|1.00|2.78|5.18|3.41|
> |0.5|0.50|4.70|5.50|4.42|
> |0.5|0.75|3.20|4.58|3.32|
> |0.5|1.00|2.52|4.10|2.79|
> |0.7|0.50|3.76|3.79|3.27|
> |0.7|0.75|3.03|3.43|2.82|
> |0.7|1.00|2.43|3.09|2.40|
>
> ---
>
> **[W2] Baselines**
>
> We believe that the important property of clustering in interpretability is to provide the explicit trade-off knob between two views to practitioners, rather than just combining two. By design, our RD clustering shows this trade-off by providing $\gamma$ with explicitly separated two distortion terms. While existing methods, such as Co-Reg [1], optimize with ‘agreement terms’ not dual criteria (i.e. two distortion terms), under assumption that two views are shared. As a result, when two views diverge, they do not provide the interpretable knob for which view is sacrificed.
> Our new experiments support that RD is indeed better to optimize our objective than the existing multi-view clustering method: (1) Co-Reg has varying distortion balance within the same $\lambda$, while RD has correlation with $\gamma$. (2) RD outperforms reducing loss.
>
> |$\lambda$|$\gamma$|RD$D^{(e)}$|RD$D^{(a)}$|CoReg$D^{(e)}$|CoReg$D^{(a)}$|$\Delta L$|
> |-|-|-|-|-|-|-|
> |0.1|0.1|0.105|5.13|0.09|5.90|0.73|
> |0.1|0.5|0.133|6.90|0.11|7.62|0.39|
> |0.1|0.9|0.193|16.35|0.18|16.41|-0.10|
> |1|0.1|0.105|5.13|0.10|5.38|0.29|
> |1|0.5|0.133|6.90|0.12|7.40|0.27|
> |1|0.9|0.193|16.35|0.18|16.36|-0.01|
> (trend holds for larger $\lambda$)
>
> ---
>
> **[W3] Complementing circuit analysis**
>
> Due to limited space, please refer to our response to koLi’s Q1.
>
> ---
>
> **[W4] Causal validation**
>
> In Table 2, we consider the baseline ‘single’, which steers the random vector within the cluster unlike our method that steers the medoid of the cluster. Although the steering vector in ‘single’ is also computed using gradient and attribution score, it significantly underperforms. This result implies that not every continuation in the cluster is guaranteed to have its gradient-based attribution score that is causally relevant to move the overall continuations in the same cluster.
>
> ---
>
> **[W5] Presentation**
>
> Thanks! We will incorporate these comments in the revision.
>
> ---
>
> **[Q1] Cluster interpretability**
>
> We think the state-of-the-art LLM could find the pattern in the assigned continuations. For instance, the interpretation example in Section 3.2 is similarly drawn by GPT 5.4, given the assigned continuation texts.
>
> ---
>
> **[Q2] Dimension of attribution vector**
>
> We set the attribution by choosing the top attribution magnitude among all possible nodes. Pruning with a high attribution score is common in circuit analysis [1], since a small set of high-influencing nodes is often ‘sufficient’ to capture the most effect. This claim is feasible, since the influence is computed by multiplying gradients and activations, where activations are sparse.
>
> ---
>
> **[Q3] Reasoning models**
>
> We conducted experiments on reasoning tasks (MATH500/GSM8K) with reasoning models Qwen3-0.6B and 1.7B. Following [2], we extend our attribution analysis as follows.
> - Generation: Sample 64 reasoning steps, delimited by ‘\n\n’. We choose the top-1 step to prepend to input for the next step.
> - Analysis: Pair all possible tuples of step (i, j) (i < j) and compute attribution from forward pass of step i to each logit in step j.
>
> Due to time and hardware constraint, we subsample each dataset with 16 and max step 12, resulting in 300 tuples in total.
> For both models, our method outperforms single-view K-means baselines with matched rate. The below table is from Qwen3-0.6B, and 1.7B model shows a the similar trend.
>
> ||$\gamma$|$\beta=0.50$|$\beta=0.75$|$\beta=1.00$|$\beta=1.25$|
> |-|-|-|-|-|-|
> |RD|0.10|2.56|2.50|2.25|1.97|
> ||0.30|1.80|2.07|1.93|1.78|
> ||0.50|1.13|1.35|1.51|1.40|
> |KM-S|0.10|3.11|3.09|2.86|2.53|
> ||0.30|2.32|2.51|2.38|2.23|
> ||0.50|1.31|1.72|1.81|1.70|
> |KM-A|0.10|2.57|2.55|2.33|2.06|
> ||0.30|1.80|2.10|1.97|1.84|
> ||0.50|1.11|1.36|1.52|1.42|
> We also conducted steering experiments Qwen3-0.6B have correlation (RD, K-means, Single) = (0.037, -0.09, 0.021), and (0.014,0.004, -0.028) for 1.7B in $\beta=0.5$, $\gamma=0.5$, showing RD surpasses baselines. We will scale and add full results in revision.
>
> ---
>
> [1] Circuit Tracing: Revealing Computational Graphs in Language Models
> [2] Thought Anchors: Which LLM Reasoning Steps Matter?

---

> > ### Author Rebuttal · Reviewer_f1iC · 2026-04-02
> >
> > The authors have made a good effort to address my concerns by adding experiments on additional models, datasets, and reasoning tasks, which strengthens the empirical support. However, some key issues remain only partially addressed. In particular, the evaluation still lacks simple fusion baselines (e.g., concatenation of features), making it difficult to isolate the benefit of the rate-distortion formulation itself. Moreover, there is still no strong evidence of generalization to standard benchmarks (e.g., MMLU). Finally, the integration with existing circuit analysis pipelines remains unclear.

---

> > > ### Author Response · Authors · 2026-04-08
> > >
> > > **[Q1] Simple fusion baselines**
> > >
> > > We compared the weighted simple concatenation method to ours. Due to text limit, please refer to our reply to the rebuttal comment of koLi.
> > >
> > > **[Q2] MMLU evaluation**
> > >
> > > We additionally evaluate Qwen3-4B/8B on a 209-question MMLU validation subset covering international law, logical fallacies, moral disputes, philosophy, professional psychology, sociology, and world religions. We analyze continuations generated from raw questions. The qualitative results on clustering and steering for Qwen3-4B are as follows.
> > >
> > > **Clustering ($D_{\\gamma}$)**
> > > |Method|$\gamma$|$\beta=0.50$|$\beta=0.75$|$\beta=1.25$|$\beta=1.50$|
> > > |---|---:|---:|---:|---:|---:|
> > > |RD|0.30|7.78|6.10|5.12|5.07|
> > > |RD|0.50|6.76|5.37|4.02|3.76|
> > > |RD|0.70|5.00|4.25|3.17|2.87|
> > > |KM-S|0.30|29.00|25.58|23.75|23.66|
> > > |KM-S|0.50|22.77|20.21|17.46|17.10|
> > > |KM-S|0.70|14.62|13.91|11.98|11.36|
> > > |KM-A|0.30|23.17|19.03|17.14|17.10|
> > > |KM-A|0.50|19.43|15.98|12.77|12.40|
> > > |KM-A|0.70|13.06|12.00|9.45|8.64|
> > >
> > > **Steering**
> > > |Method|Metric|$\beta$=0.75,$\gamma$=0.3|$\beta$=0.75,$\gamma$=0.5|$\beta$=0.75,$\gamma$=0.7|$\beta$=1.25,$\gamma$=0.3|$\beta$=1.25,$\gamma$=0.5|$\beta$=1.25,$\gamma$=0.7|$\beta$=1.50,$\gamma$=0.3|$\beta$=1.50,$\gamma$=0.5|$\beta$=1.50,$\gamma$=0.7|
> > > |---|---|---|---|---|---|---|---|---|---|---|
> > > |RD|$\rho_s$|0.322|0.218|0.217|0.328|0.347|0.190|0.359|0.336|0.219|
> > > |RD|$\rho$|0.253|0.166|0.164|0.263|0.284|0.148|0.288|0.268|0.175|
> > > |KM-S|$\rho_s$|0.014|0.004|0.002|0.085|0.072|0.012|0.086|0.062|0.027|
> > > |KM-S|$\rho$|0.011|0.004|-0.001|0.066|0.059|0.009|0.068|0.050|0.018|
> > > |Single|$\rho_s$|0.322|0.216|0.212|0.328|0.347|0.181|0.359|0.336|0.215|
> > > |Single|$\rho$|0.254|0.166|0.160|0.263|0.284|0.142|0.288|0.268|0.174|
> > >
> > > And Qwen3-8B,
> > >
> > > **Clustering ($D_{\\gamma}$)**
> > > |Method|$\gamma$|$\beta=0.50$|$\beta=0.75$|$\beta=1.25$|$\beta=1.50$|
> > > |---|---:|---:|---:|---:|---:|
> > > |RD|0.30|7.36|5.85|4.75|4.61|
> > > |RD|0.50|6.21|5.11|3.84|3.57|
> > > |RD|0.70|4.43|3.94|3.11|2.74|
> > > |KM-S|0.30|28.06|25.19|23.16|22.98|
> > > |KM-S|0.50|21.55|19.59|17.35|16.77|
> > > |KM-S|0.70|13.63|13.15|11.80|11.26|
> > > |KM-A|0.30|23.83|20.10|17.59|17.40|
> > > |KM-A|0.50|19.12|16.58|13.58|12.92|
> > > |KM-A|0.70|12.52|11.75|9.96|9.18|
> > >
> > > **Steering**
> > > |Method|Metric|$\beta$=0.75,$\gamma$=0.3|$\beta$=0.75,$\gamma$=0.5|$\beta$=0.75,$\gamma$=0.7|$\beta$=1.25,$\gamma$=0.3|$\beta$=1.25,$\gamma$=0.5|$\beta$=1.25,$\gamma$=0.7|$\beta$=1.50,$\gamma$=0.3|$\beta$=1.50,$\gamma$=0.5|$\beta$=1.50,$\gamma$=0.7|
> > > |---|---|---|---|---|---|---|---|---|---|---|
> > > |RD|$\rho_s$|0.012|0.050|0.026|0.067|0.009|0.036|0.097|0.070|0.036|
> > > |RD|$\rho$|-0.011|0.020|-0.017|0.011|-0.010|0.002|0.028|0.022|0.007|
> > > |KM-S|$\rho$|-0.009|-0.024|-0.016|0.007|-0.001|-0.019|0.015|-0.008|-0.045|
> > > |KM-S|$\rho$|0.002|-0.022|-0.018|0.005|0.004|-0.009|0.000|0.004|-0.032|
> > > |Single|$\rho_s$|0.020|0.059|0.023|0.084|0.057|0.044|0.118|0.090|0.036|
> > > |Single|$\rho$|-0.010|0.032|-0.002|0.026|0.032|0.022|0.054|0.044|0.009|
> > >
> > > Overall, Qwen3-4B results show clear outperformance, Qwen3-8B results show good performance in clustering, and the single baseline is comparable to ours. The omitted trade-off capability between two distortions varying gamma shows the similar trend as the main table in a paper.
> > >
> > > **[Q3] Integration with existing circuit analysis**
> > >
> > > To clarify the possible integration with existing circuit analysis, we provide an example  operational pipeline. The example of integrated pipeline is as follows:
> > > 1. Prepare a dataset or prompt set.
> > > 2. Choose a semantic embedding model and an attribution method. In this work, we use embeddinggemma for semantics and EAP-style attribution, though alternatives such as Qwen3Guard[1] for safety analysis and EAP-IG[2] are also applicable.
> > > 3. Perform RD clustering to select a suitable $(\beta, \gamma)$ trade-off between granularity and distortion.
> > > 4. Inspect and label the resulting clusters; these cluster features can serve as supernodes, similar to prior circuit-analysis work.
> > > 5. Use the chosen attribution method to connect edges between nodes or supernodes.
> > >
> > > This pipeline yields a circuit similar to prior approaches, but with a key difference: the target is not a single token node, but one or more virtual nodes representing an entire continuation or a cluster of continuations.
> > >
> > > We view this as a complement to existing circuit analysis for two reasons.
> > > - First, it extends analysis from single tokens to multi-token continuations, which better captures semantics in natural, in-the-wild data. This addresses a challenge noted in prior work [3], including the “crux” token issue.
> > > - Second, because similar semantics may arise from different mechanisms and vice versa, our clustering method introduces tunable controls that reduce distortion and enable more robust steering.
> > >
> > > ---
> > >
> > > [1] H. Zhao, et al., Qwen3Guard Technical Report, 2025
> > > [2] M. Hanna, et al., Have Faith in Faithfulness: Going Beyond Circuit Overlap When Finding Model Mechanisms, COLM 2024
> > > [3] Lindsey, et al., On the Biology of a Large Language Model, Transformer Circuits, 2025

---

### Official Review · Reviewer_p9VJ · 2026-03-13

**Soundness:** 3
**Presentation:** 3
**Significance:** 3
**Originality:** 3
**Overall Recommendation:** 5
**Confidence:** 4

**Summary:**

The paper proposes **distribution-level unsupervised feature discovery** for mechanistic interpretability of LLMs. Instead of the standard circuit analysis setup (one prompt + one chosen target completion), the authors analyze the *full continuation distribution* of a prompt. They utilize rate-distortion theory to develop a method leveraging both semantics (via EmbeddingGemma) and mechanisms (via sparse feature attributions from transcoders). The resulting clustering objective is optimized EM-style to cluster continuations. Experiments are reported on AmbigQA with Qwen3-8B.

**Compliance With Llm Reviewing Policy:**

Affirmed.

**Final Justification:**

**The authors addressed my concerns in detail in their rebuttal, thus, I recommend acceptance.**

**Key Questions For Authors:**

1. The method is framed as "unsupervised." However, the dual representation relies on a pretrained sentence encoder and a pretrained transcoder, both of which encode substantial learned structure. Could one argue this constitutes a self-supervised signal rather than a purely unsupervised one? I acknowledge the boundary between both is not sharp, but a brief discussion would be informative.

2. What are the key limitations of the transcoder-based approach for capturing underlying mechanisms?

3. Recent work by Arad et al., 2025 ("SAEs Are Good for Steering — If You Select the Right Features") demonstrates that input and output features of sparse autoencoders have distinct roles in steering. Since transcoders explicitly map MLP inputs to outputs, this distinction is directly relevant. How might differentiating between input-side and output-side transcoder features affect the steering results in Section 4? A robustness analysis along this axis would strengthen the causal claims.

**Limitations:**

Some of them are addressed, but the limitations section in the appendix is not referenced in the discussion. See my points in the weaknesses section above

**Strengths And Weaknesses:**

I do believe this paper proposes an interesting contribution. If my concerns (see below) are addressed, I am open to increase my score.

**The authors addressed my concerns in detail in their rebuttal, thus, I recommend acceptance.**

In summary:
- addressing the causality/limitations/result generalization/experimental concerns would already change my score to recommend acceptance, and also improve my soundness and significance claims
- addressing the presentation claims would improve my presentation score

# Strengths

1. **Timely and well-motivated problem.** Extending mechanistic interpretability from single-target circuit analysis to distribution-level analysis addresses a genuine limitation of current methods. The observation that semantically equivalent completions can arise from divergent internal mechanisms (Figure 1) is compelling and well-illustrated.

2. **Principled methodological foundation.** Grounding the clustering objective in rate-distortion theory provides a theoretically motivated framework with interpretable hyperparameters ($\beta$ for granularity, $\gamma$ for view balance), rather than relying on ad hoc clustering choices.

3. **Complementary quantitative and qualitative evaluation.** The paper combines rate-matched distortion comparisons (Table 1) with qualitative analysis of $\beta$-annealing hierarchies (Figure 2), Split/Junk case studies (Figures 4–5), and causal validation via feature steering (Table 2). This multi-faceted evaluation strengthens the claims.

4. **Distribution-level analysis.** Shifting from instance-level to distribution-level mechanistic analysis has the potential to yield more robust and generalizable findings about model internals, as it avoids brittle dependence on individual target completions.

5. **Causal evidence beyond correlation.** The steering experiments (Section 4) go beyond correlational analysis and provide *initial* causal evidence that the discovered cluster-level mechanistic signatures correspond to controllable computational factors. See the weaknesses below for caveats regarding the strength of this claim.

## Weaknesses

### Major points

**[If addressed, improves my overall score]**

1. **Algorithmic presentation lacks accessibility.** The algorithmic description in the main text (Section 2) is detailed but difficult to follow without pseudocode. Algorithm 1 is provided only in Appendix A.6. I recommend moving the pseudocode to the main text and delegating some of the derivation details (e.g., the weighted median computation, PCA-based split initialization) to the appendix.
	1. **An overview "Figure 1" of the method is missing:** providing a flowchart or similar (perhaps as an alternative to the pseudocode) would significantly improve accessibility
**[If addressed, improves presentation]**

3. **Figure and table captions are not self-contained.** Several captions rely on undefined symbols or abbreviations, making them difficult to interpret independently. Specific examples:
	- Table 1: "KM-S" (K-means semantic) and "KM-A" (K-means attribution) are not defined in the caption.
	- Figures 2, 4, 5: Notation such as $C_i^k$ is introduced only in the running text.
	- Appendix Figures 6–8 and Table 3 would likewise benefit from self-contained captions. For Figure 6 in particular, it is unclear what constitutes "high" attribution magnitude — normalizing the y-axis or providing a reference scale would aid interpretation.
**[If addressed, improves presentation]**

4. **Theorem labels are misleading.** The labels "Theorem 2.1" and "Theorem 2.2" are misleading: these statements posit desiderata for when to split or remove clusters, not theorems in the classical sense. Even from the appendix derivations (A.3–A.4), it remains unclear why these should be *the* criteria for cluster management. I suggest either rephrasing them (e.g., as "Criteria" or "Propositions") or providing a clearer explanation of why these conditions are the appropriate ones.
**[If addressed, improves presentation]**

5. **Causal claims require more careful framing.**
	1. Research question (iii) in the introduction — "Can cluster-level analysis have general causality for clusters?" — is unclear. It is unclear what "causality for clusters" means precisely. I recommend defining this term explicitly upon first use.
	2. The causal validation in Section 4 reports correlations between steering strength $\epsilon$ and demeaned logit change $\Delta L_k^{(e)}$. While this is suggestive, correlation between an intervention parameter and an outcome does not rule out confounders. The paper should explicitly discuss alternative non-causal explanations and add this caveat to the limitations section - the checklist [in this paper](https://arxiv.org/abs/2602.16698) could be useful for thus.
**[If addressed, improves soundness]**

6. **Convergence result is underspecified.** Proposition 2.3 states that $\{\mathcal{L}_{\text{RD}}^{(t)}\}$ is monotonically non-increasing and converges, but it is not stated to what quantity convergence occurs (to 0 I presume). It thus remains unclear whether there is, e.g., bias in the converged solution, whether it is a local or global minimum, or how sensitive it is to initialization. Additionally, the proof in Appendix A.5 invokes the monotone convergence theorem by name but does not provide a formal citation.
**[If addressed, improves soundness/presentation]**

7. **Several design choices lack justification.**
	1. The Split and Junk criteria follow from the objective, but the paper does not discuss alternative cluster management strategies or why rate-distortion-derived criteria are preferable to other approaches (e.g., information-theoretic model selection criteria such as BIC).
	2. The choice of non-thinking mode for Qwen3 (Appendix B.1) is mentioned but not motivated. Reasoning traces could plausibly affect both continuation diversity and mechanistic signatures; the rationale for excluding them should be stated.
**[If addressed, improves presentation]**

8. **The conclusion does not reference the limitations.** Given the substantive limitations discussed in Appendix E (computational overhead, feature labeling), the main-text conclusion should explicitly direct the reader to that section. **[If addressed, improves soundness]**

9. **Single dataset, single model family**: All main results are on AmbigQA + Qwen3-8B. How well does this generalize to other tasks (open-ended generation, reasoning, code) and architectures? At least this point should be explicitly discussed in the limitations section. **[If addressed, improves soundness and significance]**

### Minor points

**[If addressed, improves presentation]**

1. **Background on rate-distortion theory is missing.** The paper assumes familiarity with rate-distortion theory but provides no primer. A short paragraph motivating the connection between rate-distortion theory and the clustering problem would improve accessibility for the broader ICML audience.

2. **Notation in Equation (9).** I recommend adding an underbrace labeled "Distortion" spanning the two $D$ terms. While the notation is suggestive, explicitly marking both terms as the distortion component of the objective would improve readability.

3. **Undefined terms.**
	1. The silhouette score (Figure 1) is used without definition or citation.
	2. "Rate-matched" (Section 3.1, Table 1) is a central evaluation concept but is defined only implicitly. An explicit definition upon first use would help.
	3. The term "medoid" should be defined. Medoid (used in Section 4 and Table 2) may not be familiar to all readers. A brief parenthetical definition would suffice.

5. **Qualitative examples on page 5 are difficult to parse.** The motivating examples in Figure 1 and Section 3.2 are valuable, but the latter are dense with cluster notation ($C_i^k$) and numerical Jaccard similarities. I suggest foregrounding the qualitative interpretation and relegating some numerical details to the appendix.

---

> ### Author Rebuttal · Authors · 2026-03-31
>
> **[W1] Low accessibility in algorithm presentation**
>
> We agree that the algorithm section can be more accessible. We will revise Section 2 to emphasize the high-level idea and move technical details to the appendix.
>
> ---
>
> **[W2] Missing definitions in figures and tables**
>
> We will clarify notation and captions throughout and revise Figure 6 to include the additional baselines discussed with Reviewer koLi's W1.
>
> ---
>
> **[W3] Theorem mislabeled**
>
> We will relabel them as propositions and add more intuitive explanations in Appendix A.
>
> ---
>
> **[W4] Scope of causality claim**
>
> We agree that our use of “causality” was underspecified. Under Pearl’s ladder, our claim is at Rung 2 (interventional), since steering demonstrates sufficiency to change model behavior, but not necessity, which would require counterfactual evidence. We will narrow this claim in the Introduction and Section 4, and expand the discussion in the appendix.
>
> ---
>
> **[W5] Convergence underspecfication**
>
> We will clarify that the loss decreases monotonically and converges to a finite value $L^{\star} \ge 0$, so the loss difference goes to zero, in formulas. As with K-means, our method converges only to a local minimum because hard cluster assignments make the problem non-convex.
> To address initialization sensitivity, note that randomness appears only in PCA-vector initialization for Split (Eq. 34). Sweeping 10 random seeds, the final loss statistics are stable (mean 8.65, mean std 0.14, median std 0.08), indicating low sensitivity.
>
> ---
>
> **[W6] Several design choices lack justification**
>
> 1. Our cluster management with Split and Junk is used to optimize the objective with the rate budget controlled by $\beta$. Unlike BIC-style criteria, which select among separately trained clusterings, our goal is to control representation granularity *during* optimization. As Figure 2 shows, varying $\beta$ yields different meaningful granularities. Thus, Split and Junk are integrated design choices aligned with the objective, not post-hoc model selection.
>
> 2. We used non-thinking mode, since AmbigQA is not usually evaluated with reasoning.
>
> ---
>
> **[W7] Missing reference to limitation appendix**
>
> We will add the reference to limitation in conclusion to help readers follow.
>
> ---
>
> **[W8] Single dataset, single model family**
>
> Due to text limit, please refer to our response to f1iC’s W1.
>
> ---
>
> **[P1] Rate-Distortion theory background**
>
> We agree that more background would help. We will add a brief introduction to rate-distortion theory and explain how it motivates our method.
>
> ---
>
> **[P2] Notation**
>
> It is better to pair rate and ‘distortion’, so we will modify the underbrace as you suggested.
>
> ---
>
> **[P3] Clarifying definitions**
>
> We will add the silhouette-score reference, define the relevant terms in the appendix, clarify the relation between rate and $K$ in the main text, and define medoid in a footnote.
>
> ---
>
> **[P4] Parsing numerical details from the figure**
>
> We will revise the Figure 3.2 and its caption, focusing on more high-level interpretation and clearing numbers, leaving details in Appendix.
>
> ---
>
> **[Q1] Notion of unsupervised manner**
>
> We agree that the boundary between self-supervised and unsupervised has become blurry. We used “unsupervised” because recent work [1,2] uses the term for behavior discovery without behavior labels. Still, we agree this distinction is worth discussing more carefully in interpretability settings, especially when methods rely on upstream signals such as pretrained SAEs rather than data themselves.
>
> ---
>
> **[Q2] Limitations of transcoders**
>
> We think the main challenges of transcoders, as well as common SAEs as follows:
>
> 1. They are often local to token position, making it difficult to capture features distributed across multiple tokens or time steps, such as semantic structure [3].
>
> 2. Steering assumes that attention patterns remain sufficiently similar after intervention, so later features behave as in the clean run, which may fail in practice.
>
> ---
>
> **[Q3] Input and output features in transcoders**
>
> After reviewing the cited paper, we understand its “input” and “output” features as role-based notions: 1) features whose logit-lens tokens and input tokens co-occur consistently, and 2) features whose steered output tokens and logit-lens tokens co-occur consistently.
> Because these notions differ from the mechanistic MLP input/output view in transcoders, properly evaluating them would require substantial additional transcoder experiments, we will leave it as a future work.
>
> ---
>
> [1] Z. Zhang, et al., Fantastic Reasoning Behaviors and Where to Find Them: Unsupervised Discovery of the Reasoning Process, 2026
> [2] S. Ghosh, et al.,  A Simple Yet Effective Method for Non-Refusing Context Relevant Fine-grained Safety Steering in LLMs, EMNLP 2025
> [3] U. Bhalla, et al., Temporal Sparse Autoencoders: Leveraging the Sequential Nature of Language for Interpretability, ICLR 2026.

---

> > ### Author Rebuttal · Reviewer_p9VJ · 2026-04-02
> >
> > Thank you so much for the detailed and structured rebuttal and addressing my concerns in depth. I increased my score.

---

> > > ### Author Response · Authors · 2026-04-08
> > >
> > > We highly appreciate your sincere review. Your comments have been very helpful in improving our paper.

---

### Official Review · Reviewer_koLi · 2026-03-13

**Soundness:** 3
**Presentation:** 2
**Significance:** 3
**Originality:** 3
**Overall Recommendation:** 5
**Confidence:** 3

**Summary:**

The paper is focused on a key problem in current mechanistic interpretability work, which is that many of them focus on finding features or circuits corresponding to a specific prompt and specific output. It is unclear if a mechanism discovered on this input-output pair will generalize to other paraphrases of the same prompt or other samples from the same output distribution, let alone broadly to other related inputs. Similarly, for a set of samples from the same output distributions or paraphrases of the same output text, there could be different mechanisms that generate each variant. To address this issue, the authors introduce an unsupervised latent feature discovery method that serves as an output-distribution level explanation. The key insight is to not only use features of the internal model activations (e.g. transcoder or SAE features) but also the semantic information in the output text (using sentence embeddings), and to jointly cluster them to find informative features. The proposed method uses a rate-distortion objective that jointly clusters both the sentence-level and activation-level embeddings, with the number of clusters set automatically (using adaptive Split and Junk operations) and tunable knobs corresponding to cluster granularity and the balance of semantic vs. mechanistic distortions. The result is a set of clusters that are empirically shown to have coherent meaning as well as consistent internal representations, and these clusters can also be used to find meaningful steering vectors.

**Compliance With Llm Reviewing Policy:**

Affirmed.

**Final Justification:**

After reviewing the additional results provided during the rebuttal stage, many of my concerns have been resolved. The linearity argument gives the aggregated vector a cleaner interpretation as an average sequence-level effect, and I appreciate the corresponding narrowing of the "divergent mechanisms" framing, which would be useful to integrate into the final paper. The MMLU Qwen3-8B steering results are weaker than on AmbigQA, which I would also recommend addressing in the text. That said, the empirical results as a whole are much stronger at this point and I have increased my score accordingly.

**Key Questions For Authors:**

Aside from the concerns in the Weaknesses section, I have the following questions:

1. The paper motivates the clustering method as a complement to circuit analysis, but it would be helpful to discuss the connection to concrete downstream use cases. Is this meant to serve as a drop-in replacement for single-feature or supernode-level steering, or are there other ways it would fit into the mech interp workflow?
2. What is the runtime of the clustering method, and how does it compare to K-Means? It appears that cluster discovery has to be re-run for every prompt, so this information will help guide its usage in practice.
3. Was the method run on multiple seeds? If so, how stable are the final cluster assignments across different random seeds?
4. The steering analysis in Table 2 reports logit changes only for the target cluster. When steering toward cluster k, what happens to ΔL for the other clusters? A summary statistic like mean ΔL across non-target clusters would help clarify whether the discovered clusters correspond to selectively controllable mechanisms, or whether steering produces diffuse effects across multiple clusters.

**Limitations:**

Yes

**Strengths And Weaknesses:**

### Strengths

1. The core observation that semantically similar output text, even from the same logit distribution, can be produced using very different internal mechanisms (and vice versa) is useful and important. The experiments also show that clustering on semantic embeddings does a poor job when steering, implying that semantic similarity is a poor proxy for mechanistic similarity.
2. The rate distortion framework appears principled and well-executed, and the heirarchical structure revealed in the experiments is compelling: finer clusters inherit mechanistic signatures from their parents instead of subdividing arbitrarily, and the interpretable sub-clusters (e.g. "knowledge cutoff", "hedging") are illustrative.
### Weaknesses

**Aggregation in the mechanistic embeddings:**
The approach produces a single "attribution vector" for each continuation using the full set of transcoder features. To do this, they aggregate the attribution score across all output token positions to obtain one scalar per transcoder feature, and select the top $d_a$ features based on this to obtain a vector. Typically in circuit analysis, the set of top activating features for token position $t$ are very different from the set of top activating features at position $t+1$, and it is the interaction structure across these features that result in meaningful circuits. The defense of this approach in Appendix C is limited:
1. They show the L1 norm of the attribution vector at token position 31 is still more than half of the peak value, but this just says that "prefix features generally matter for later tokens",  not "the same prefix features matter in the same way for all continuation tokens". It would be helpful to show a measure of consistency across the token positions, e.g. cosine similarities or overlap of the attribution vectors at different token positions.
2. The bar for "causal validation" is essentially that steering the aggregation of the top-10 features is better than steering the aggregation of 10 random features, which appears to be a strawman. The claim that needs support isn't that "pooled features are better than random", it is that "pooled features can discriminate between continuations that use different mechanisms". The paper would be stronger with more informative baselines, e.g. using attribution vectors at the first, last, or concatenating the first few token positions instead of aggregating across time.

**Multi-view clustering evaluation**
The rate-distortion framework is elegant, but compared with single-view methods as baselines instead of other multi-view clustering algorithms or just concatenating the two embeddings (semantic and mechanistic) and running K-means. As is, the comparisons show that when evaluated on a multi-view objective, single-view methods have lower scores. This is useful for the opening argument that sentence embeddings and feature embeddings can have very different clusterings, but not for the validation of the clustering approach itself. It is unclear whether existing, simpler multi-view clustering algorithms are compatible with the fundamental approach, or if the proposed rate-distortion approach is necessary to obtain the specific advantages claimed in the paper.

**Evaluation Datasets**
The entire evaluation is focused on a single dataset, AmbigQA. It would be helpful to include other datasets, particularly from existing mechanistic interpretability papers. It would be particularly interesting if the clusters correspond to components that have been independently identified through traditional circuit analysis, or find insights that existing papers missed.

---

> ### Author Rebuttal · Authors · 2026-03-31
>
> **[W1] Aggregation in mechanistic embeddings**
>
> Since our object of interest is the *full continuation span* from *prefix*, we believe the aggregation is a natural summary of its attribution pattern. However, we have further analyzed for deeper insight.
> - **Token-wise attribution similarity.** For a single continuation, we compared attribution vectors across token positions using L1 distance and Jaccard similarity over the top-100 features. As a baseline, we also compared the target continuation with a random continuation from the same prefix. Similarity is locally consistent, but degrades when an earlier token is involved. It falls below the random when token position exceeds around 7 for Jaccard and 10 for L1.
> - **Prefix vs. continuation influence.** We separately computed attribution from prefix tokens and prior continuation tokens to the target token. Prefix attribution has larger L1 magnitude up to position 13, after which continuation attribution becomes larger. By position 30, the influence ratio is about 4:6. This suggests that later predictions are increasingly mediated by the continuation, though the span does not sharply diverge overall.
> - **Steering at different positions/spans.** We tested steering on the full continuation span, the first token, the last token, and the first five tokens. The full span and first-five-token span show a clear monotonic increase in demeaned logit difference as steering strength increases. The first-token and last-token settings show positive correlation only for strengths between -1 and 0.1, then weaken beyond that. This suggests that span-level aggregation captures a compositional effect across continuation tokens.
>
> Overall, we still consider aggregated attribution over continuation positions to be a reasonable representation of *prefix-to-continuation influence*, especially given the steering results. We will add these results and revise the text to state this claim more precisely.
>
> ---
>
> **[W2] Multi-view clustering evaluation**
>
> Due to limited space, please refer to our response to f1iC’s W2.
>
> ---
>
> **[W3] Evaluation datasets**
>
> Due to limited space, please refer to our response to f1iC’s W1.
>
> ---
>
> **[Q1] Complementing existing circuit analysis**
>
> Our method is most closely related to supernode steering at the distribution level, and complements prior work in two ways.
> First, we identify features from continuations in an **unsupervised manner**, whereas prior work [2] discovers them from token activation patterns using large external datasets. For example, in Figure 2, we can label the supernode feature associated with “explaining genre-based groups” by grouping the top-activated features in the cluster attribution center.
> Second, our method enables **more robust steering**. Table 2 shows a stronger causal effect than steering with a single random vector from the same cluster. This suggests that failure of an arbitrary vector does not imply failure of the underlying cluster feature, which is a limitation of prior approaches.
>
> ---
>
> **[Q2] Runtime of the clustering method**
>
> Our method’s E, M steps are fairly equivalent to K-means optimization. So the Split and Junk and iteration matters. We sample 50 clustering results, where the final cluster count K = {3,5,10} and the mean wall clock time for each step is as follows:
> |Step|K=3|K=5|K=10|
>  |:---|:---|:---|:---|
> |E+M|0.45s|0.80s|1.82s|
> |Split|3.30s|5.46s|6.93s|
> |Junk|0.14s|0.38s|0.49s|
> These results show that **Split** is the main bottleneck, while the relative cost of **E+M** increases with larger K.
>
> ---
>
> **[Q3] Randomness in clustering**
>
> Our randomness only occurs in PCA vector initialization for Split (Eq.34). To observe the effect of randomness of initializing vectors, we analyzed 100 Split operations during the main experiment (Qwen3-8B, AmbigQA). Computing pairwise Jaccard similarity between seeds and averaged over all pairs with all samples, the similarity score is 0.94, which is significantly high.
>
> ---
>
> **[Q4] Feature specificity**
>
> To evaluate cross-cluster steering specificity, we randomly selected 30 AmbigQA samples with Qwen3-8B and compared our method with the “single” baseline. Let $\mathrm{Corr}_{i,j}$ denote the correlation obtained when steering with the direction from cluster (i) and evaluating on target cluster (j). For each cluster (i), we define:
>
> 1. **Active:** $\mathrm{Corr}\_{i,i} > \max_j \mathrm{Corr}\_{i,j}$
>
> 2. **Passive:** $\mathrm{Corr}\_{i,i} > \max_j \mathrm{Corr}\_{j,i}$
>
> RD clustering achieves **Active** 0.0217 and **Passive** 0.1894, while the Single baseline achieves 0.0289 and 0.0111. These results suggest that our method causes slightly more cross-cluster interference during steering, but provides substantially better cluster discrimination during readout.
>
> ---
>
> [1] A. Kumar, et al., Co-regularized Multi-view Spectral Clustering, NIPS 2011
> [2] Circuit Tracing: Revealing Computational Graphs in Language Models

---

> > ### Author Rebuttal · Reviewer_koLi · 2026-04-04
> >
> > Thank you for the detailed and responsive rebuttal, particularly the new experiments with additional datasets and cluster steering results. These substantially strengthen the empirical foundation of the paper. I have a few follow-up questions:
> >
> > 1. The token-wise similarity analysis shows that per-position attribution vectors diverge fairly quickly (below the random baseline) by position 7-10, which raises concerns about aggregating across all positions. Given that the per-position vectors being aggregated are themselves inconsistent, could you clarify what the aggregated vector is meant to capture about the per-position attribution pattern, and how this reconciles with the framing around divergent "mechanisms"?
> >
> > 2. I agree with Reviewer f1iC that a simple concatenation baseline would help with isolating the contribution of the rate-distortion formulation itself, though the rebuttal notes that this loses the "tunable knob". Would a weighted concatenation not satisfy this design motivation?

---

> > > ### Author Response · Authors · 2026-04-08
> > >
> > > **[Q1] Clarification for attribution aggregation**
> > >
> > > The aggregated vector should be interpreted as capturing the average sequence-level effect of per-position attribution patterns, rather than implying a consistent attribution trend across all continuation positions. This distinction does not weaken our claim. Our target quantity is different from that of single-token attribution analyses. Specifically, we define the effect on demeaned sequence logits and show, by linearity of gradient-based attribution, that the resulting sequence-level effect is exactly the average of token-level effects. Formally,
> > >
> > > following the notations of effect in [1,2],
> > > let $ j \equiv (i,l,f) $ index a prefix transcoder feature and define token-wise logit,
> > > $$\\tilde{\\ell}\_{y\_t}:=\\ell\_{y\_t}-\\bar{\\ell}\_{y\_t},\\qquad \\bar{\\ell}\_{y\_t}:=\\frac{1}{|V|}\\sum\_{v\\in V}\\ell\_v.$$
> > >
> > > then,
> > > $$\\tau\_{(i,l,f),t}:=\\tilde{\\ell}\_{y\_t}\\left(x\_{\\mathrm{clean}}\\mid do(F\_{i,l,f}=1)\\right)-\\tilde{\\ell}\_{y\_t}\\left(x\_{\\mathrm{clean}}\\mid do(F\_{i,l,f}=0)\\right).$$
> > >
> > > $$\\tilde{\\ell}^{\\mathrm{mean}}\_{1:T}:=\\frac{1}{T}\\sum\_{t=1}^T \\tilde{\\ell}\_{y\_t},\\qquad \\tau\_{(i,l,f)}^{\\mathrm{mean}}:=\\tilde{\\ell}^{\\mathrm{mean}}\_{1:T}\\left(x\_{\\mathrm{clean}}\\mid do(F\_{i,l,f}=1)\\right)-\\tilde{\\ell}^{\\mathrm{mean}}\_{1:T}\\left(x\_{\\mathrm{clean}}\\mid do(F\_{i,l,f}=0)\\right).$$
> > > $$\\begin{aligned}
> > > \\tau\_{(i,l,f)}^{\\mathrm{mean}}
> > > &=\\left(\\frac{1}{T}\\sum\_{t=1}^T \\tilde{\\ell}\_{y\_t}\\left(x\_{\\mathrm{clean}}\\mid do(F\_{i,l,f}=1)\\right)\\right) \\\\
> > > &\\quad-\\left(\\frac{1}{T}\\sum\_{t=1}^T \\tilde{\\ell}\_{y\_t}\\left(x\_{\\mathrm{clean}}\\mid do(F\_{i,l,f}=0)\\right)\\right) \\\\
> > > &=\\frac{1}{T}\\sum\_{t=1}^T \\tau\_{(i,l,f),t}.
> > > \\end{aligned}$$
> > >
> > > Second, several prior works [3,4] also aggregate attribution signals across output tokens.
> > > Taken together, the divergence of mechanisms shown in Fig. 1 should be interpreted as divergence in the sequence-level effects of features on output logits, rather than as evidence that each position is derived by a single shared token-level attribution pattern.
> > >
> > > **[Q2] Multi-view clustering baseline**
> > >
> > > We compare our rate-distortion method against a weighted-concatenation baseline with fused embedding:
> > > $$[ \alpha e_i ; \(1-\alpha) a_i]$$
> > > , and both embeddings are normalized and optimized by K-means with L2 norm. We examine three questions: whether the baseline can match our objective, whether tuning $\alpha$ provides a meaningful distortion trade-off, and whether it yields comparably strong causal correlation.
> > > For the first question, we pair a ($\beta$, $\gamma$) cluster result with a single ($K$, $\lambda$ or $\alpha$) with the smallest difference in rate (matched rate).
> > > |$\beta$|$\gamma$|RD$D_{\gamma}$|RD$L_{\mathrm{RD}}$|Concat$D_{\gamma}$|Concat$L_{\mathrm{RD}}$|
> > > |---:|---:|---:|---:|---:|---:|
> > > |0.50|0.30|8.28|9.49|9.40|10.67|
> > > |0.50|0.50|7.10|7.80|7.38|8.25|
> > > |0.50|0.70|5.12|5.36|4.80|5.45|
> > > |0.75|0.30|6.63|8.56|8.06|9.89|
> > > |0.75|0.50|5.73|7.08|6.49|7.88|
> > > |0.75|0.70|4.53|5.11|4.59|5.40|
> > > |1.00|0.30|5.94|8.24|7.62|9.64|
> > > |1.00|0.50|4.90|6.76|5.90|7.68|
> > > |1.00|0.70|3.95|4.94|4.28|5.38|
> > > |1.25|0.30|5.58|8.09|7.49|9.59|
> > > |1.25|0.50|4.46|6.63|5.64|7.61|
> > > |1.25|0.70|3.49|4.87|3.95|5.36|
> > > |1.50|0.30|5.41|8.03|7.38|9.52|
> > > |1.50|0.50|4.18|6.57|5.46|7.52|
> > > |1.50|0.70|3.18|4.86|3.79|5.44|
> > >
> > > Next, we observe two distortions varying hyperparameters for Concat. The result shows that there is almost zero effect of balancing between two distortions.
> > >
> > > |K|$\alpha=0.1$|$\alpha=0.3$|$\alpha=0.5$|$\alpha=0.7$|$\alpha=0.9$|
> > > |---|---|---|---|---|---|
> > > |2|0.23/15.26|0.23/15.25|0.23/15.25|0.21/15.53|0.20/16.81|
> > > |4|0.21/12.52|0.21/12.52|0.20/12.50|0.18/13.11|0.17/15.23|
> > > |6|0.20/11.34|0.19/11.32|0.19/11.35|0.17/12.20|0.15/14.67|
> > > |8|0.19/10.49|0.18/10.47|0.18/10.52|0.15/11.53|0.14/14.10|
> > > |10|0.18/9.88|0.18/9.87|0.17/9.94|0.14/11.06|0.13/13.68|
> > >
> > > Lastly, we observe steering results (top-5 features, 200 samples). The result shows that our method has more clear correlation.
> > >
> > > **Concat**
> > >
> > > |$K$|$\alpha$|$\rho_s$|$\rho$|
> > > |---|---|---|---|
> > > |4|0.3|0.13|0.13|
> > > |4|0.5|0.13|0.15|
> > > |4|0.7|0.08|0.09|
> > > |6|0.3|0.11|0.12|
> > > |6|0.5|0.17|0.18|
> > > |6|0.7|0.03|0.04|
> > > |8|0.3|0.10|0.10|
> > > |8|0.5|0.06|0.06|
> > > |8|0.7|0.03|0.04|
> > >
> > > **Ours**
> > >
> > > |$\beta$|$\gamma$|$\rho_s$|$\rho$|
> > > |---|---|---|---|
> > > |0.5|0.3|0.24|0.27|
> > > |0.5|0.5|0.27|0.29|
> > > |0.5|0.7|0.28|0.30|
> > > |0.75|0.3|0.20|0.23|
> > > |0.75|0.5|0.30|0.33|
> > > |0.75|0.7|0.26|0.28|
> > > |1|0.3|0.18|0.23|
> > > |1|0.5|0.20|0.23|
> > > |1|0.7|0.22|0.25|
> > > |1.25|0.3|0.23|0.27|
> > > |1.25|0.5|0.25|0.28|
> > > |1.25|0.7|0.19|0.21|
> > > |1.5|0.3|0.32|0.37|
> > > |1.5|0.5|0.32|0.36|
> > > |1.5|0.7|0.21|0.23|
> > >
> > > ---
> > >
> > > [1] Attribution Patching Outperforms Automated Circuit Discovery
> > > [2] Sparse Feature Circuits: Discovering and Editing Interpretable Causal Graphs in Language Models
> > > [3] Can Input Attributions Explain Inductive Reasoning in In-Context Learning?
> > > [4] Towards Long-Horizon Interpretability: Efficient and Faithful Multi-Token Attribution for Reasoning LLMs

---

### Decision · Program_Chairs · 2026-04-30

**Decision:**

Accept (spotlight)

**Comment:**

The core idea of the paper is on point, which is that circuit analysis is interesting in-so-far as it enables out ability to achieve a generalized understanding of how neural networks function, and that understanding how a neural network functions on a single input is fundamentally limited. The paper is well-written, the experiments are well designed, and the results are convincing!

All reviewers agree we should accept the paper, and the engaged responses from the author seemed to have improved to work.